# The origins and spread of domestic horses from the Western Eurasian steppes

Domestication of horses fundamentally transformed long-range mobility and warfare[1]. However, modern domesticated breeds do not descend from the earliest domestic horse lineage associated with archaeological evidence of bridling, milking and corralling[2-4] at Botai, Central Asia around 3500 BC[3]. Other longstanding candidate regions for horse domestication, such as Iberia[5] and Anatolia[6], have also recently been challenged. Thus, the genetic, geographic and temporal origins of modern domestic horses have remained unknown. Here we pinpoint the Western Eurasian steppes, especially the lower Volga-Don region, as the homeland of modern domestic horses. Furthermore, we map the population changes accompanying domestication from 273 ancient horse genomes. This reveals that modern domestic horses ultimately replaced almost all other local populations as they expanded rapidly across Eurasia from about 2000 BC, synchronously with equestrian material culture, including Sintashta spoke-wheeled chariots. We find that equestrianism involved strong selection for critical locomotor and behavioural adaptations at the *GSDMC* and *ZFPM1* genes. Our results reject the commonly held association[7] between horseback riding and the massive expansion of Yamnaya steppe pastoralists into Europe around 3000 BC[8,9] driving the spread of Indo-European languages[10]. This contrasts with the scenario in Asia where Indo-Iranian languages, chariots and horses spread together, following the early second millennium BC Sintashta culture[11,12].

We gathered horse remains encompassing all suspected domestication centres, including Iberia, Anatolia and the steppes of Western Eurasia and Central Asia (Fig 1a). The sampling targeted previously under-represented time periods, with 201 radiocarbon dates spanning 44426 to 202 BC, and five beyond 50250 to 47950 BC (Supplementary Table 1).

The DNA quality enabled shotgun sequencing of 264 ancient genomes at 0.10× to 25.76× average coverage (239 genomes above 1× coverage), including 16 genomes for which further sequencing added to previously reported data. Enzymatic[13] and computational removal of post mortem DNA damage produced high-quality data with derived mutations decreasing with sample age, as expected if mutations accumulate through time (Extended Data Fig. 1). We added ten published modern genomes, and nine ancient genomes characterized with consistent technology or covering relevant time periods and locations, to obtain the most extensive high-quality genome time series for horses.

## Pre-domestication population structure

Neighbour-joining phylogenomic inference revealed four geographically defined monophyletic groups (Fig 1b). These closely mirrored clusters identified using an extension of the Struct-f4 method[5] (Fig 1d–f, Extended Data Fig. 2, Supplementary Methods), except for the Neolithic Anatolia group (NEO-ANA), where the tree-to-data goodness of fit suggested phylogenetic misplacement (Fig 1c, Supplementary Methods).

The most basal cluster included *Equus lenensis* (ELEN), a lineage identified in northeastern Siberia from the Late Pleistocene to the late fourth millennium BC[5,14,15]. A second group covered Europe, including Late Pleistocene Romania, Belgium, France and Britain, and the region from Spain to Scandinavia and Hungary, Czechia and Poland during the sixth-to-third millennium BC. The third cluster comprised the earliest known domestic horses from Botai and Przewalski's horses, as previously reported[3], and extended to the Altai and Southern Urals during the fifth-to-third millennium BC. Finally, modern domestic horses clustered within a group that became geographically widespread and prominent following about 2200 BC and during the second millennium BC (DOM2). This cluster appears genetically close to horses that lived in the Western Eurasia steppes (WE) but not further west than the Romanian lower Danube, south of the Carpathians, before and during the third millennium BC. Significant correlation between genetic and geographic distances, and inference of limited long-distance connectivity with estimated effective migration surface[16] (EEMS), confirmed the strong geographic differentiation of horse populations before about 3000 BC (Fig 2a, Extended Data Fig. 3a).

Horse ancestry profiles in Neolithic Anatolia and Eneolithic Central Asia, including at Botai, maximized a genetic component (coloured green in Fig. 1e, f) that was also substantial in Central and Eastern Europe during the Late Pleistocene (RONPC06_Rom_m34801) and the fourth or third millennium BC (Figs. 1e, 3a, Extended Data Fig. 4). It was, however, absent or moderately present in the Romanian lower Danube (ENEO-ROM), the Dnieper steppes (Ukr11_Ukr_m4185) and the western lower Volga-Don (C-PONT) populations during the sixth to third millennia BC. This indicates possible expansions of Anatolian horses into both Central and Eastern Europe and Central Asia regions, but not into the Western Eurasia steppes. The absence of typical NEO-ANA ancestry rules out expansion from Anatolia into Central Asia across the Caucasus mountains but supports connectivity south of the Caspian Sea prior to about 3500 BC.

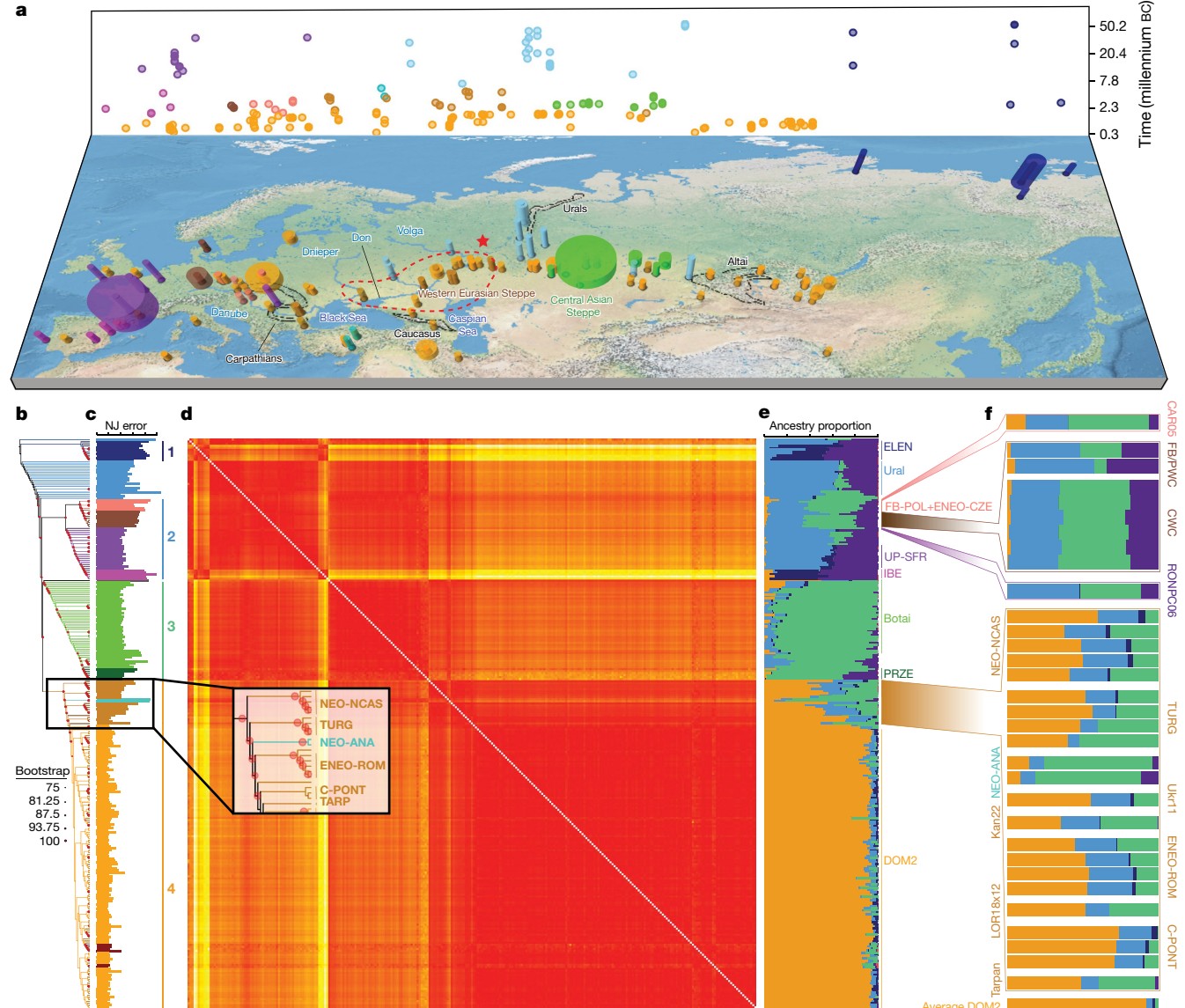

**Fig. 1 | Ancient horse remains and their genomic affinities. a,** Temporal and geographic sampling. The red star indicates the location of the two TURG horses (late Yamnaya context) showing genetic continuity with DOM2. The dashed line indicates the inferred homeland of DOM2 horses in the lower Volga-Don region. Colours refer to regions and/or time periods delineating genetically close horses. The radius of each cylinder is proportional to the number of samples analysed (for <10 specimens; radius constant above this), and the height refers to the time range covered. **b,** Neighbour-joining phylogenomic tree (100 bootstrap pseudo-replicates). Samples are coloured according to **a** and the main phylogenetic clusters are numbered from 1 to 4. **c,** Fold difference between neighbour-joining-based and raw pairwise genetic distances. **d,** Pairwise distance matrix of Struct-f4 genetic affinities between samples. Increasing genetic affinities are indicated by a yellow-to-red gradient. **e,** Struct-f4 ancestry component profiles. **f,** Ancestry profiles of selected key horse groups and samples. PRZE, Przewalski; UP-SFR, Upper Palaeolithic Southern France.

## The origins of DOM2 horses

The C-PONT group not only possessed moderate NEO-ANA ancestry, but also was the first region where the typical DOM2 ancestry component (coloured orange in Fig. 1e, f) became dominant during the sixth millennium BC. Multi-dimensional scaling further identified three horses from the western lower Volga-Don region as genetically closest to DOM2, associated with Steppe Maykop (Aygurskii), Yamnaya (Repin) and Poltavka (Sosnovka) contexts, dated to about 3500 to 2600 BC (Figs. 2a, b, 3a). Additionally, genetic continuity with DOM2 was rejected for all horses predating about 2200 BC, especially those from the NEO-ANA group (Supplementary Table 2), except for two late Yamnaya specimens from approximately 2900 to 2600 BC (Turganik (TURG)), located further east than the western lower Volga-Don region (Figs. 2a, b, 3a). These may therefore have provided some of the direct ancestors of DOM2 horses.

Modelling of the DOM2 population with qpADM[17], rotating[18] all combinations of 2, 3 or 4 population donors, eliminated the possibility of a contribution from the NEO-ANA population, but indicated possible formation within the WE population, including a genetic contribution of approximately 95% from C-PONT and TURG horses (Supplementary Table 3). This was consistent with OrientAGraph[19] modelling from nine lineages representing key ancestry combinations, which confirmed the absence of NEO-ANA genetic ancestry in DOM2 and confirmed DOM2 as a sister population to the C-PONT horses (Fig. 3b).

Identifying discrete populations and modelling admixture as single unidirectional pulses, however, was highly challenging given the extent of spatial genetic connectivity. Indeed, the typical DOM2 ancestry component was maximized in the C-PONT group, but declined sharply eastwards (TURG and Central Asia) in the third millennium BC as the proportion of NEO-ANA ancestry increased (Fig. 2a). This suggests a

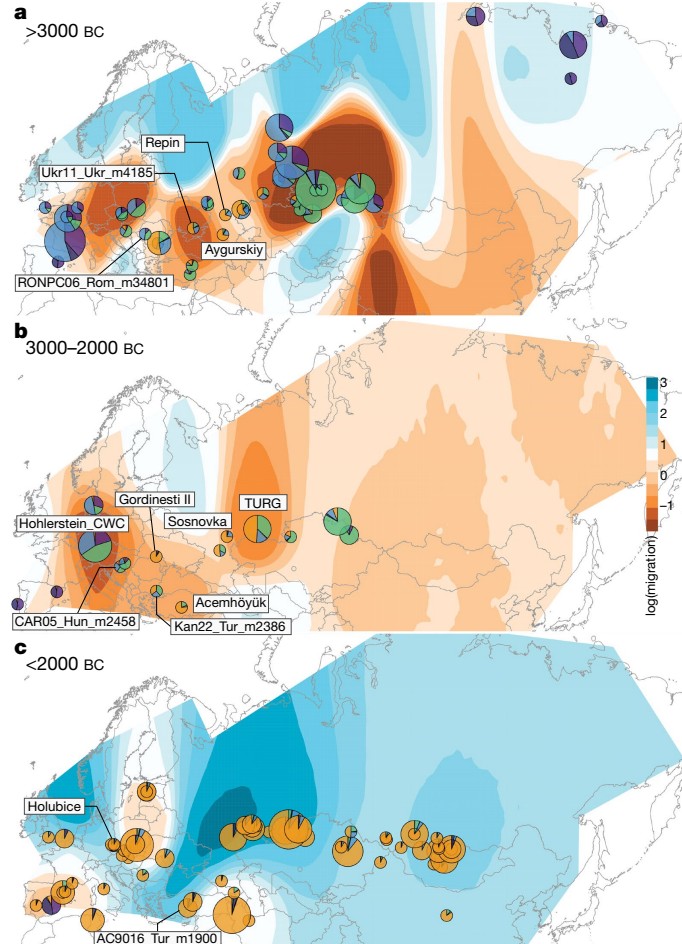

**a** >3000 BC

Repin
Ukr11_Ukr_m4185
Aygurskiy
RONPC06_Rom_m34801

**b** 3000–2000 BC

Gordinesti II
Hohlerstein_CWC
Sosnovka
TURG
Acemhöyük
CAR05_Hun_m2458
Kan22_Tur_m2386

**c** <2000 BC

Holubice
AC9016_Tur_m1900

**Fig. 2 | Horse geographic and genetic affinities. a–c**, EEMS-predicted migration barriers[16] and average ancestry components found in each archaeological site from before 3000 BC (**a**), during the third millennium BC (**b**) and after around 2000 BC (**c**). The size of the pie charts is proportional to the number of samples analysed in a given location (<10, constant above). Pie chart colours refer to $K = 6$ ancestry components, averaged per location. Regions inferred as geographic barriers are shown in shades of brown, and regions affected by migrations are shown in shades of blue. The base map was obtained from rworldmap[46].

cline of genetic connectivity east of the Western Eurasia steppes and Central Asia, ruling out DOM2 ancestors further east than the western lower Volga-Don and Turganik. A similar genetic cline characterized the region located west of C-PONT, where the typical DOM2 ancestry component declined steadily in the Dnieper steppes, Poland, Turkish Thrace and Hungary in the fifth to third millennia BC. This eliminates the possibility of DOM2 ancestors further west than C-PONT and the Dnieper steppes. Furthermore, patterns of spatial autocorrelations in the genetic data[20] indicated Western Eurasia steppes as the most likely geographic location of DOM2 ancestors (Fig. 3c). Combined, our results demonstrate that DOM2 ancestors lived in the Western Eurasia steppes, especially the lower Volga-Don, but not in Anatolia, during the late fourth and early third millennia BC.

## Expansion of steppe-related pastoralism

Analyses of ancient human genomes have revealed a massive expansion from the Western Eurasia steppes into Central and Eastern Europe during the third millennium BC, associated with the Yamnaya culture[8,9,11,12,21]. This expansion contributed at least two thirds of steppe-related

ancestry to populations of the Corded Ware complex (CWC) around 2900 to 2300 BC[8]. The role of horses in this expansion remained unclear, as oxen could have pulled Yamnaya heavy, solid-wheeled wagons[7,22]. The genetic profile of horses from CWC contexts, however, almost completely lacked the ancestry maximized in DOM2 and Yamnaya horses (TURG and Repin) (Figs. 1e, f, 2a, b) and showed no direct connection with the WE group, including both C-PONT and TURG, in OrientAGraph modelling (Fig. 3b, Extended Data Fig. 5).

The typical DOM2 ancestry was also limited in pre-CWC horses from Denmark, Poland and Czechia, associated with the Funnel Beaker and early Pitted Ware cultures (FB/PWC, FB/POL and ENEO-CZE, respectively). DOM2 ancestry reached a maximum 12.5% in one Hungarian horse dated to the mid-third millennium BC and associated with the Somogyvár-Vinkovci Culture (CAR05_Hun_m2458). qpAdm[17] modelling indicated that its DOM2 ancestry was acquired following gene flow from southern Thrace (Kan22_Tur_m2386), but not from the Dnieper steppes (Ukr11_Ukr_m4185) (Supplementary Table 3). Combined with the lack of increased horse dispersal during the early third millennium BC (Fig. 2b, Extended Data Fig. 3b), these results suggest that DOM2 horses did not accompany the steppe pastoralist expansion north of the Carpathians.

By around 2200–2000 BC, the typical DOM2 ancestry profile appeared outside the Western Eurasia steppes in Bohemia (Holubice), the lower Danube (Gordinesti II) and central Anatolia (Acemhöyük), spreading across Eurasia shortly afterwards, eventually replacing all pre-existing lineages (Fig 2c, Extended Data Fig. 3c). Eurasia became characterized by high genetic connectivity, supporting massive horse dispersal by the late third millennium and early second millennium BC. This process involved stallions and mares, indicated by autosomal and X-chromosomal variation (Extended Data Fig. 3d), and was sustained by explosive demographics apparent in both mitochondrial and Y-chromosomal variation (Extended Data Fig. 3e, f). Altogether, our genomic data uncover a high turnover of the horse population in which past breeders produced large stocks of DOM2 horses to supply increasing demands for horse-based mobility from around 2200 BC.

Of note, the DOM2 genetic profile was ubiquitous among horses buried in Sintashta kurgans together with the earliest spoke-wheeled chariots around 2000–1800 BC[7,9,23,24] (Extended Data Fig. 6). A typical DOM2 profile was also found in Central Anatolia (AC9016_Tur_m1900), concurrent with two-wheeled vehicle iconography from about 1900 BC[25,26]. However, the rise of such profiles in Holubice, Gordinesti II and Acemhöyük before the earliest evidence for chariots supports horseback riding fuelling the initial dispersal of DOM2 horses outside their core region, in line with Mesopotamian iconography during the late third and early second millennia BC[27]. Therefore, a combination of chariots and equestrianism is likely to have spread the DOM2 diaspora in a range of social contexts from urban states to dispersed decentralized societies[28].

## DOM2 biological adaptations

Human-induced DOM2 dispersal conceivably involved selection of phenotypic characteristics linked to horseback riding and chariotry. We therefore screened our data for genetic variants that are over-represented in DOM2 horses from the late third millennium BC (Extended Data Fig. 7). The first outstanding locus peaked immediately upstream of the *GSDMC* gene, where sequence coverage dropped at two L1 transposable elements in all lineages except DOM2. The presence of additional exons in other mammals suggests that independent L1 insertions remodelled the DOM2 gene structure. In humans, *GSDMC* is a strong marker for chronic back pain[29] and lumbar spinal stenosis, a syndrome causing vertebral disk hardening and painful walking[30].

The second most differentiated locus extended over approximately 16 Mb on chromosome 3, with the *ZFPM1* gene being closest to the selection peak. *ZFPM1* is essential for the development of dorsal raphe

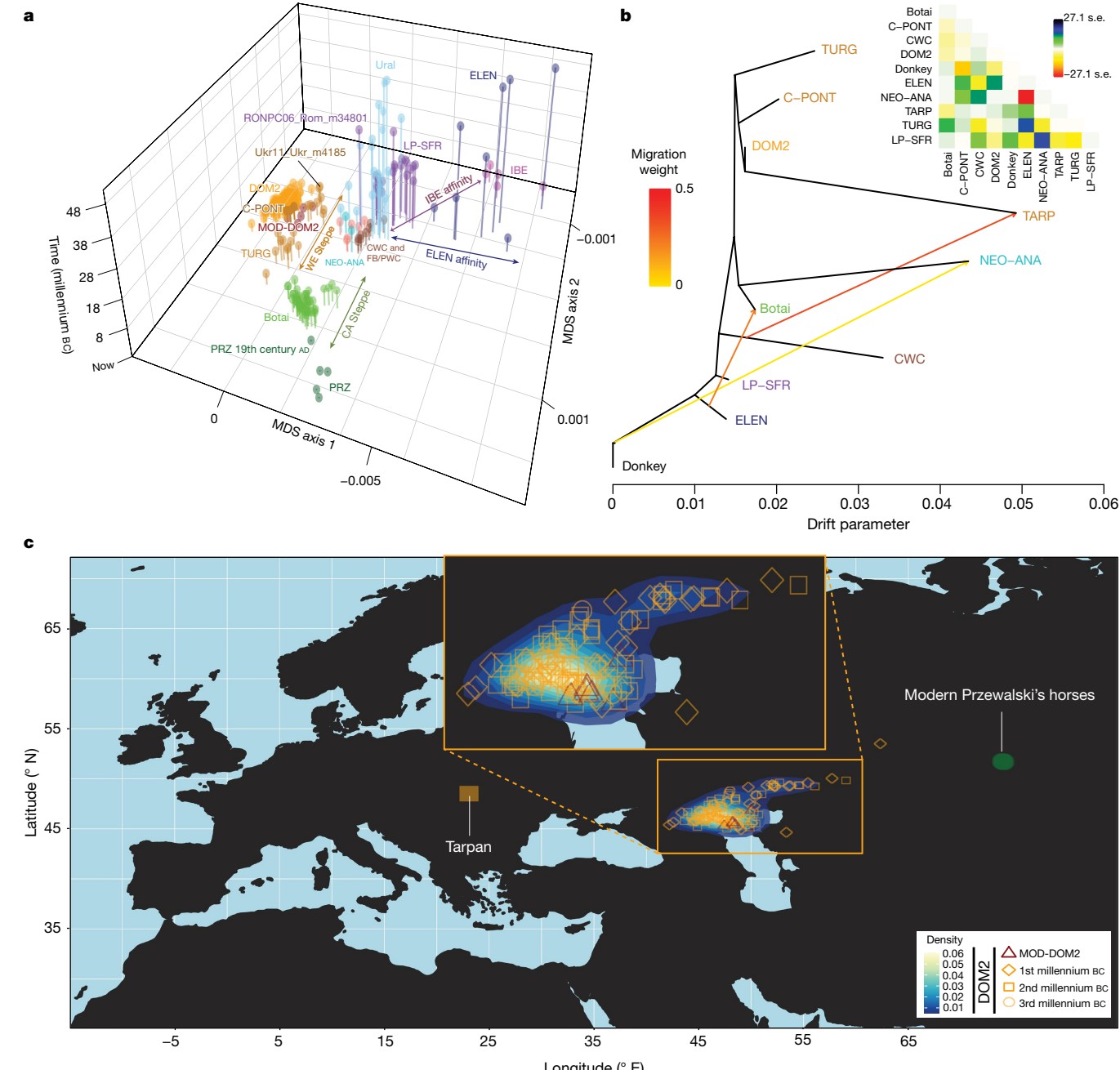

**Fig. 3 | Population genetic affinities, evolutionary history and geographic origins. a**, Multi-dimensional scaling plot of $f_4$-based genetic affinities. The age of the samples is indicated along the vertical axis. CA, Central Asia. **b**, Horse evolutionary history inferred by OrientAGraph[19] with three migration edges and nine lineages representing key genomic ancestries (coloured as in Fig 1a). The model explains 99.99% of the total variance. The triangular pairwise matrix provides model residuals. The external branch leading to donkey was set to zero to improve visualization. **c**, LOCATOR[20] predictions of the geographic region where the ancestors of DOM2, tarpan and modern Przewalski's horses lived. The tarpan and modern Przewalski's horses do not descend from the same ancestral population as modern domestic horses. The map was drawn using the maps R package[47].

serotonergic neurons involved in mood regulation[31] and aggressive behaviour[32]. *ZFPM1* inactivation in mice causes anxiety disorders and contextual fear memory[31]. Combined, early selection at *GSDMC* and *ZFPM1* suggests shifting use toward horses that were more docile, more resilient to stress and involved in new locomotor exercise, including endurance running, weight bearing and/or warfare.

## Evolutionary history and origins of tarpan horses

Our analyses elucidate the geographic, temporal and biological origins of DOM2 horses. This study features a diverse ancient horse genome dataset, revealing the presence of deep mitochondrial and/or Y-chromosomal haplotypes in non-DOM2 horses (Supplementary Fig 1). This suggests that yet-unsampled divergent populations contributed to forming several lineages excluding DOM2. This was especially true in the Iberian group (IBE), where the expected genetic distance to the donkey was reduced (Extended Data Fig. 5f), but also in NEO-ANA according to OrientAGraph modelling (Fig 3b). Disentangling exact divergence and ancestry contributions of such unsampled lineages is difficult with the currently available data. It can, however, be stressed that Iberia and Anatolia represent two well-known refugia[33], where populations could have survived and mixed during Ice Ages.

Finally, our analyses have solved the mysterious origins of the tarpan horse, which became extinct in the early 20th century. The tarpan horse came about following admixture between horses native to Europe (modelled as having 28.8–34.2% and 32.2–33.2% CWC ancestry in OrientAGraph[19] and qpAdm[17], respectively) and horses closely related to DOM2. This is consistent with LOCATOR[20] predicting ancestors in western Ukraine (Fig 3c) and refutes previous hypotheses depicting tarpans as the wild ancestor or a feral version of DOM2, or a hybrid with Przewalski's horses[34].

## Discussion

This work resolves longstanding debates about the origins and spread of domestic horses. Whereas horses living in the Western Eurasia steppes in the late fourth and early third millennia BC were the ancestors of DOM2 horses, there is no evidence that they facilitated the expansion of the human genetic steppe ancestry into Europe[8,9] as previously hypothesized[7]. Instead of horse-mounted warfare, declining populations during the European late Neolithic[35] may thus have opened up an opportunity for a westward expansion of steppe pastoralists. Yamnaya horses at Repin and Turganik carried more DOM2 genetic affinity than presumably wild horses from hunter-gatherer sites of the sixth millennium BC (NEO-NCAS, from approximately 5500–5200 BC), which may suggest early horse management and herding practices. Regardless, Yamnaya pastoralism did not spread horses far outside their native range, similar to the Botai horse domestication, which remained a localized practice within a sedentary settlement system[2,36]. The globalization stage started later, when DOM2 horses dispersed outside their core region, first reaching Anatolia, the lower Danube, Bohemia and Central Asia by approximately 2200 to 2000 BC, then Western Europe and Mongolia soon afterwards, ultimately replacing all local populations by around 1500 to 1000 BC. This process first involved horseback riding, as spoke-wheeled chariots represent later technological innovations, emerging around 2000 to 1800 BC in the Trans-Ural Sintashta culture[7]. The weaponry, warriors and fortified settlements associated with this culture may have arisen in response to increased aridity and competition for critical grazing lands, intensifying territoriality and hierarchy[37]. This may have provided the basis for the conquests over the subsequent centuries that resulted in an almost complete human and horse genetic turnover in Central Asian steppes[11,21]. The expansion to the Carpathian basin[38], and possibly Anatolia and the Levant, involved a different scenario in which specialized horse trainers and chariot builders spread with the horse trade and riding. In both cases, horses with reduced back pathologies and enhanced docility would have facilitated Bronze Age elite long-distance trade demands and become a highly valued commodity and status symbol, resulting in rapid diaspora. We, however, acknowledge substantial spatiotemporal variability and evidential bias towards elite activities, so we do not discount additional, harder to evidence, factors in equine dispersal.

Our results also have important implications for mechanisms underpinning two major language dispersals. The expansion of the Indo-European language family from the Western Eurasia steppes has traditionally been associated with mounted pastoralism, with the CWC serving as a major stepping stone in Europe[39–41]. However, while there is overwhelming lexical evidence for horse domestication, horse-drawn chariots and derived mythologies in the Indo-Iranian branch of the Indo-European family, the linguistic indications of horse-keeping practices at the deeper Proto-Indo-European level are in fact ambiguous[42] (Supplementary Discussion). The limited presence of horses in CWC assemblages[43] and the local genetic makeup of CWC specimens reject scenarios in which horses were the primary driving force behind the initial spread of Indo-European languages in Europe[44]. By contrast, DOM2 dispersal in Asia during the early-to-mid second millennium BC was concurrent with the spread of chariotry and Indo-Iranian languages, whose earliest speakers are linked to populations that directly preceded the Sintashta culture[11,12,45]. We thus conclude that the new package of chariotry and improved breed of horses, including chestnut coat colouration documented both linguistically (Supplementary Discussion) and genetically (Extended Data Fig. 8), transformed Eurasian Bronze Age societies globally within a few centuries after about 2000 BC. The adoption of this new institution, whether for warfare, prestige or both, probably varied between decentralized chiefdoms in Europe and urbanized states in Western Asia. The results thus open up new research avenues into the historical developments of these different societal trajectories.

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

Pablo Librado[1], Naveed Khan[1,121], Antoine Fages[1], Mariya A. Kusliy[1,2], Tomasz Suchan[1,3], Laure Tonasso-Calvière[1], Stéphanie Schiavinato[1], Duha Alioglu[1], Aurore Fromentier[1], Aude Perdereau[4], Jean-Marc Aury[5], Charleen Gaunitz[1], Lorelei Chauvey[1], Andaine Seguin-Orlando[1], Clio Der Sarkissian[1], John Southon[6], Beth Shapiro[7,8], Alexey A. Tishkin[9], Alexey A. Kovalev[10], Saleh Alquraishi[11], Ahmed H. Alfarhan[11], Khaled A. S. Al-Rasheid[11], Timo Seregély[12], Lutz Klassen[13], Rune Iversen[14], Olivier Bignon-Lau[15], Pierre Bodu[15], Monique Olive[15], Jean-Christophe Castel[16], Myriam Boudadi-Maligne[17], Nadir Alvarez[18,19], Mietje Germonpré[20], Magdalena Moskal-del Hoyo[3], Jarosław Wilczyński[21], Sylwia Pospuła[21], Anna Lasota-Kuś[22], Krzysztof Tunia[22], Marek Nowak[23], Eve Rannamäe[24], Urmas Saarma[25], Gennady Boeskorov[26], Lembi Lõugas[27], René Kyselý[28], Lubomír Peške[29], Adrian Bălăşescu[30], Valentin Dumitraşcu[30], Roxana Dobrescu[30], Daniel Gerber[31,32], Viktória Kiss[33], Anna Szécsényi-Nagy[31], Balázs G. Mende[31], Zsolt Gallina[34], Krisztina Somogyi[35], Gabriella Kulcsár[33], Erika Gál[33], Robin Bendrey[36], Morten E. Allentoft[37,38], Ghenadie Sirbu[39], Valentin Dergachev[40], Henry Shephard[41], Noémie Tomadini[42], Sandrine Grouard[42], Aleksei Kasparov[43], Alexander E. Basilyan[44], Mikhail A. Anisimov[45], Pavel A. Nikolskiy[44], Elena Y. Pavlova[45], Vladimir Pitulko[43], Gottfried Brem[46], Barbara Wallner[46], Christoph Schwall[47], Marcel Keller[48,49], Keiko Kitagawa[50,51,52], Alexander N. Bessudnov[53], Alexander Bessudnov[43], William Taylor[54], Jérome Magail[55], Jamiyan-Ombo Gantulga[56], Jamsranjav Bayarsaikhan[57,58], Diimaajav Erdenebaatar[59], Kubatbeek Tabaldiev[60], Enkhbayar Mijiddorj[59], Bazartseren Boldgiv[61], Turbat Tsagaan[56], Mélanie Pruvost[17], Sandra Olsen[62], Cheryl A. Makarewicz[63,64], Silvia Valenzuela Lamas[65], Silvia Albizuri Canadell[66], Ariadna Nieto Espinet[67], Ma Pilar Iborra[68], Jaime Lira Garrido[69,70], Esther Rodríguez González[71], Sebastián Celestino[71], Carmen Olària[72], Juan Luis Arsuaga[70,73], Nadiia Kotova[74], Alexander Pryor[75], Pam Crabtree[76], Rinat Zhumatayev[77], Abdesh Toleubaev[77], Nina L. Morgunova[78], Tatiana Kuznetsova[79,80], David Lordkipanize[81,82], Matilde Marzullo[83], Ornella Prato[83], Giovanna Bagnasco Gianni[83], Umberto Tecchiati[83], Benoit Clavel[42], Sébastien Lepetz[42], Hossein Davoudi[84], Marjan Mashkour[42,84], Natalia Ya. Berezina[85], Philipp W. Stockhammer[86,87], Johannes Krause[49,86], Wolfgang Haak[49,86,88], Arturo Morales-Muñiz[89], Norbert Benecke[90], Michael Hofreiter[91], Arne Ludwig[92,93], Alexander S. Graphodatsky[2], Joris Peters[94,95], Kirill Yu. Kiryushin[9], Tumur-Ochir Iderkhangai[59], Nikolay A. Bokovenko[43], Sergey K. Vasiliev[96], Nikolai N. Seregin[9], Konstantin V. Chugunov[97], Natalya A. Plasteeva[98], Gennady F. Baryshnikov[99], Ekaterina Petrova[100], Mikhail Sablin[99], Elina Ananyevskaya[100], Andrey Logvin[101], Irina Shevnina[101], Victor Logvin[102], Saule Kalieva[102], Valeriy Loman[103], Igor Kukushkin[103], Ilya Merz[104], Victor Merz[104], Sergazy Sakenov[105], Victor Varfolomeyev[103], Emma Usmanova[103], Viktor Zaibert[106], Benjamin Arbuckle[107], Andrey B. Belinskiy[108], Alexej Kalmykov[108], Sabine Reinhold[90], Svend Hansen[90], Aleksandr I. Yudin[109], Alekandr A. Vybornov[110], Andrey Epimakhov[111,112], Natalia S. Berezina[113], Natalia Roslyakova[110], Pavel A. Kosintsev[98,114], Pavel F. Kuznetsov[110], David Anthony[115,116], Guus J. Kroonen[117,118], Kristian Kristiansen[119,120], Patrick Wincker[5], Alan Outram[75] & Ludovic Orlando[1]✉

[1]Centre d'Anthropobiologie et de Génomique de Toulouse, Université Paul Sabatier, Toulouse, France. [2]Department of the Diversity and Evolution of Genomes, Institute of Molecular and Cellular Biology SB RAS, Novosibirsk, Russia. [3]W. Szafer Institute of Botany, Polish Academy of Sciences, Kraków, Poland. [4]Genoscope, Institut de biologie François-Jacob, Commissariat à l'Energie Atomique (CEA), Université Paris-Saclay, Evry, France. [5]Génomique Métabolique, Genoscope, Institut de biologie François Jacob, CEA, CNRS, Université d'Evry, Université Paris-Saclay, Evry, France. [6]Earth System Science Department, University of California, Irvine, Irvine, CA, USA. [7]Department of Ecology and Evolutionary Biology, University of California, Santa Cruz, Santa Cruz, CA, USA. [8]Howard Hughes Medical Institute, University of California, Santa Cruz, Santa Cruz, CA, USA. [9]Department of Archaeology, Ethnography and Museology, Altai State University, Barnaul, Russia. [10]Department of Archaeological Heritage Preservation, Institute of Archaeology of the Russian Academy of Sciences, Moscow, Russia. [11]Zoology Department, College of Science, King Saud University, Riyadh, Saudi Arabia. [12]Institute for Archaeology, Heritage Conservation Studies and Art History, University of Bamberg, Bamberg, Germany. [13]Museum Østjylland, Randers, Denmark. [14]Saxo Institute, section of Archaeology, University of Copenhagen, Copenhagen, Denmark. [15]ArScAn-UMR 7041, Equipe Ethnologie préhistorique, CNRS, MSH-Mondes, Nanterre Cedex, France. [16]Muséum d'histoire naturelle, Secteur des Vertébrés, Geneva, Switzerland. [17]UMR 5199 De la Préhistoire à l'Actuel : Culture, Environnement et Anthropologie (PACEA), CNRS, Université de Bordeaux, Pessac Cedex, France. [18]Geneva Natural History Museum, Geneva, Switzerland. [19]Department of Genetics and Evolution, University of Geneva, Geneva, Switzerland. [20]OD Earth & History of Life, Royal Belgian Institute of Natural Sciences, Brussels, Belgium. [21]Institute of Systematics and Evolution of Animals, Polish Academy of Sciences, Kraków, Poland. [22]Institute of Archaeology and Ethnology Polish Academy of Sciences, Kraków, Poland. [23]Institute of Archaeology, Jagiellonian University, Kraków, Poland. [24]Department of Archaeology, Institute of History and Archaeology, Tartu, Estonia. [25]Department of Zoology, Institute of Ecology and Earth Sciences, University of Tartu, Tartu, Estonia. [26]Diamond and Precious Metals Geology Institute, SB RAS, Yakutsk, Russia. [27]Archaeological Research Collection, Tallinn University, Tallinn, Estonia. [28]Department of Natural Sciences and Archaeometry, Institute of Archaeology of the Czech Academy of Sciences, Prague, Czechia. [29], Prague, Czechia. [30]Vasile Pârvan Institute of Archaeology, Department of Bioarchaeology, Romanian Academy, Bucharest, Romania. [31]Institute of Archaeogenomics, Research Centre for the Humanities, Eötvös Loránd Research Network, Budapest, Hungary. [32]Department of Genetics, Eötvös

Loránd University, Budapest, Hungary. [33]Institute of Archaeology, Research Centre for the Humanities, Eötvös Loránd Research Network, Budapest, Hungary. [34]Ásatárs Ltd., Kecskemét, Hungary. [35]Rippl-Rónai Municipal Museum with Country Scope, Kaposvár, Hungary. [36]School of History, Classics and Archaeology, University of Edinburgh, Old Medical School, Edinburgh, UK. [37]Trace and Environmental DNA (TrEnD) Lab, School of Molecular and Life Sciences, Curtin University, Perth, Western Australia, Australia. [38]Lundbeck Foundation GeoGenetics Centre, GLOBE Institute, University of Copenhagen, Copenhagen, Denmark. [39]Department of Academic Management, Academy of Science of Moldova, Chişinău, Republic of Moldova. [40]Center of Archaeology, Institute of Cultural Heritage, Academy of Science of Moldova, Chişinău, Republic of Moldova. [41]Archaeological Institute of America, Boston, MA, USA. [42]Centre National de Recherche Scientifique, Muséum national d'Histoire naturelle, Archéozoologie, Archéobotanique (AASPE), CP 56, Paris, France. [43]Institute for the History of Material Culture, Russian Academy of Sciences (IHMC RAS), St Petersburg, Russia. [44]Geological Institute, Russian Academy of Sciences, Moscow, Russia. [45]Arctic and Antarctic Research Institute, St Petersburg, Russia. [46]Institute of Animal Breeding and Genetics, University of Veterinary Medicine Vienna, Vienna, Austria. [47]Department of Prehistory and Western Asian/Northeast African Archaeology, Austrian Archaeological Institute, Austrian Academy of Sciences, Vienna, Austria. [48]Estonian Biocentre, Institute of Genomics, University of Tartu, Tartu, Estonia. [49]Department of Archaeogenetics, Max Planck Institute for the Science of Human History, Jena, Germany. [50]SFB 1070 Resource Cultures, University of Tübingen, Tübingen, Germany. [51]Department of Early Prehistory and Quaternary Ecology, University of Tübingen, Tübingen, Germany. [52]UMR 7194 Muséum National d'Histoire Naturelle, CNRS, UPVD, Paris, France. [53]Semenov-Tyan-Shanskii Lipetsk State Pedagogical University, Lipetsk, Russia. [54]Museum of Natural History, University of Colorado-Boulder, Boulder, CO, USA. [55]Musée d'Anthropologie préhistorique de Monaco, Monaco, Monaco. [56]Institute of Archaeology, Mongolian Academy of Sciences, Ulaanbaatar, Mongolia. [57]Department of Archaeology, Max Planck Institute for the Science of Human History, Jena, Germany. [58]Chinggis Khaan Museum, Ulaanbaatar, Mongolia. [59]Department of Archaeology, Ulaanbaatar State University, Ulaanbaatar, Mongolia. [60]Department of History, Kyrgyz-Turkish Manas University, Bishkek, Kyrgyzstan. [61]Department of Biology, National University of Mongolia, Ulaanbaatar, Mongolia. [62]Division of Archaeology, Biodiversity Institute, University of Kansas, Lawrence, KS, USA. [63]Institute for Prehistoric and Protohistoric Archaeology, Kiel University, Kiel, Germany. [64]ROOTS Excellence Cluster, Kiel University, Kiel, Germany. [65]Archaeology of Social Dynamics, Institució Milà i Fontanals d'Humanitats, Consejo Superior de Investigaciones Científicas (IMF-CSIC), Barcelona, Spain. [66]Departament d'Història i Arqueologia–SERP, Universitat de Barcelona, Barcelona, Spain. [67]Grup d'Investigació Prehistòrica, Universitat de Lleida, PID2019-110022GB-I00, Lleida, Spain. [68], Valencia, Spain. [69]Departamento de Medicina Animal, Facultad de Veterinaria, Universidad de Extremadura, Cáceres, Spain. [70]Centro Mixto UCM-ISCIII de Evolución y Comportamiento Humanos, Madrid, Spain. [71]Instituto de Arqueología (CSIC–Junta de Extremadura), Mérida, Spain. [72]Laboratori d'Arqueologia Prehistòrica, Universitat Jaume I, Castelló de la Plana, Spain. [73]Departamento de Geodinámica, Estratigrafía y Paleontología, Facultad de Ciencias Geológicas, Universidad Complutense de Madrid, Madrid, Spain. [74]Department of Eneolithic and Bronze Age, Institute of Archaeology National Academy of Sciences of Ukraine, Kyiv, Ukraine. [75]Department of Archaeology, University of Exeter, Exeter, UK. [76]Center for the Study of Human Origins, Anthropology Department, New York University, New York, NY, USA. [77]Department of Archaeology, Ethnology and Museology, Al Farabi Kazakh National University, Almaty, Kazakhstan. [78]Scientific Research Department, Orenburg State Pedagogical University, Orenburg, Russia. [79]Department of paleontology, Faculty of Geology, Moscow State University, Moscow, Russia. [80]Institute of Geology and Petroleum Technologies, Kazan Federal University, Kazan, Russia. [81]Georgian National Museum, Tbilisi, Georgia. [82]Tbilisi State University, Tbilisi, Georgia. [83]Università degli Studi di Milano, Dipartimento di Beni Culturali e Ambientali, Milan, Italy. [84]University of Tehran, Central Laboratory, Bioarchaeology Laboratory, Archaeozoology Section, Tehran, Iran. [85]Research Institute and Museum of Anthropology, Lomonosov Moscow State University, Moscow, Russia. [86]Department of Archaeogenetics, Max Planck Institute for Evolutionary Anthropology, Leipzig, Germany. [87]Institute for Pre- and Protohistoric Archaeology and Archaeology of the Roman Provinces, Ludwig Maximilian University, Munich, Munich, Germany. [88]School of Biological Sciences, The University of Adelaide, Adelaide, South Australia, Australia. [89]Department of Biology, Universidad Autónoma de Madrid, Madrid, Spain. [90]Eurasia Department of the German Archaeological Institute, Berlin, Germany. [91]Evolutionary Adaptive Genomics, Institute of Biochemistry and Biology, Faculty of Mathematics and Science, University of Potsdam, Potsdam, Germany. [92]Department of Evolutionary Genetics, Leibniz-Institute for Zoo and Wildlife Research, Berlin, Germany. [93]Albrecht Daniel Thaer-Institute, Faculty of Life Sciences, Humboldt University Berlin, Berlin, Germany. [94]ArchaeoBioCenter and Institute of Palaeoanatomy, Domestication Research and the History of Veterinary Medicine, LMU Munich, Munich, Germany. [95]SNSB, State Collection of Anthropology and Palaeoanatomy, Munich, Germany. [96]ArchaeoZOOlogy in Siberia and Central Asia—ZooSCAn International Research Laboratory, Institute of Archeology and Ethnography of the Siberian Branch of the RAS, Novosibirsk, Russia. [97]Department of Eastern European and Siberian Archaeology, State Hermitage Museum, St Petersburg, Russia. [98]Paleoecology Laboratory, Institute of Plant and Animal Ecology, Ural Branch of the Russian Academy of Sciences, Ekaterinburg, Russia. [99]Zoological Institute, Russian Academy of Sciences, St Petersburg, Russia. [100]Department of Archaeology, History Faculty, Vilnius University, Vilnius, Lithuania. [101]Laboratory for Archaeological Research, Faculty of History and Law, Kostanay State University, Kostanay, Kazakhstan. [102]Department of History and Archaeology, Surgut Governmental University, Surgut, Russia. [103]Saryarka Archaeological Institute, Buketov Karaganda University, Karaganda, Kazakhstan. [104]Toraighyrov University, Joint Research Center for Archeological Studies, Pavlodar, Kazakhstan. [105]Faculty of History, L. N. Gumilev Eurasian National University, Nur-Sultan, Kazakhstan. [106]Institute of Archaeology and Steppe Civilizations, Al-Farabi Kazakh National University, Almaty, Kazakhstan. [107]Department of Anthropology, Alumni Building, University of North Carolina at Chapel Hill, Chapel Hill, NC, USA. [108]Nasledie Cultural Heritage Unit, Stavropol, Russia. [109]Research Center for the Preservation of Cultural Heritage, Saratov, Russia. [110]Department of Russian History and Archaeology, Samara State University of Social Sciences and Education, Samara, Russia. [111]Russian and Foreign History Department, South Ural State University, Chelyabinsk, Russia. [112]South Ural Department, Institute of History and Archaeology, Ural Branch of the Russian Academy of Sciences, Ekaterinburg, Russia. [113]Archaeological School, Chuvash State Institute of Humanities, Cheboksary, Russia. [114]Department of History of the Institute of Humanities, Ural Federal University, Ekaterinburg, Russia. [115]Department of Human Evolutionary Biology, Harvard University, Cambridge, MA, USA. [116]Anthropology Faculty, Hartwick College, Oneonta, NY, USA. [117]Department of Nordic Studies and Linguistics, University of Copenhagen, Copenhagen, Denmark. [118]Leiden University Center for Linguistics, Leiden University, Leiden, The Netherlands. [119]Department of Historical Studies, University of Gothenburg, Gothenburg, Sweden. [120]Lundbeck Foundation GeoGenetics Centre, Copenhagen, Denmark. [121]Present address: Department of Biotechnology, Abdul Wali Khan University, Mardan, Pakistan.

## Methods

### Radiocarbon dating

A total of 170 new radiocarbon dates were obtained in this study. Dating was carried out at the Keck Carbon Cycle AMS Laboratory, UC Irvine following collagen extraction and ultra-filtration from approximately 1 g of osseous material. IntCal20 calibration[48] was performed using OxCalOnline[49].

### Genome sequencing

All samples were collected with permission from the organizations holding the collections and documented through official authorization letters for partially destructive sampling from local authorities. Samples were processed for DNA extraction, library construction and shallow sequencing in the ancient DNA facilities of the Centre for Anthropobiology and Genomics of Toulouse (CAGT), France. The overall methodology followed the work from Seguin–Orlando and colleagues[50]. It involved: (1) powdering a total of 100–590 mg of osseous material using the Mixel Mill MM200 (Retsch) Micro-dismembrator; (2) extracting DNA following the procedure Y2 from Gamba and colleagues[51], tailored to facilitate the recovery of even the shortest DNA fragments; (3) treating DNA extracts with the USER (NEB) enzymatic cocktail to eliminate a fraction of post mortem DNA damage[13]; (4) constructing from double-stranded DNA templates DNA libraries in which two internal indexes are added during adapter ligation and one external index is added during PCR amplification; and (5) amplification, purification and quantification of DNA libraries before pooling 20–50 DNA libraries for low-depth sequencing on the Illumina MiniSeq instrument (paired-end mode, 2 × 80). All three indexes of each library were unique in a given sequencing pool.

Raw fastQ files were demultiplexed, trimmed and collapsed when individual read pairs showed significant overlap using AdapterRemoval2[52] (version 2.3.0), disregarding reads shorter than 25 bp. Processed reads were then aligned against the nuclear and mitochondrial horse reference genomes[53,54], and appended with the Y-chromosome contigs from[55] using the Paleomix bam_pipeline (version 1.2.13.2) with the mapping parameters recommended by Poullet and Orlando[56]. Sequencing reads representing PCR duplicates or showing a mapping quality below 25 were disregarded. DNA fragmentation and nucleotide misincorporation patterns were assessed on the basis of 100,000 random mapped reads using mapDamage2[57] (version 2.0.8). Paleomix returned provisional estimates of endogenous DNA content and clonality, as defined by the fraction of retained reads mapping uniquely against the horse reference genomes and those mapping at the same genomic coordinates, respectively. These numbers guided further experimental decisions, including (1) the sequencing effort to be performed per individual library; (2) the preparation of additional libraries from left-over aliquots of USER-treated DNA extracts, or following treatment of DNA extract aliquots with the USER enzymatic cocktail; and (3) the preparation of additional DNA extracts. After initial screening for library content, sequencing was carried out on the Illumina HiSeq4000 instruments from Genoscope (paired-end mode, 2 × 76; France Génomique), except for four samples (BPTDG1_Fra_m11800, Closeau3_Fra_m10400, Novoil1_Kaz_m1832 and Novoil2_Kaz_m1832), for which sequencing was done at Novogene Europe on an Illumina NovaSeq 6000 instrument (S4 lanes, paired-end mode, 2 × 150). Overall, we obtained sequence data for a total of 264 novel ancient horse specimens and 1,029 DNA libraries (980 new), summing up to 31.86 billion sequencing read pairs and 100.82 billion collapsed read pairs, which was sufficient to characterize 226 novel ancient genomes showing a genomic depth-of-coverage of at least 1× (median 2.80-fold, maximum 25.76-fold) (Supplementary Table 1).

### Allele sampling, sequencing error rates, genome rescaling and trimming

Following previous work[5,58], error rates are defined as the excess of mutations that are private to the ancient genome, relative to a modern genome considered as error-free. Mutations were polarized using an outgroup genome representing a consensus built from seven male specimens of diverse equine species (*Equus africanus somaliensis*, *Equus asinus*, *Equus burchelli*, *Equus grevyi*, *Equus hartmannae*, *Equus hemionus onager* and *Equus kiang*[59]), according to a majority rule in which at maximum 2 of the 7 individuals showed an alternative allele. Minor and major alleles were identified using ANGSD[60] (version 0.933-86-g3fefdc4, htslib: 1.10.2-106-g9c35744) and the following parameters: -baq 0 -doMajorMinor 2 -uniqueOnly 1 -minMapQ 25 -minQ 30 -minInd 7 -doCounts 1 -doMaf 1.

Error rate estimates ranged between 0.000337 and 0.003966 errors per site and revealed that nucleotide C→T and G→A misincorporation rates were still inflated relative to their reciprocal substitution types (T→C and A→G), despite USER treatment. Therefore, individual BAM alignment files were processed to further reduce nucleotide misincorporation rates. To achieve this, we used PMDtools[61] (version 0.60) to bin apart reads likely containing post mortem DNA damage (--threshold 1; DAM) from those that did not (--upperthreshold 1; NODAM). NODAM-aligned reads were then directly trimmed by 5 bp at their ends, where individual base qualities generally drop. The base quality of aligned DAM reads was first rescaled using mapDamage2[57] (version 2.0.8), penalizing all instances of potential derivatives of post mortem cytosine deamination, then further trimmed by 10 bp at both ends. The resulting NODAM and DAM aligned reads were merged again to obtain final BAM sequence alignments. Final error rate estimates ranged between 0.000080 and 0.000933 errors per site (Supplementary Table 1).

### Uniparentally inherited markers and coat colouration

Mitochondrial genomes for the 264 newly sequenced samples were characterized from quality-filtered BAM alignment files (minMapQ=25, minQ=30), using a majority rule requiring at least five individual reads per position. Their resulting complete mitochondrial genome sequences were aligned together with a total of 193 sequences previously characterized[3,5,14,15,58,62,63] using mafft[64] (version 7.407). Sequence alignments were split into six partitions, following previous work[5], including the control region, all tRNAs, both rRNAs and each codon position considered separately. Maximum-likelihood phylogenetic reconstruction was performed using RAxML[65] (version 8.2.11) with default parameters, and assessing node support from a total of 100 bootstrap pseudo-replicates. The same partitions were provided as input for BEAST[66] (version 2.5.1), together with calibrated radiocarbon years (Supplementary Table 1). Specimens lacking direct radiocarbon dates or identified as not belonging to the DOM2 cluster were disregarded (Supplementary Table 1). While the former ensured precise tip-calibration for molecular clock estimation (assuming uncorrelated log-normal relaxed model), the latter prevented misinterpreting spatial variation in the population structure as changes in the effective population size[67]. The best substitution model was selected from ModelGenerator[68] (version 0.85) and Bayesian Skyline plots[69] were retrieved following 1,000,000,000 generations, sampling 1 every 1,000 and disregarding the first 30% as burn-in. Convergence was visually checked in Tracer[70] (version 1.7.2).

The Y-chromosome maximum-likelihood tree was constructed calling individual haplotypes from trimmed and rescaled BAM sequence alignments against the contigs described by Felkel and colleagues[55], filtered for single copy MSY regions. The final multifasta sequence alignment included sites covered in at least 20% of the specimens, pseudo-haploidizing each position and filtering out transitions, as done with autosomal data. It was further restricted to specimens showing at least 20% of the final set of positions covered. This represented a total of 3,195 nucleotide transversions for 142 specimens. The final tree was computed using IQtree (version 1.6.12), following AICc selection of the best substitution model and 1,000 ultrafast bootstrap approximation for assessing node support[71,72]. The Y-chromosome Bayesian

skyline plot was obtained following the same procedure as above. Maximum-likelihood trees and Bayesian skyline plots are shown in Supplementary Fig 1 and Extended Data Fig. 3e, f, respectively.

The presence of alleles associated with or causative for a diversity of coat colouration changes was investigated using individual BAM read alignments. For a total of 43 genomic locations representing biallelic SNPs, we simply counted the proportion of reads supporting the associated or causative allele. Results were summarized in the heat map shown in Extended Data Fig. 8, with respect to the sample ordering displayed in the neighbour-joining phylogenetic reconstruction, and limited to those 13 loci that were polymorphic in our horse panel for clarity.

## Neighbour-joining phylogeny, genetic continuity and population modelling

Phylogenetic affinities were first estimated by performing a BioNJ tree reconstruction with FastME[73] (version 2.1.4), based on the pairwise matrix of genetic distances inferred from the bed2diffs_v1 program[16]. Node supports were assessed using a total of 100 bootstrap pseudo-replicates. The 'goodness-of-fit' of the neighbour-joining tree to the data was evaluated by comparing the patristic distances and raw pairwise distances. Patristic distances were obtained from the ape[74] R package (version 5.5) and their ratios to raw pairwise distances were averaged for each given individual (Fig 1c). Averaged ratios equal to one support perfect phylogenetic placement for the specimen considered.

Genetic continuity between each individual specimen predating about 2200 BC and DOM2 horses was tested following the methodology from Schraiber[75], which implements a likelihood-ratio test to compare the statistical support for placing DOM2 and the ancient specimen in a direct line of ancestry or as two sister groups. This methodology relies on exact allele frequency estimates within DOM2 and read counts for putatively ancestral ancient samples. To exclude residual sequencing errors within DOM2 horses, we, thus, conditioned these analyses on variants segregating at least as doubletons in positions covered in at least 75% of the DOM2 samples. Linked variation was pruned using Plink[76] (version v.1.9), with the following parameters, --indep-pairwise 50 10 0.2, which provided a panel of about 1.4 million transversions. Allele frequencies were polarized considering the outgroup genome used for measuring error rates. Results from direct ancestry tests are summarized in Supplementary Table 2.

The complex genetic makeup of some individuals (CAR05_Hun_m2458 and Kan22_Tur_m2386) and/or group of individuals (DOM2) was investigated using the $f_4$-statistics-based ancestry decomposition approach implemented in qpAdm[17] (version 7.0), in which one particular (group of) individual(s) is modelled as a linear, additive combination of candidate population sources ('left' populations). We followed the rotating strategy recommended by Harney and colleagues[18] to assess all possible combinations of two, three and four donors ('left') selected from a total of 18 populations. The remaining 14, 15 and 16 populations were used as reference ('right') populations (Supplementary Table 3).

We selected a total of nine horse lineages representing the main phylogenetic clusters, and carrying genetic ancestry profiles representative of the complete dataset, to model the population evolutionary history using OrientAGraph[19] (version 1.0). By implementing a network orientation subroutine that enables throughout exploration of the graph space, OrientAGraph constitutes a marked advancement in the automated inference of admixture graphs. We considered scenarios from zero to five migration pulses ($M = 0$ to $5$; Extended Data Fig. 5a–e), and the population model assuming $M = 3$ is represented in Fig 3b. This analysis was conditioned on sites covered at least in one specimen of each population group. This filter yielded a set of 7,936,493 fully orthologous nucleotide transversions.

## Struct-f4, ancestry components and multi-dimensional scaling

We extended the Struct-f4 package so as to assess individual genetic affinities within a panel of genomes, and to decompose them into $K$ genetic ancestries. Struct-f4, thus, achieves similar objectives to other clustering methods, such as ADMIXTURE[77] and Ohana[78], but does not assume Hardy–Weinberg equilibrium. The latter assumption is known to cause misinterpretation of highly drifted samples as ancestral homogeneous groups instead of highly derived mixtures from multiple populations, as thoroughly described elsewhere[79]. To circumvent this, Struct-f4 relies on the calculation of the widely used $f_4$ statistics, which were originally devised not only to test for admixture, but also to quantify the drift between the internal nodes of a population tree. The latter provides a direct representation of the true ancestral populations. Overall, Struct-f4 thus implements a more natural and robust (model-free) approach than other clustering alternatives.

Struct-f4 is based on a mixture model that parametrizes the drift that occurred between a given number of $K$ pre-defined ancestral populations, and the mixing coefficient of each individual. Model parameters are estimated using an adaptive Metropolis–Hastings Markov chain Monte Carlo integration, identifying optimal numerical solutions for parameters by means of likelihood maximization. Struct-f4 was validated following extensive coalescent simulations with fastsimcoal2[80] (version 2.6.0.3). An example of such simulation designed to mimic the complex horse evolutionary history is provided in Extended Data Fig. 2, based on mutation and recombination rates of $2.3 \times 10^{-8}$ and $10^{-8}$ events per generation and bp, respectively. Struct-f4 is implemented in Rcpp and only takes the full set of $f_4$-statistics as input to automatically return individual ancestry coefficients, without requiring pre-defined, ad-hoc sets of reference and test populations.

Multi-dimensional scaling was carried out based on the co-ancestry semi-matrix summarizing the drift measured between each pair of individuals, as returned by Struct-f4, removing the domestic donkey outgroup prior to using the cmdscale R function.

## Isolation by distance and spatial connectivity

Spatial barriers to gene flow prior to about 3000 BC, between about 3000 and 2000 BC and following about 2000 BC were run using EEMS[16] (built with Eigen version 3.2.2 and Boost version 1.57, and using rEEMSplots version 0.0.0.9000) for 50 million iterations and considering a burn-in of 15 million iterations. Convergence was ensured from visual inspection of likelihood trajectories as well as by the strong correlation obtained between the observed and fitted genetic dissimilarities. Pie-charts depicting the ancestry proportions inferred by Struct-f4 were overlaid on the migration surfaces to facilitate tracking the geographic position of each excavation site, averaging ancestry proportions or using individual ancestry profiles if only one sample was characterized genetically at that location. Spatial pie-chart projection was carried out using the draw.pie R function from the mapplots package[81] (version 1.5.1). The size of each individual pie-chart was commensurate with the number of samples excavated at a given geographic location, provided that the number of samples was lower than 10, while set to a constant maximum radius otherwise.

Partial Mantel tests measuring the correlation between geographic and genomic distances over time were carried out using the ncf R package[82] (version 1.2.9). This test corrected for the time variation present within each window, similar to the approach described by Loog and colleagues[83]. Haversine geographic distances between pairs of ancient samples were computed using the geosphere package (version 1.5.10) in R[84], from the corresponding longitude and latitude coordinates, while radiocarbon date ages were considered as point estimates (Supplementary Table 1). The matrix of pairwise genetic distances was obtained from the bed2diffs_v1 program provided together with the EEMS software[16]. The analysis was carried out for autosomes and the X chromosome separately, so as to investigate possible sex-bias in horse dispersal. Confidence intervals were calculated by sampling with replacement individuals within each time window.

Sliding time windows (step size = 250 years) were broadened forward in time until including at least ten specimens covering two-thirds of the

total geographic area sampled in this study. The area delimited by a set or subset of GPS coordinates was calculated using the GeoRange R package[85] (version 0.1.0) and the age of the window was set to the average age amongst the samples included. Additionally, pairwise distances involving samples located less than 500 km away and separated by less than 500 years were masked in the corresponding matrices to estimate the patterns of isolation by distance between demes, instead of within demes. This whole scheme was designed to prevent regional effects, caused by the over-representation of particular regions in specific time intervals.

The LOCATOR[20] program (version 1.2) was run using a geolocated reference panel consisting of all non-DOM2 horses ($n = 136$), except the tarpan and the four Przewalski's horses present in our dataset, and considering nucleotide transversions covered at least in 75% of the samples, for a total of 3,194,008 SNPs. The geographic origin of each DOM2 horse was then estimated from the geographic structure defined by the populations present in the reference panel. Default parameters were used, except that the width of each neural layer was 512 (instead of 256). The best run was selected as the one showing the lowest validation error from a total of 50 independent runs. The analysis was repeated for the tarpan as well as the four Przewalski's horses present in our dataset.

## Selection scans

To pinpoint genetic changes potentially underlying biological adaptation within DOM2 horses, we contrasted the frequency of each nucleotide transversion in our dataset ($n = 10,205,277$) in DOM2 ($n = 141$) and non-DOM2 horses ($n = 142$). The extensive number of samples represented provided unprecedented resolution into patterns of allele frequency differentiation, and encompassed the largest diversity of non-DOM2 horses characterized to date. Weir and Cockerham $F_{ST}$ index values between both groups were calculated using Plink[76] (version 1.9) and visualized using the GViz R package[86] (version 1.36.2), together with external genomic tracks provided by the gene models annotated for EquCab3 (Ensembl v0.102) and the interrupted repeats precomputed for the same assembly and stored in the UCSC browser.

## Reporting summary

Further information on research design is available in the Nature Research Reporting Summary linked to this paper.

## Data availability

All collapsed and paired-end sequence data for samples sequenced in this study are available in compressed fastq format through the European Nucleotide Archive under accession number PRJEB44430, together with rescaled and trimmed bam sequence alignments against both the nuclear and mitochondrial horse reference genomes. Previously published ancient data used in this study are available under accession numbers PRJEB7537, PRJEB10098, PRJEB10854, PRJEB22390 and PRJEB31613, and detailed in Supplementary Table 1. The genomes of ten modern horses, publicly available, were also accessed as indicated in their corresponding original publications[59,63,87–89].

## Code availability

The Struct-f4 software is available without restriction on Bitbucket (https://bitbucket.org/plibradosanz/structf4/src/master/).

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

**Acknowledgements** We thank all members of the AGES group at CAGT. We are grateful for the Museum of the Institute of Plant and Animal Ecology (UB RAS, Ekaterinburg) for providing specimens. The work by G. Boeskorov is done on state assignment of DPMGI SB RAS. This project was supported by the University Paul Sabatier IDEX Chaire d'Excellence (OURASI); Villum Funden miGENEPI research programme; the CNRS 'Programme de Recherche Conjoint' (PRC); the CNRS International Research Project (IRP AMADEUS); the France Génomique Appel à Grand Projet (ANR-10-INBS-09-08, BUCEPHALE project); IB10131 and IB18060, both funded by Junta de Extremadura (Spain) and European Regional Development Fund; Czech Academy of Sciences (RVO:67985912); the Zoological Institute ZIN RAS (AAAA-A19-119032590102-7); and King Saud University Researchers Supporting Project (NSRSP–2020/2). The research was carried out with the financial support of the Russian Foundation for Basic Research (19-59-15001 and 20-04-00213), the Russian Science Foundation (16-18-10265, 20-78-10151, and 21-18-00457), the Government of the Russian Federation (FENU-2020-0021), the Estonian Research Council (PRG29), the Estonian Ministry of Education and Research (PRG1209), the Hungarian Scientific Research Fund (Project NF 104792), the Hungarian Academy of Sciences (Momentum Mobility Research Project of the Institute of Archaeology, Research Centre for the Humanities); and the Polish National Science Centre (2013/11/B/HS3/03822). This project has received funding from the European Union's Horizon 2020 research and innovation programme under the Marie Skłodowska-Curie (grant agreement 797449). This project has received funding from the European Research Council (ERC) under the European Union's Horizon 2020 research and innovation programme (grant agreements 681605, 716732 and 834616).

**Author contributions** Designed, conceived and coordinated the study: L.O. Provided samples, reagents and material: A. Perdereau, J.-M.A., B.S., A.A.T., A.A.K., S.A., A.H.A., K.A.S.A.-R., T. Seregély, L.K., R.I., O.B.-L., P.B., M.O., J.-C.C., M.B.-M., N.A., M.G., M.M.-d.H., J.W., S.P., A.L.-K., K. Tunia, M.N., E.R., U.S., G. Boeskorov, L.L., R.K., L.P., A. Bălăşescu, V. Dumitraşcu, R.D., D.G., V.K., A.S.-N., B.G.M., Z.G., K.S., G.K., E.G., R.B., M.E.A., G.S., V. Dergachev, H.S., N.T., S.G., A. Kasparov, A.E.B., M.A.A., P.A.N., E.Y.P., V.P., G. Brem, B.W., C.S., M.K., K. Kitagawa, A.N.B., A. Bessudnov, W.T., J.M., J.-O.G., J.B., D.E., K. Tabaldiev, E.M., B.B., T.T., M.P., S.O., C.A.M., S.V.L., S.A.C., A.N.E., M.P.I., J.L.G., E.R.G., S.C., C.O., J.L.A., N. Kotova, A. Pryor, P.C., R.Z., A.T., N.L.M., T.K., D.L., M. Marzullo, O.P., G.B.G., U.T., B.C., S.L., H.D., M. Mashkour, N.Y.B., P.W.S., J.K., W.H., A.M.-M., N.B., M.H., A. Ludwig, A.S.G., J.P., K.Y.K., T.-O.I., N.A.B., S.K.V., N.N.S., K.V.C., N.A.P., G.F.B., E.P., M.S., E.A., A. Logvin, I.S., V. Logvin, S.K., V. Loman, I.K., I.M., V.M., S. Sakenov, V.V., E.U., V.Z., B.A., A.B.B., A. Kalmykov, S.R., S.H., A.I.Y., A.A.V., A.E., N.S.B., N.R., P.A.K., P.F.K., D. Anthony, G.J.K., K. Kristiansen, P.W., A.O. and L.O. Performed radiocarbon dating: J.S. Performed wet-lab work: N. Kahn, A. Fages, M.A.K., T. Suchan, L.T.-C., S. Schiavinato, A.F., A. Perdereau, C.G., L.C., A.S.-O and C.D.S., with input from L.O. Analysed genomic data: P.L. and L.O. Analysed uniparental markers: D. Alioglu, with input from P.L. and L.O. Prepared the linguistic index: G.J.K. Interpreted data: P.L. and L.O., with input from B.A., S.R., S.H., D. Anthony, G.J.K., K. Kristiansen and A.O. Wrote the article: L.O., with input from P.L., B.A., S.R., S.H., D. Anthony, G.J.K., K. Kristiansen, A.O. and all co-authors. Wrote the Supplementary Information: P.L., A. Fages, G.J.K. and L.O., with input from all co-authors.

**Competing interests** The authors declare no competing interests.

**Additional information**
**Correspondence and requests for materials** should be addressed to Ludovic Orlando.

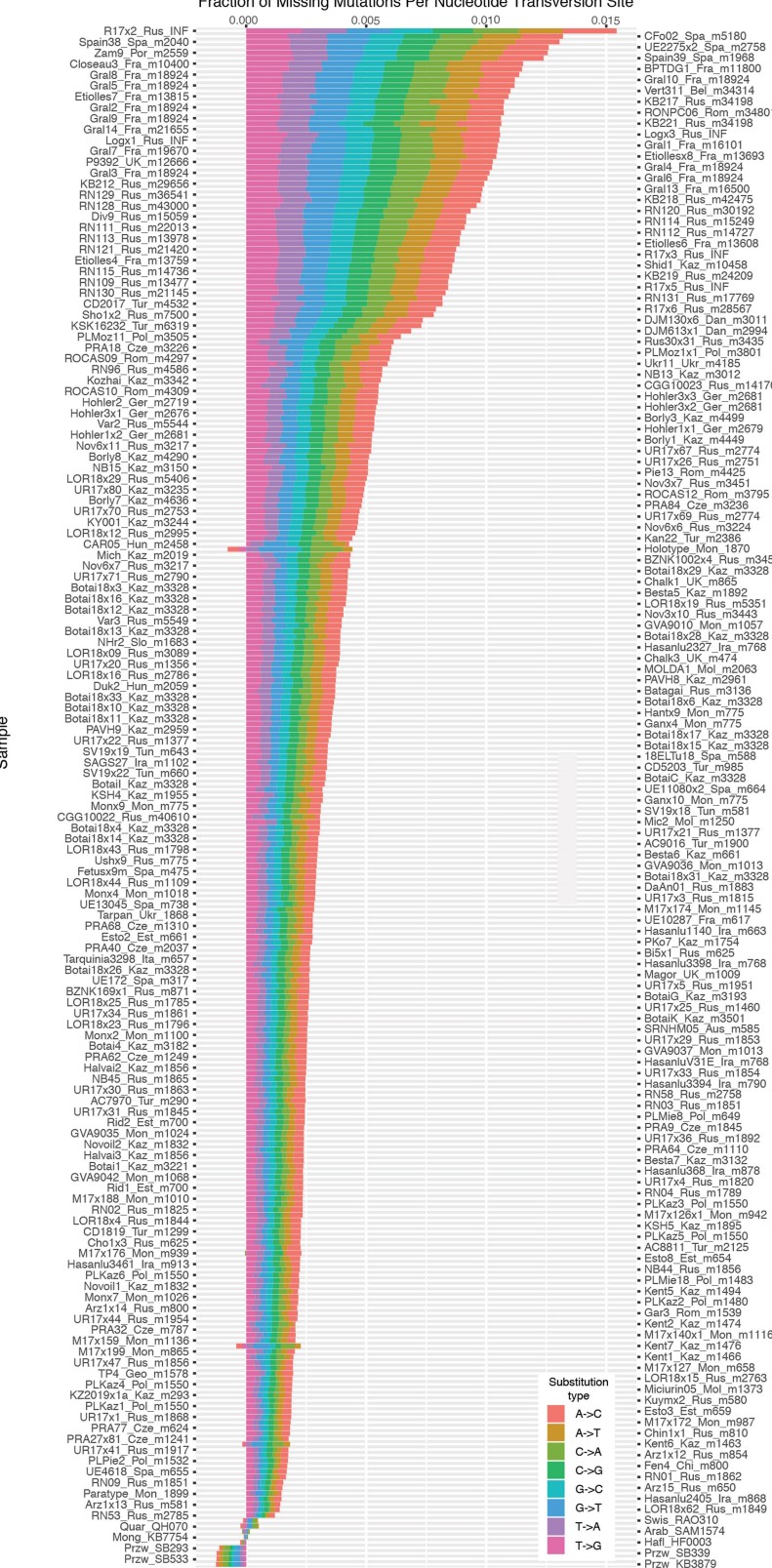

**Extended Data Fig. 1 | Proportion of missing derived mutations at sites representing nucleotide transversions.** Proportions are provided relative to the genome of a modern Icelandic[89] (P5782) horse (Spearman correlation coefficient between total transversion errors and time, R=−0.77 p-value =0).

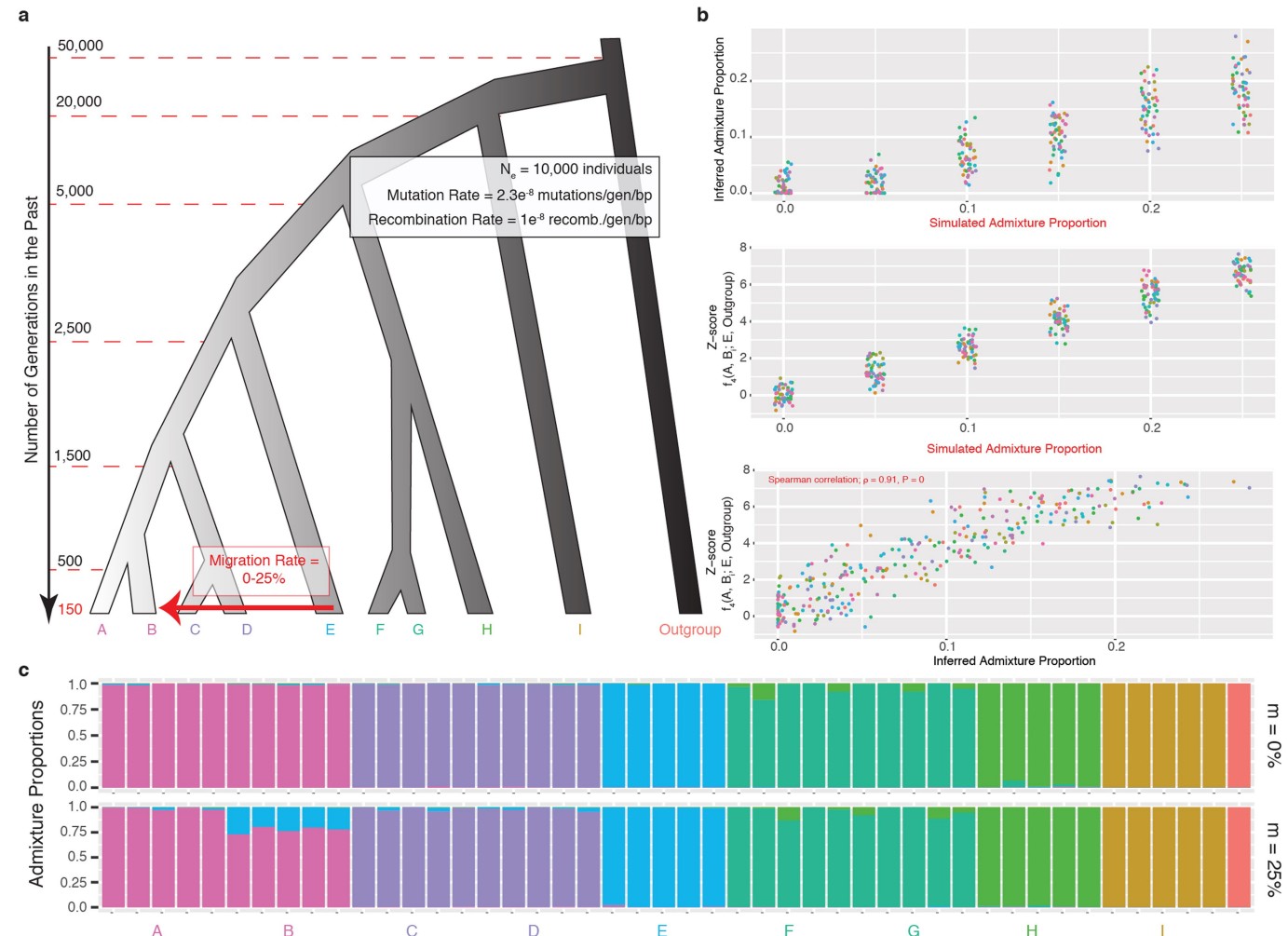

**Extended Data Fig. 2 | Struct-f4 validation. a**, Simulated demographic model. A single migration pulse is assumed to have occurred 150 generations ago from population E into B. The magnitude of the migration represents 5% to 25% of the effective size of population B. The model was also simulated in the absence of migration (i.e. m=0%). Five individuals are simulated per population considered, except for the outgroup where only one individual was considered. **b**, Correlation of the expected levels of gene-flow with the predicted E-ancestry component in individuals $i$ belonging to population B, as well as with the average Z-scores of the $f_4(A, B_i; E, Outgroup)$ configurations, which reflects the stochasticity resulting from the simulations, prior to any inference. Each point represents a simulated individual. Colors indicate the 10 independent simulation replicates carried out. **c**, Predicted ancestry profiles in the absence (m=0%) and with gene flow (m=25% and K=7, as per the number of internal nodes immediately ancestral to the 10 extant populations).

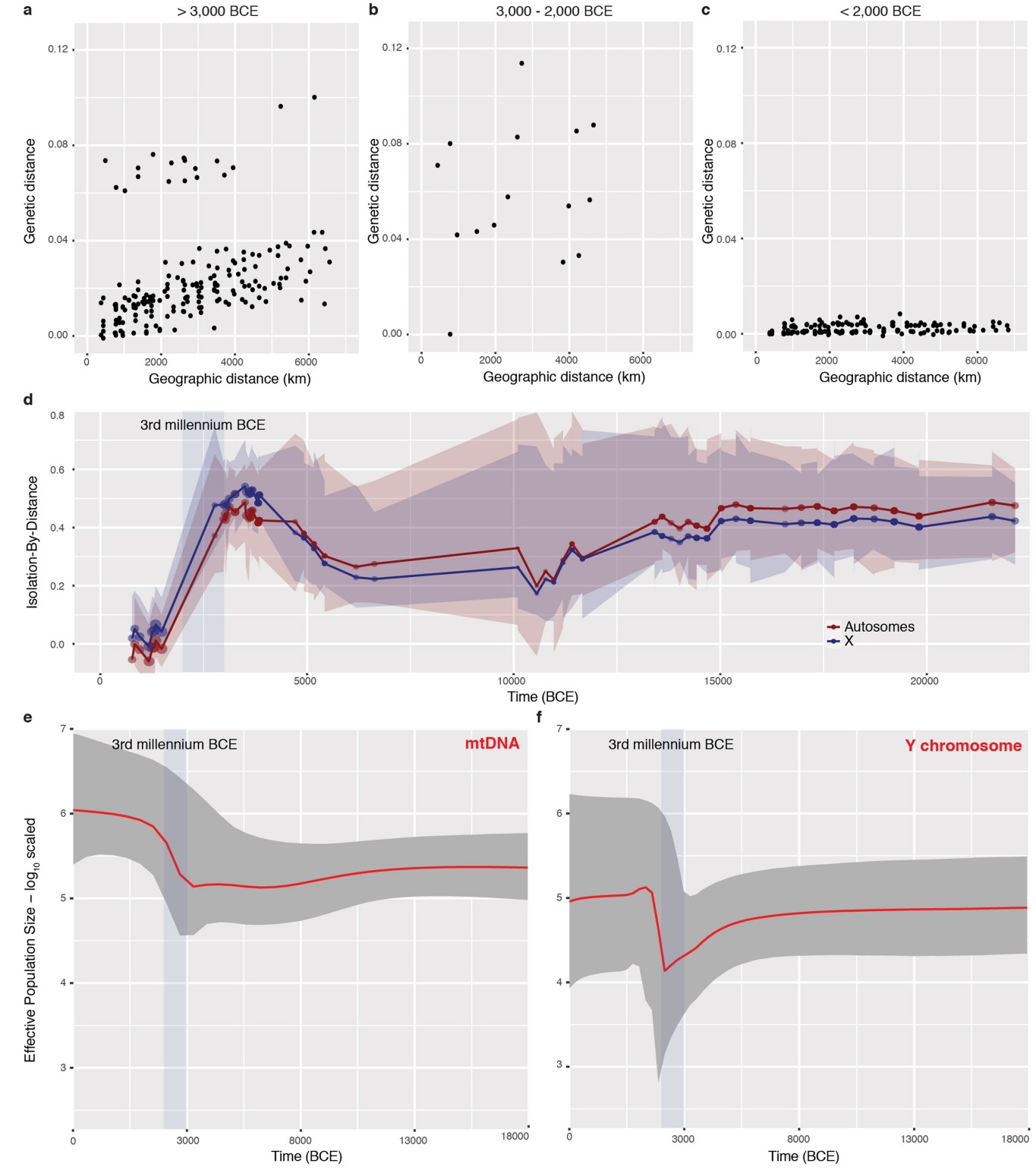

**Extended Data Fig. 3 | Mobility and demographic shifts. a–c**, Correlation between observed pairwise genetic distances between demes as inferred by EEMS[16] and Haversine geographic distances prior to -3,000 BCE (**a**), during the third millennium BCE (**b**) and after -2,000 BCE (**c**). **d**, Isolation-by-distance patterns through time inferred from autosomal (red) and X-chromosomal (blue) variation. **e**–**f**, Bayesian Skyline plots reconstructed from mtDNA (**e**) and Y-chromosomal variation (**f**). The third millennium BCE is highlighted in blue. The red line indicates the median of the 95% confidence range, shown in grey.

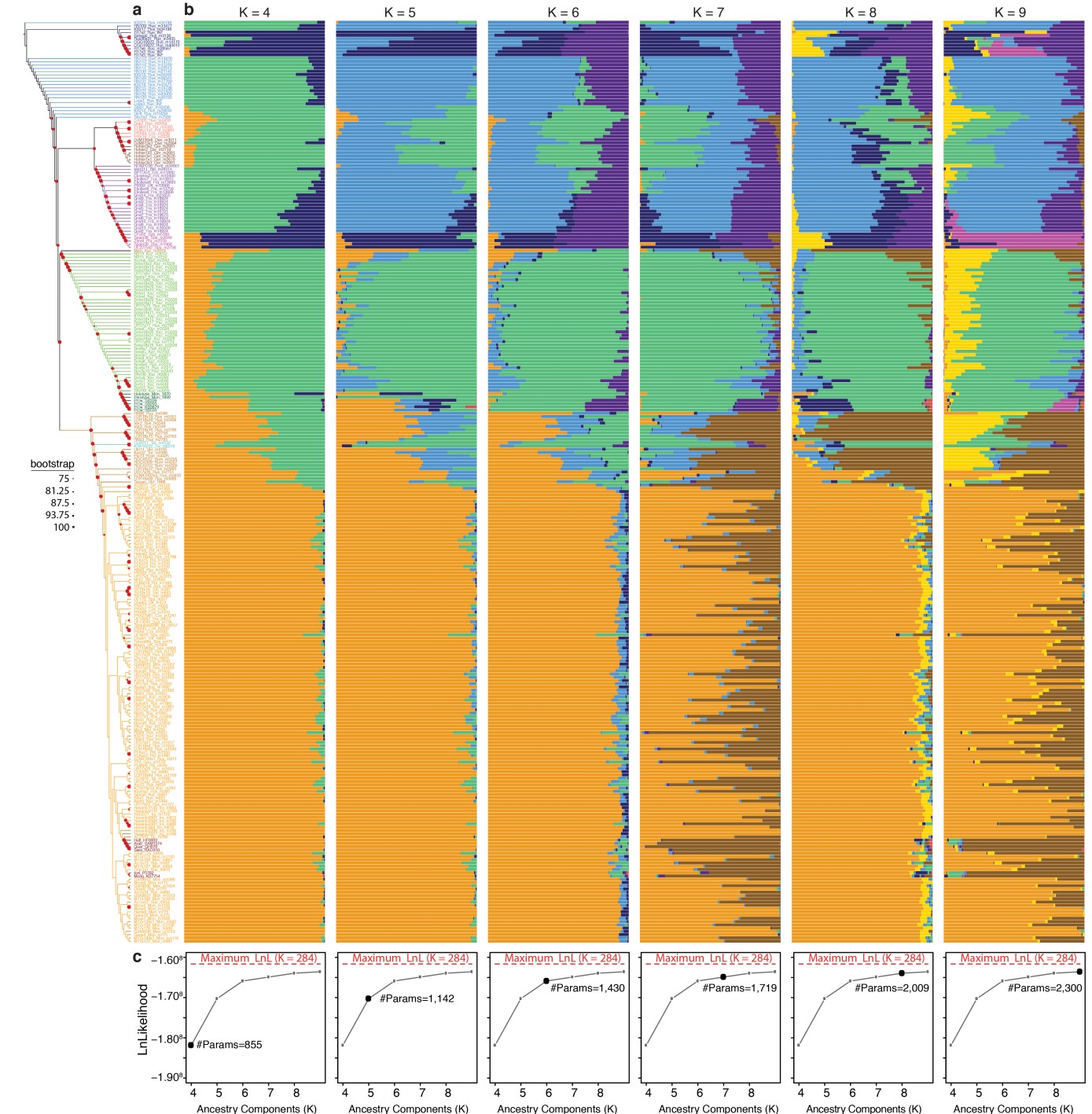

**Extended Data Fig. 4 | Individual ancestry profiles. a**, NJ-tree shown in Fig 1b with sample labels as defined in Supplementary Table 1. **b**, Struct-f4 individual ancestry profiles. **c**, Model likelihood. A total of K=4 to K=9 ancestral populations are assumed. LnL = natural log-likelihood.

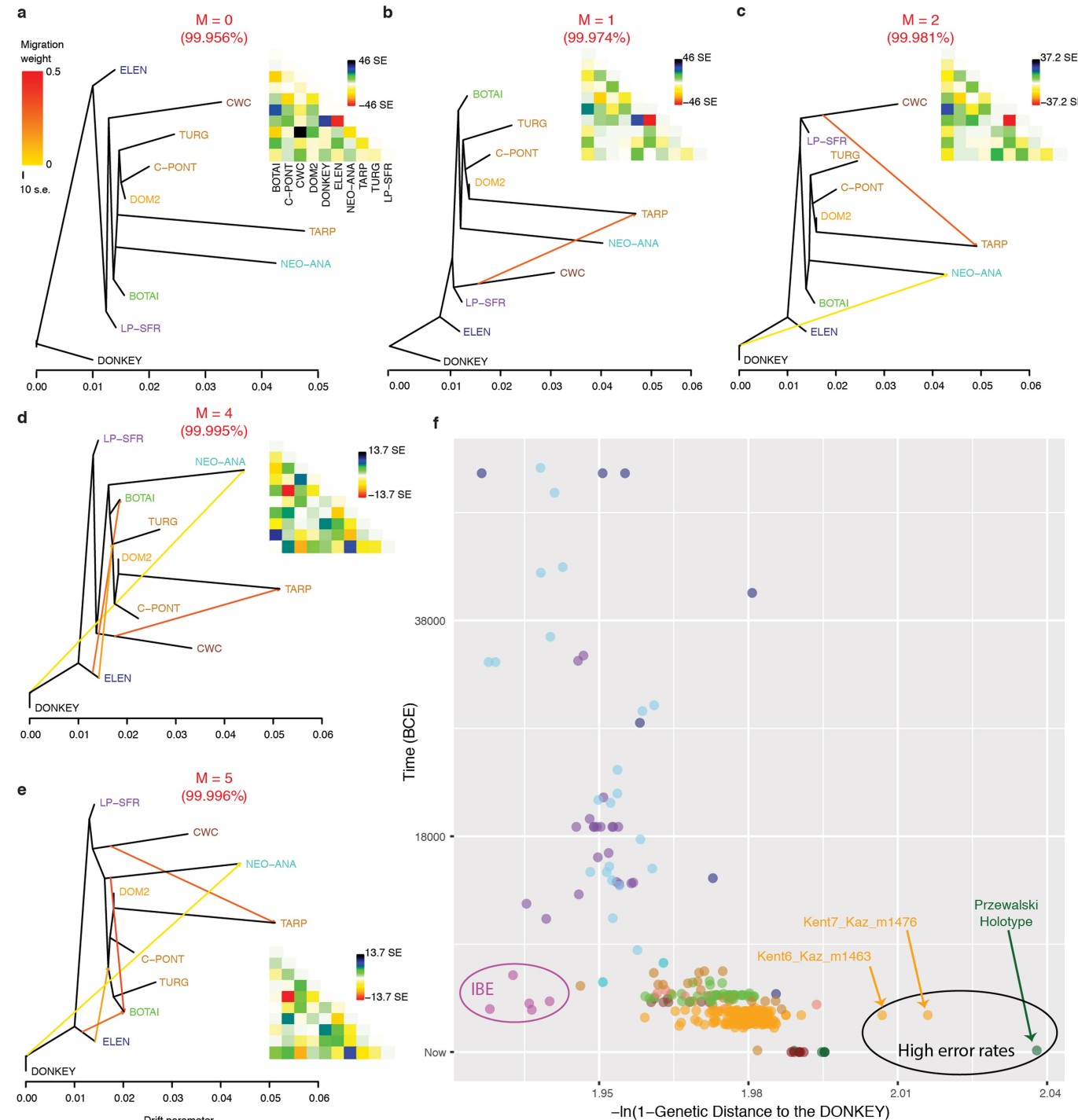

**Extended Data Fig. 5 | OrientAGraph[19] population histories and genetic distances to the domestic donkey. a–e,** OrientAGraph[19] models and residuals assuming M=0 to M=5 migration edges and considering nine lineages representing key genomic ancestries (colored as in Fig 1a). M=3 is shown in Fig 3b. **f,** Pairwise genetic distances between a given horse and the domestic donkey plotted as a function of the age of the horse specimen considered.

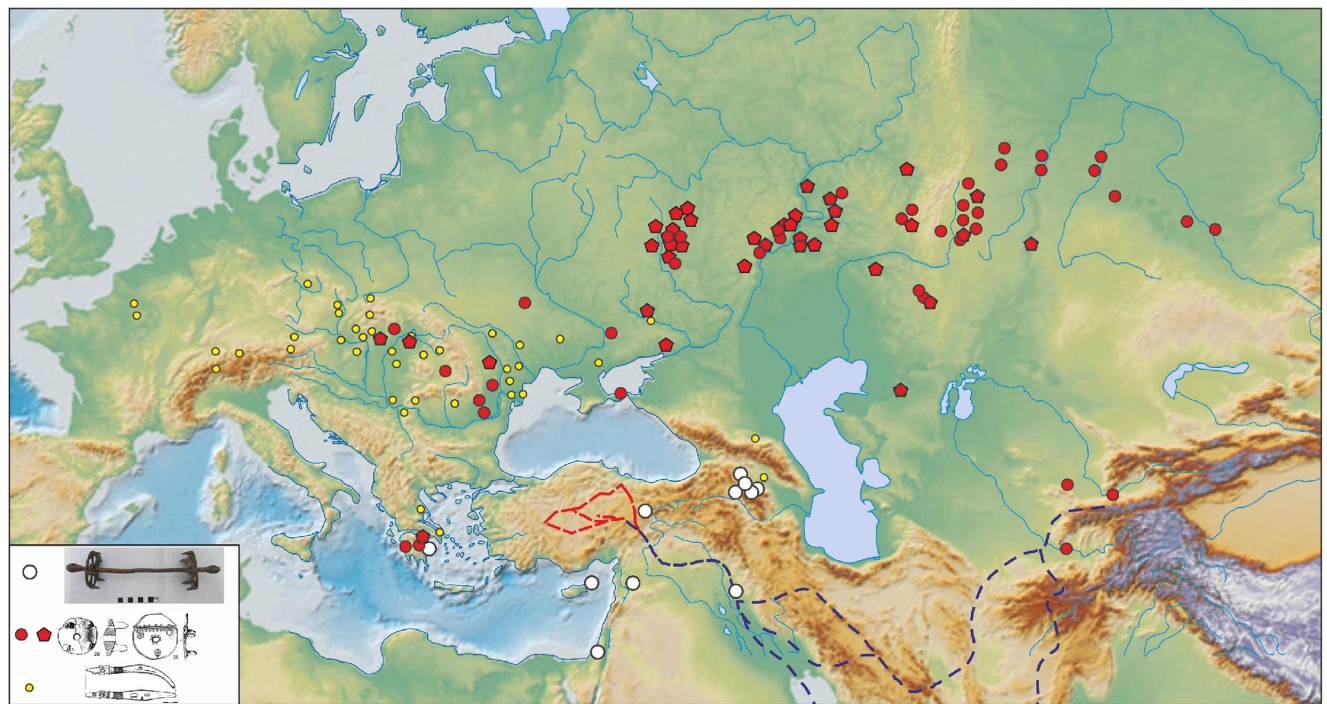

**Extended Data Fig. 6 | Inter-regional trade and chariot networks, marked by horse cheek pieces, connecting Bronze Age steppe societies, mineral rich Caucasian societies and the Old Assyrian trade network during the** **period 1,950-1,750 BCE.** Documented Near Eastern trade routes are marked with stippled lines (after[23], supplemented with data from[90,91] and Pavel F. Kuznetsov).

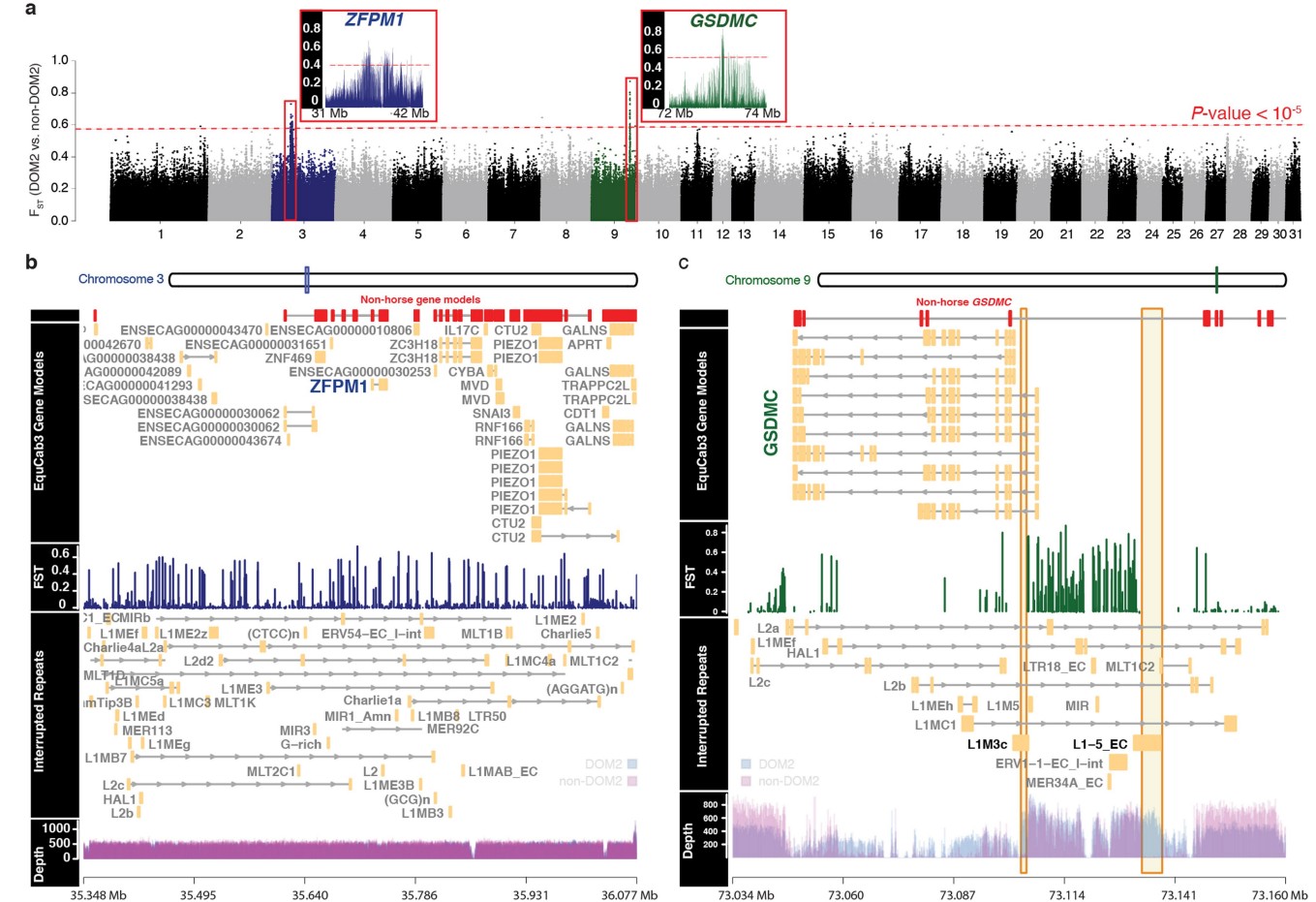

**Extended Data Fig. 7 | DOM2 selection signatures. a**, Manhattan plot of $F_{ST}$-differentiation index between DOM2 and non-DOM2 horses along the 31 EquCab3 autosomes. $F_{ST}$ outliers are highlighted using an empirical *P*-value threshold of $10^{-5}$ (red dashed line). The two outlier regions on chromosomes 3 and 9 are highlighted within red frames. **b**, $F_{ST}$-differentiation index and genomic tracks around the *ZFPM1* gene. Depth represents the accumulated number of reads per position within DOM2 (blue) and non-DOM2 (magenta) genomes. **c**, Same as Panel b at *GSDMC*.

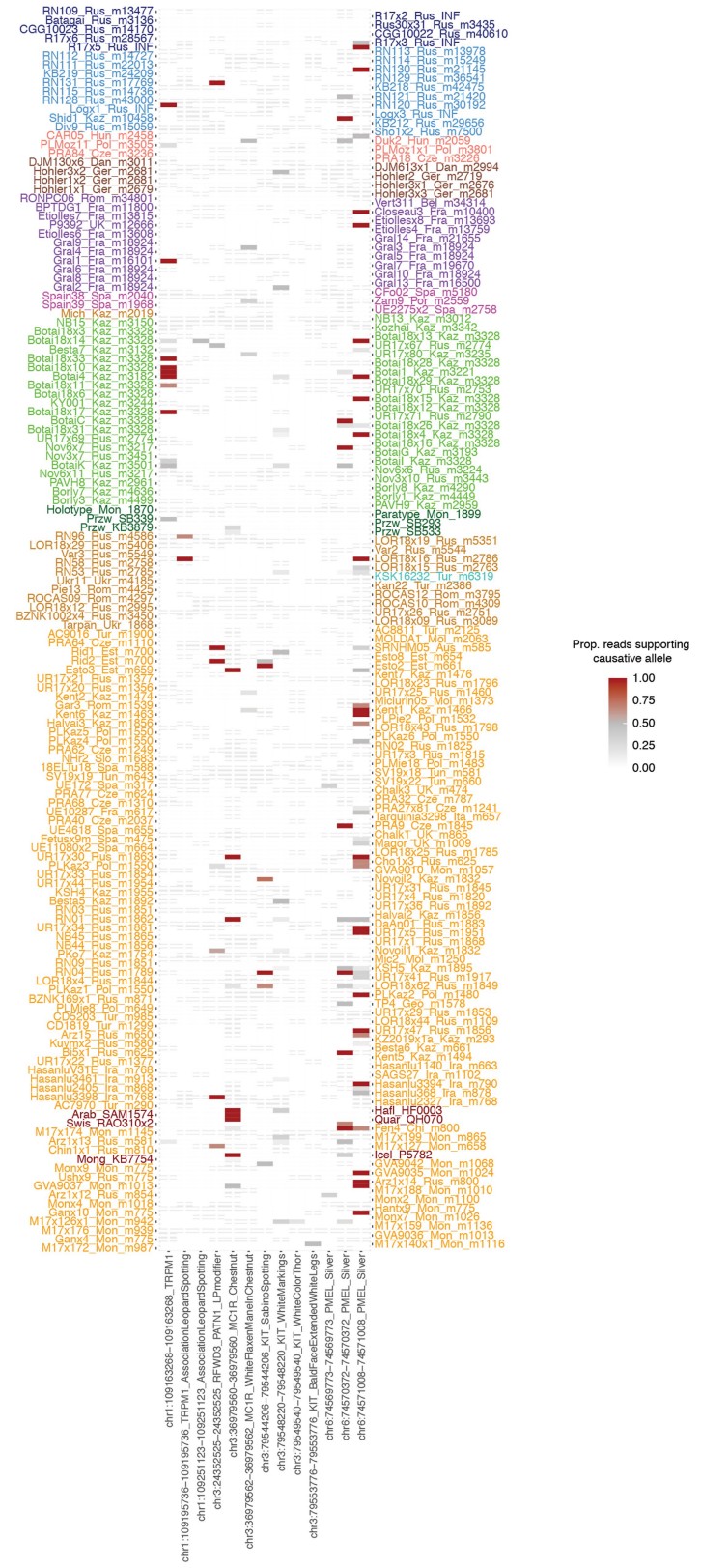

**Extended Data Fig. 8 | Normalized read coverage supporting the presence of causative alleles for coat coloration variation.** Each column represents a particular genome position where genetic polymorphisms associated or causative for coat coloration patterns have been described. The exact EquCab3 genome coordinates are indicated in the locus label. Specimens (rows) are ordered according to their phylogenetic relationships, as shown in Fig 1b. The color gradient is proportional to the fraction of reads carrying the causative variant. Loci that are not covered following trimming and rescaling of individual BAM sequence alignment files are indicated with a white cross.

# Reporting Summary

## Statistics

For all statistical analyses, confirm that the following items are present in the figure legend, table legend, main text, or Methods section.

| n/a | Confirmed | |
|---|---|---|
| ☐ | ☒ | The exact sample size (*n*) for each experimental group/condition, given as a discrete number and unit of measurement |
| ☐ | ☒ | A statement on whether measurements were taken from distinct samples or whether the same sample was measured repeatedly |
| ☒ | ☐ | The statistical test(s) used AND whether they are one- or two-sided *Only common tests should be described solely by name; describe more complex techniques in the Methods section.* |
| ☐ | ☒ | A description of all covariates tested |
| ☐ | ☒ | A description of any assumptions or corrections, such as tests of normality and adjustment for multiple comparisons |
| ☐ | ☒ | A full description of the statistical parameters including central tendency (e.g. means) or other basic estimates (e.g. regression coefficient) AND variation (e.g. standard deviation) or associated estimates of uncertainty (e.g. confidence intervals) |
| ☒ | ☐ | For null hypothesis testing, the test statistic (e.g. *F*, *t*, *r*) with confidence intervals, effect sizes, degrees of freedom and *P* value noted *Give P values as exact values whenever suitable.* |
| ☐ | ☒ | For Bayesian analysis, information on the choice of priors and Markov chain Monte Carlo settings |
| ☐ | ☒ | For hierarchical and complex designs, identification of the appropriate level for tests and full reporting of outcomes |
| ☐ | ☒ | Estimates of effect sizes (e.g. Cohen's *d*, Pearson's *r*), indicating how they were calculated |

*Our web collection on statistics for biologists contains articles on many of the points above.*

## Software and code

Policy information about availability of computer code

| Data collection | All metadata pertaining to the experimental work underlying ancient DNA characterization at CAGT is managed through the open-source CASCADE laboratory information management system (see Dolle et al. Frontiers Ecol Evol 2020). |
|---|---|
| Data analysis | The Struct-f4 software is available without restriction on Bitbucket at https://bitbucket.org/plibradosanz/structf4/src/master/, together with a companion manual providing installation and running instructions. All other analyses relied on available software, fully referenced in the manuscript, including:<br>-AdapterRemoval2 (version 2.3.0),<br>-Paleomix (version 1.2.13.2),<br>-mapDamage2 (version 2.0.8),<br>-ANGSD (version 0.933-86-g3fefdc4, htslib: 1.10.2-106-g9c35744) ,<br>-PMDtools (version 0.60),<br>-mafft (version v7.407),<br>-RAxML (version 8.2.11),<br>-BEAST (version 2.5.1),<br>-ModelGenerator (version 0.85),<br>-Tracer (version 1.7.2),<br>-IQtree (version 1.6.12),<br>-FastME (version 2.1.4),<br>-bed2diffs_v1 from EEMS (built with Eigen version 3.2.2 and Boost version 1_57, and using rEEMSplots version 0.0.0.9000),<br>-the ape R package (version 5.5),<br>-Plink (version v1.9),<br>-qpAdm (version 7.0),<br>-OrientAGraph (version 1.0),<br>-fastsimcoal2 (version 2.6.0.3), |

-the mapplots package (version 1.5.1),
-the ncf R package (version 1.2-9),
-the geosphere R package (version 1.5-10) ,
-the GeoRange R package (version 0.1.0),
-LOCATOR (version 1.2), and;
-the GViz R package (version 1.36.2).

For manuscripts utilizing custom algorithms or software that are central to the research but not yet described in published literature, software must be made available to editors and reviewers. We strongly encourage code deposition in a community repository (e.g. GitHub). See the Nature Portfolio guidelines for submitting code & software for further information.

## Data

Policy information about availability of data

All manuscripts must include a data availability statement. This statement should provide the following information, where applicable:
- Accession codes, unique identifiers, or web links for publicly available datasets
- A description of any restrictions on data availability
- For clinical datasets or third party data, please ensure that the statement adheres to our policy

All collapsed and paired-end sequence data for samples sequenced in this study are available in compressed fastq format through the European Nucleotide Archive under accession number PRJEB44430, together with rescaled and trimmed bam sequence alignments against both the nuclear and mitochondrial horse reference genomes. Previously published ancient data used in this study are available under accession numbers PRJEB7537, PRJEB10098, PRJEB10854, PRJEB22390 and PRJEB31613, and detailed in Supplementary Table 1. The genomes of ten modern horses, publicly available, were also accessed as indicated in their corresponding original publications (see Jonsson et al. PNAS 2014, Der Sarkissian et al. Curr Biol 2015, Renaud et al. 2018, Jagannathan et al. 2019, Andersson et al. 2012).

# Field-specific reporting

Please select the one below that is the best fit for your research. If you are not sure, read the appropriate sections before making your selection.

☐ Life sciences        ☐ Behavioural & social sciences        ☒ Ecological, evolutionary & environmental sciences

For a reference copy of the document with all sections, see nature.com/documents/nr-reporting-summary-flat.pdf

# Ecological, evolutionary & environmental sciences study design

All studies must disclose on these points even when the disclosure is negative.

| Study description | We have sequenced 264 ancient horse genomes, including from regions and/or time periods that remained uncharacterized at the genetic level. Additionally, we complemented the data previously generated for nine ancient horses and included a total of ten modern horse genomes, selected to represent a whole diversity range of breeds/populations. We applied procedures aimed at minimizing the impact of post-mortem DNA damage on sequence quality and identified a total of 10M+ high-quality polymorphic sites in our data set. For sites of critical importance, ancient genome variation was characterized from more than a single genome, providing individual replicates of the signatures identified. Metadata considered in the analyses included the GPS coordinates of the excavation sites and the age of the samples analyzed, most often assessed from radiocarbon dating. The data set gathered helped solved long-standing controversies about horse domestication. |
|---|---|
| Research sample | Ancient horse remains (Equus ferus caballus) were collected and screened by sequencing following DNA extraction and next-generation DNA library preparation in state-of-the-art ancient DNA facilities. Shallow DNA sequencing helped measure DNA preservation levels so as to identify those specimens for which whole genome sequences could be characterized by means of shotgun DNA sequencing. We mainly focused on the time period spanning the first to fourth millennium BCE (Before Common Era), as it encompassed the whole time frame of horse domestication. We, however, also included older samples so as to characterize the pre-domestication population structure. Finally, we also sequenced the genome of one historical 'Tarpan' specimen, given the contentious status of this population regarding horse domestication. Archaeological sites, individual specimens as well as their respective radiocarbon dates and sex assignments (as inferred from DNA data) are provided in Supplementary Table 1. Supplementary Methods provide additional information on each archaeological site. |
| Sampling strategy | Sampling was conditioned by the availability of ancient remains. In order to cover the whole temporal and geographic range relevant for horse domestication, we have gathered together an extensive team of archaeologists and curators in charge of material collection from across Eurasia and North-Africa, and with full authority to undertake research-based activity on such material. The resulting data represents a large collection of 264 ancient genomes. Combined with a selection of genomes previously published, both modern and ancient, they provided adequate data and statistical power for statistical testing. The robustness of our analyses was assessed through a range of appropriate statistical methods, including bootstrapping, Maximum Likelihood, replicates and formal statistical tests. Sampling procedures were aimed at minimizing destruction, and were focused, whenever possible, on osseous remains such as petrosal bones, that are generally associated with better average DNA preservation. |
| Data collection | A majority of the remains consisted of loose petrosal bones and teeth that lost connection with their original skulls. These were either directly shipped to Ludovic Orlando by the archaeologists and/or curators in charge, or delivered in person to him (as he visited his collaborators, or when his collaborators visited his laboratory). Sampling for DNA extraction and radiocarbon dating were performed in the ancient DNA facilities of the CAGT laboratory using appropriate drilling instruments under flow-hoods and in an environment with filtered, positive air pressure. Routine procedures at CAGT are aimed at minimizing both destruction and contamination. |

| | |
|---|---|
| Timing and spatial scale | All archaeological remains investigated in this study are described in Supplementary Table 1 and have been collected, mostly from 2017 (although the earliest was collected in 1995). A study that was then ongoing in our laboratory suggested that horse husbandry at Botai did not give rise to modern domestic horses, which implied that another domestication centre was yet to be found. As the Botai culture is located in the second half of the third millennium BCE of central Asia, we mainly decided to focus our attention on the following 1000 years, but also extended sampling to other regions, including those previously described as potential domestication centres. Both due to the complexity of the horse population genetic structure at the time, and the relatively limited abundance of horse archaeological remains during the third millennium BCE, we extended sampling to prior the third millennium in order to increase resolution and gain insights on those lineages pre-dating domestication. Finally, the discovery that a massive population turnover took place from the late third to early second millennium BCE led us characterize the second millennium BCE more extensively at the genetic level. Overall, over 2,000 archaeological remains have been collected since 2017. They have been screened for DNA and subjected to shotgun DNA sequencing whenever showing sufficient DNA quality, following a routine data production pipeline at CAGT. Decisions on which locations and time periods should be given priority or added to our data set were made as the project made progress following preliminary analyses of the data collected, and re-assessed on approximately a monthly basis. The genetic data analyzed in this study correspond to a main data freeze that was done in July 2020, and supplemented with additional data from November 2020. |
| Data exclusions | No data generated in this study were excluded from our analyses. Following standard procedures in ancient DNA research, transition substitutions were masked from most analyses in order to limit the noise added by post-mortem DNA damage, largely inflating DNA sequencing rates. We limited our analyses to only a fraction of horse genomes previously reported in order to (1) avoid technical batch effects that may have structured the data due to the different DNA library construction and sequencing technologies used, (2) reduce computational burden and (3) only those time and/or geographic regions that were relevant for the present study. |
| Reproducibility | The quality of the genome sequences was validated by calculating their respective sequencing error rates. Data uncertainty was accounted for in our analyses (e.g. base qualities) to assess robustness.  Genome originating from the same stratigraphic layers in a given site carried similar genetic information. |
| Randomization | Groups were defined to reflect ancient horse populations, eg horses from Iberia carrying similar genome variation were clustered together, and considered to form an ancient Iberian population. Such clustering was based on phylogenetic inference and on the temporal and geographic provenance of the specimens. |
| Blinding | Blinding was not applicable to this study since the geographic and temporal metadata associated with each sample was key to carry research (and prioritize those areas worth further sampling and investigations). However, computational analyses were first carried out at the individual level, before groups/populations were defined. |

Did the study involve field work?  ☐ Yes  ☒ No

# Reporting for specific materials, systems and methods

We require information from authors about some types of materials, experimental systems and methods used in many studies. Here, indicate whether each material, system or method listed is relevant to your study. If you are not sure if a list item applies to your research, read the appropriate section before selecting a response.

## Materials & experimental systems

| n/a | Involved in the study |
|---|---|
| ☒ | ☐ Antibodies |
| ☒ | ☐ Eukaryotic cell lines |
| ☐ | ☒ Palaeontology and archaeology |
| ☒ | ☐ Animals and other organisms |
| ☒ | ☐ Human research participants |
| ☒ | ☐ Clinical data |
| ☒ | ☐ Dual use research of concern |

## Methods

| n/a | Involved in the study |
|---|---|
| ☒ | ☐ ChIP-seq |
| ☒ | ☐ Flow cytometry |
| ☒ | ☐ MRI-based neuroimaging |

## Palaeontology and Archaeology

| | |
|---|---|
| Specimen provenance | The samples that were analyzed in this study were collected from a range of environmental conditions, spanning Northern African to Siberian excavation conditions. As this involved sampling from across Eurasia and different procedures between countries and institutions, key contact persons were identified in each country so as to access relevant material and coordinate legal authorization to sample material for DNA analysis and radiocarbon dating. Samples were collected with permission from the organizations holding the collections and documented through official agreement letters provided by the named archaeologists and/or curators and/or directors of relevant institutions, all named below. Official agreements to share material for partially destructive research were sent to Ludovic Orlando in the form of authorization letters, as part of the ethical framework established for the ERC PEGASUS Consolidator grant (681601). We sought every opportunity to access samples as part of collaborations with other research projects so as to both save resources and avoid double sampling, and, thus, ultimately minimize destruction. The archaeological sites where each individual specimen was excavated are listed in Supplementary Table 1. The exact locations of each remain are provided with full reference to the names of the archaeological sites and their GPS coordinates. The following list provides the sites and names of those collaborators (together with the head of their institution when necessary) who granted access to the corresponding material, with reference to letters and permits where appropriate:<br><br>-Kostenki 15 (layer 25183 (179), square 623/K-25), Medvezhiya (layer 34763 (7.2-7.4 square N-1, 4.4-4.6 square 15-n, 1.5-2.0 square |

N3 and 5.0-5.2 square N1)): Dr. N. S. Chernetsov (collections of the Archaeological Institute of the Russian Academy of Sciences, Sankt-Petersburg; authorization letter nb 12505-2145/239),

-Tepe Hasanlu, Tepe Sagzabad: Dr. M. Mashkour, Dr. M. S. Salehi (Archaeology Institute of the University of Teheran, authorization letter nb 654-854), Dr. B. Omrani (Research Institute of the Iranian Cultural Heritage and Tourism, authorization letter nb 9810308),

-Aygurskiy: V. A. Babenko (Stavropol, excavation 'Nasledie' 2000-2001, license nb 2000-776 and nb 2001-776, exported in 2016 to the German Archaeological Institute, Berlin, Germany),

-Ullu: Dr. A.B. Belinskij (Stavropol, excavation 'Nasledie' & DAI Eurasia-Department 201, license nb 2013-633, exported in 2013),

-Tarpan from the Kherson Region: Dr. T. V. Kuznetsova (Department of Paleontology, Faculty of Geology, Moscow State University; collections of the Zoological Institute, Russian Academy of Sciences, Sankt-Petersburg, Russia; collection nb O.521),

-Yukaghir: Dr. A.V. Prokopiev (Geological Museum of the Diamond and Precious Metal Geology Institute, Siberian branch, Russian Academy of Sciences; letter nb 304-03-21-0443/503),

-Yana complex of sites, Divnogor'ye 9: Pr. V. Pitulko, Pr. A. A. Bessudnov, O. I. Boguslavskiy (Institute of Material Culture, Russian Academy of Sciences; projects 16-18-10265 and 21-18-00457 from the Russian Science Foundation, field permits 468 (2016), 738 (2017) 779 (2018); confirmation letter nb 14102/33-772.4-263),

-Algay, burial mound at Berezovaya Mountain, Krasnosamarskoe, Noviye Kluchi III, Oroshaemo I, Potapovka, Repin Khutor, Turganik, Ouren, Utevka VI, Uvarovka II, Varfolomeevka: Pr. P. Kuznetsov, and Rector Dr. O.D. Mochalov (collections from the Archaeological Laboratory at the Samara State University of Social Sciences and Education, Russia; confirmation letter nb 03-01-Myzea),

-Sosnovka, Sintashta, Bol'shekaraganskii, Kameennyi Ambar 5, Aleksandrovskoe IV, Serpievskaya, Nikolskaya, Sholma-1, Pershinskaya, Verkhnegubakhinskaya: Dr. M. G. Golovatin (Institute of Plant and Animal Ecology, Ural Branch of the Russian Academy of Sciences; confirmation letter nb 16353-2115/214),

-Semenovka 1: Pr. N. S. Kotova, Pr. A. B. Bujskikh (Institute of Archaeology, National Ukrainian Academy of Sciences, Kiev, Ukraine; confirmation letter nb 125/01-19-334),

-Ganga-Tsagaan-Ereg, Khantain-tov, Monostoy-Nuga, Ushkin-Uver, Arzhan-1, Chinge-Tey-I, Hyena's Lair, Novoilinka-III, Novoilinka-VI, Bijke-V, Choburak-I, Kuyum: Pr. A. Tishkin, Dr. I. I. Nazarov (Institute of History, Barnaul University; project 19-59-15001 co-funded by CNRS and the Russian Foundation for Basic Research),

-Morin Mort, Bor Shoroonii, Zeerdegchingiin Khoshuu, Zunii Gol, Zuunkhangai, Ulaan Tolgoi: Pr. J. Bayarsaikhan (collections from the National Museum of Mongolia, Ulaanbaatar; exported in 2015 and 2015 under research agreement nb 20150315), and Dr. W. Taylor (Fulbright US Student research award nb 34154234, National Geeographic Young Explorer's grant nb 9713-15), National Science Foundation Doctoral Dissertation Improvement Grant nb 1522024),

-Burgast-1, Tatsyn Ereg: Dr. Associate Pr. G. Eregzen (collections of the Institute of Archaeology, Mongolian Academy of Sciences, Ulaanbaatar, Mongolia; confirmation letter nb 01/48),

-Pietrele, Dr. E. Nicolae; Căscioarele, Nandru Peștera Curată: Dr. A. Bălășescu (Bioarchaeology Department, Vasile Pârvan Institute of Archaeology, Romanian Academy),

-Cova Fosca: Pr. C. Rosa Olaria Puyoles (Catedra de Prehistoria, Universitat Jaume I, Castello Spain),

-Els Vilars, Cantorella, Sigarra, La Monédière: Dr. A. Nieto Espinet (History Department, University of Leida, Spain),

-Casas del Turuñuelo, Pr. S. Celestino (Instituto Arqueologia Merida, Spain; projects IB10131 and IB18060, both funded by Junta de Extremadura and European Regional Development Fund),

-Althiburos: Dr. Silvia Valenzuela-Lamas (Institucio Mila i Fontanals, Barcelona, Spain),

-Fengtai, Zambujal, Kirklareli-Kanligecit, Garbovat, Dunaújváros-Kosziderpadlás, Miciurin, Nitriansky Hrádok, Arzhan-II: Prof. Dr. N. Benecke (German Archaeological Institute, Berlin, Germany), and Prof. Dr. A. Ludwig (Leibniz-Institut fur Zoo- und Wildtierforschung, Berlin, Germany),

-Shilikty: Pr. A. T. Toleubayev, Associate Pr. R. S. Zhumatayev, Dean of Faculty M. S. Nogaibayeva (collections from the Department of Archaeology and Ethnology, Al-Farabi Kazakh National University, Almaty, Republic of Kazakhstan, confirmation letter nb 15-23-720); Krasnyi Yar, Botai: Pr. Viktor Zaibert, Dean of Faculty M. S. Nogaibayeva (collections from The Research Institutee 'Archaeology and the Steppe Civilisations", Al-Farabi Kazakh National University, confirmation letter nb 1523-602),

-Kent, Ashchisu, Novoil'inovskiy 2: Dr. V. G. Loman (collections of the Saryarka Archaeological Institute, E. A. Buketov Karaganda University, Karaganda, Republic of Kazakhstan; confirmation letter nb 4-21/151),

-Michurino I, Shiderty III, Borly, Borly 4: Dr. Viktor K. Merz, Acting Deputy Chairman of the Board for academic work P.O. Bykov (collections of the A. Kh. Margulan Joint Archaeological Research Centre Toraighyrov University, Republic of Kazakhstan; confirmation letter 107/1232),

-Belkaragai, Bestamak, Kozhai, Halvai: Pr. A. Logvin, Dr. I. Shevnina, Acting Vice-Rector on Science, Internationalization and Digitalization G. Shakamal (A. Baitursynov Kostanay Regional University, Kostanay, Republic of Kazakhstan; confirmation letter nb 15-30-09/1052),

-Kaposujlak-Vardomb: Pr. G. Kulcsar (head of Department of Prehistory, Institute of Archaeology, Research Centre for the Humanities, Hungarian Academy of Sciences Centre of Excellence; confirmation letter BTK KP/1557-1/2021),

-Bad Pirawarth: Dr. E. Pucher (Naturhistorisches Museum Wien); affiliated archaeological project researcher: Dr. C. Schwall (Department of Prehistory and Western Asian/Northeast African Archaeology, Austrian Archaeological Institute, Austrian Academy of Sciences; sample made available through project nb OAW Innovationsfonds 23100 – OAW4002),

-Březno u Loun (pit 629 ID 510), Černý Vůl (104/1975-1977 ID 250), Holubice (2/2005 ID 146), Litovice (it 118/2004 ID 352), Stránská skála, Tištín (pit 553/2002 ID 175312), Toušeň – Hradišťko (trench 5/1976 ID P3416), Vliněves (8805/2007 ID 4349), Tuchoměřice (36/2005 ID 428), Tuchoměřice – Kněžívka (73/2007 ID 1176): Mgr. J. Marik (Institute of Archaeology of the Czech Academy of Sciences, Prague, Czech Republic; confirmation letter nb ARVP-3464/2021),

-Gordinesti III burial ground, Gordinesti II-Sinca goala settlement: Dr. H. Shephard (Archaeological Institute of America, US), Dr. V. Ghilas (Institute of Cultural Heritage, Centre of Archaeology, Academy of Science of Moldova; confirmation letters nb 06/90-06/91 27.04.2017; custom declaration from May 4th 2017),

-Etiolles, Le Closeau, Tureau des Gardes: Dr. O. Bignon-Lau (ArScAn-UMR 7041 CNRS, Nanterre, France),

-Igue du Gral, co-directors of excavation campaigns at the site: Dr. J.-C. Castel, Dr. M. Boudadi-Maligne (Museum de Genève, Swizerland & PACEA, UMR 5199, University of Bordeaux, France),

-Tarquinia monumental complex: Pr. Dr. G. Bagnasco Gianni (Dipartimento di Beni Culturali E Ambientali Etruscologia, Universita Degli Studi di Milano, Italy),

-Miechow 3, Mozgawa, Pielgrzymowice, Kazimierza Wielka, Slawecinek: Pr. J. Wilczynski (collections of the Institute of Systematics and Evolution of Animals, Polish Academy of Sciences, Krakow, Poland); Mozgawa site: Dr. M. Moskal-del Hoyo excavation under project 2013/11/B/HS3/03822, funded by the Polish National Science Centre 2014-2017),

-Goyet: Dr. A. Folie (Paleontological Collections, Royal Belgian Institute of Natural Sciences, Brussels, Belgium; destructive analysis request DAD-2015-06),

-Hohler Stein bei Schwabthal: Dr. T. Seregély (collections of the Institute for Archaeology, Heritage Conservation Studies and Art History, University of Bamberg, Germany; excavation campaign from 2008),

-Acemhoyuk, Cadir Hoyuk, Kosk Hoyuk: Pr. Dr. A. Oztan, PR. G. McMahon, Pr. B. Arbuckle (Department of Anthropology, University of North Carolina at Chapel Hill; authorization from Aksaray, Nigde and Yozgat museums; permit references B.16.0.KVM.0.13.01.00.155.02 (YA.2011.52) 36712, B.16.0.KVM.4.51.00.01/160.02/483),

-Asva, Ridala: Pr. L. Lougas (Archaeological Research Collection, Tallin University; sampling report AI-PP-345, sample references AI 3799:467 and AI4261),

-Kent's cavern: B. Chandler (Collections and Engagement Manager, Torquay Museum, United Kingdom),

-Chalk Hill, Magor: Dr. R. Bendrey (School of History, Classics and Archaeology, collections from the University of Edinburgh, United Kingdom), and;

-Ginnerup: Dr. L. Klassen (collection from the Research Department, East Jutland Museum, Randers, Denmark).

The sample TP4 from Tachti Perda was made available from the remaining DNA material from one of our previous studies (Gaunitz et al. Science 2018).

| Specimen deposition | The specimens are available upon direct request to the archaeologists and/or curators in charge. Photographs taken while processing the samples for DNA are also made available in Ludovic Orlando's laboratory. |
| --- | --- |
| Dating methods | A total of 207 archaeological samples have been radiocarbon dated at the Keck Carbon Cycle Accelerator Mass Spectrometry Laboratory, UC Irvine. Raw and calibrated radiocardon dates are provided in Supplementary Table 1 (IntCal20 calibration curve, OxCal Online). |

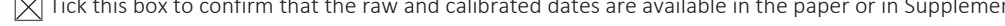 Tick this box to confirm that the raw and calibrated dates are available in the paper or in Supplementary Information.

| Ethics oversight | No ethical oversight was required as the material was sampled following procedures aimed at minimizing destruction and damage, following discussion, supervision and agreement with those curators in charge of the collection. Additionally, DNA analyses aimed at sequencing whole genomes instead of a restricted list of target loci, which eliminates the need for additional sampling in the future. Combined, this complies to the 3R principles. |
| --- | --- |

Note that full information on the approval of the study protocol must also be provided in the manuscript.

