## [Peer Review File · Nature]

Manuscript Title: The origins and spread of domestic horses from the Western Eurasian steppes

Redactions – Third Party Material

Reviewer Comments & Author Rebuttals

Reviewer Reports on the Initial Version:

Referee #1 (Remarks to the Author):

No other animal has had such a great historical impact on the ebb and flow of human civilizations. Yes, camels could go farther, cattle could feed us, and dogs have been with us longer, but only the horse was a king maker. The when and where of the beginnings of our relationship with the horse was the single greatest mystery among study of domesticated species. Previously these authors turned the history books on their heads, naming the oldest archeological record of equine-human collaboration as belonging to an entirely different species, the now wild-again Przewalski's horse. In a paleo-genomics tour-de-force their large collaborative team has now generated an impressive The older, slower, migration of domesticated horses with the expanse of the Yamnaya and PIE languages was a very logical hypothesis. I am indeed quite shocked that this wealth of new evidence supports an entirely different story. With the discovery that the Botai were not the origin of modern domesticated horses, my second guess would have been an older origin, and a slower entrance into domestication, as might have corresponded with the travel of the Yamnaya. What is most shocking, is the rapidity with which the new DOM2 line seems to have entirely replaced all other human-associated equid species. That this shift in species-use was not also associated with a shift in the human genetic diversity of the region is quite staggering. This implies two things: first, that the new species of DOM2 was vastly superior as a domesticated partner, and second, the peoples who developed and utilized these horses traded or gave them away. One need only look at the later example of Genghis Khan to see what potential power this new species conferred, and the magnitude of his impact on the human genome is readily visible as a result. And yet, the new horse took the ancient world not accompanied by force, but apparently, peaceably.

To have convinced so many diverse cultures to abandon their local equine domesticates like the Tarpan and Przewalski's horse, replacing them with breeding stock from a foreign source, the new DOM2 horse must have been an enormous improvement in equine suitability for human purposes. From the physiological standpoint, the basis of this change is a tantalizing target for future study. The presented data on signatures of selection characteristic of the new DOM2 horse are remarkably clear for this type of analysis, suggesting that changes at just two loci may have driven much of the improvement in the suitability for domestication phenotype.

These two loci hold genes with putative roles in features that remain key to the economic value of the domesticated horse today. Arguably, the behavior of the horse is only more important today than in pre-history, as their role has shifted from transportation and beast of burden to one of a companion animal and athletic partner. Intractable behavior in an animal that is easily 5-10 times (or more!) as massive as its human handler often quickly ends in disaster, and behavioral issues are often cited as the second most common rationale for relinquishing a horse to a rescue shelter (with incurable medical problems and ranking first). Given the complexities of the mammalian brain, documentation of the action of a neurodevelopmental pathway contributing to anxiety has obvious comparative biomedical applications.

The identification of GSDMC is also a notable one, as the equine back remains an intractable source of performance limiting pain, and a source of frustration for owners and veterinarians alike. In addition to the cited role of this gene as a putative biomarker for spinal stenosis, it also recently appeared in a GWAS for chronic back pain (Suri et al. 2018, Genome-wide meta-analysis of 158,000 individuals of European ancestry identifies three loci associated with chronic back pain) and this phenotype is perhaps more analogous to the modern clinical signs experienced by the equine athlete. As to the etiology of this syndrome in horses, there is much that is still unknown, though body size and conformation are likely risk factors. Modern domesticated horses vary

considerably in the skeletal morphology of the back, encompassing polymorphisms in the number of thoracic and lumbar vertebrae, and in the rate of sacralization of the lumbar vertebrae (Stetcher, 1962, *Anatomical Variations of the Spine in the Horse*). I do not know that this variation has specifically been examined in historical or pre-historical samples, or for that matter, across equid species other than the rather early paper by Stetcher. The supplemental document notes that in many cases the skeletal remains sampled for DNA were unfortunately incomplete. It would be fascinating to know if these ancient species diverged for vertebral number, and therefore, likely in back conformation. Some evidence for back pathology has been observed in ancient, domesticated horses previously (Levine et al. 2004, *Abnormal thoracic vertebrae and the evolution of horse husbandry*) and may be worth mentioning.

In any case, I hope I have articulated the reasons why the work is novel, and of extreme importance, not only to the field, but to related studies in biology and agriculture, as well as history and the social sciences. I also believe it will hold interest among a diverse audience. As befits this journal, it is well-written with accessible language to a scientific audience. The statistical methods are rigorous and well-applied to datasets that are described in excellent detail. The addition of the new Struc-f4 tool will be a valuable one to the field, providing a new way to visualize these highly dimensional comparisons. I have noted two typos in the main document that need attention: "bootstrap" rather than "bootstrap" and "spreading spread across".

Referee #3 (Remarks to the Author):

The article from Librado et. al. reports the second (after Fages et al. 2019 of comparable size) most exhaustive collection of genomes from ancient horses across Eurasia produced so far (N = 264 or 242 see comment below). Most importantly, they cover specimen from previously unreported time periods in key geographic regions such as the Bronze Age in the Pontic Steppe. They also produce the largest collection so far (to my knowledge) of very ancient, Paleolithic to Neolithic, horse genomes from various geographic regions across Eurasia (Including far east Siberia, the Urals, Europe and two Neolithic Anatolian wild specimen, I believe the first ever sequenced).

This impressive dataset allowed the authors to investigate the evolution horses and especially the spread of domestication across time and space at an unprecedented level, pinpointing a very plausible and convincing location and time period for the origin and spread of modern domestic horses (the so called DOM2 lineage). They further report indirect-insights into ancient human populations histories, showing that the genetic turnover in early Bronze Age Europe linked to the Yamnaya expansion and the spread of Indo-European languages was not accompanied by genetic turnover in horses. Whereas in Asia the spread of steppe people temporally coincides with horses' turnovers.

GENERAL OPINION

I find the size and novelty of the data generated and of the findings presented in this study to be worth for a Nature publication. I don't have major concerns regarding the experimental analyses, processing of the data and the validity of the population genetic analyses and the overall interpretation of the main findings presented.

Although I have two main points that I would like the authors to address. First, I find in general quite difficult to match the descriptions of the results in the text with the figures. This is especially true for Figure 1 and also it is in general hard to follow the link between the results from different analyses also because of use of acronyms in the plots that are not clearly defined in the text (see detail in Main points below). Second, I understand and agree that the findings of the Bronze Age post-Yamnaya expansion of DOM2 from the Pontic steppe is the main focus, but I still think that it would be worth dedicating more space or to better clarify the evolutionary relationship of the more ancient Eurasian horses (Paleolithic to Neolithic/Eneolithic) that I think constitute a second important novelty of the study and in the current draft there are some inconsistencies not fully

solved or explained (see 2 major comments below).

MAIN POINTS

Throughout the text, names of geographic regions, genetic clusters and specific time periods are used to explain and interpret the results. Although, Figure 1 reports the entire set of ancient genomes, phylogeny (1b) and f4-struct-genetic-components (1d) plots with only the macro geographic coloring and the cryptic ID of each individual to orient the reader. I think this figure as it is could be a good supplementary figure as it exhaustively reports all the data but the main figure should allow the reader to match the descriptions of the main text. Maybe the authors can consider making a more summarized figure 1, collapsing some lineages in the phylogeny for example and providing "group" labels in the f4-Struct and phylogeny matching with the one mentioned in the text and the ones in the Extended Table 1 to better orient the reader. Beside this, I think that Fig 1a, is quite small and it's difficult to see properly the location of the samples on the map. Also, the current time scale on the y-axis doesn't allow to appreciate the more recent and narrow time periods < 10,000 BP that is the more extensively discussed time period in the article. I would suggest to enlarge the time scale or add a zoom-in into this time window. To gain space I think that figure 1c (the f4-struct matrix) can be summarized with less entries or reduced since the macro clusters that the matrix distinguish would be still visible.

The second main point is that I cannot fully reconcile some patterns reported by different analyses (e.g. by the Strct-f4 analysis with the evolutionary models of OrientAGraph) concerning the more ancient lineages (while the main result, the origin and expansion of DOM2, is instead quite clear and solid). This is in part due to the difficulty of matching the clusters used for the analyses in Figure 3 with the description of the results in the text and the individual-based analyses of Figure 1 (so related to the first main point). Also, please check the definition of the acronyms used, for example I didn't find anywhere in the text UP-SFR and ELEN, used in OrientAGraph plots.

More precisely, I cannot fully reconcile the following aspects:

1) The CWC (Corded Ware) group in OrientAGraph is placed ancestral before the split of Botai/NEO-ANA and the lineages leading to DOM2. But in Strct-f4 the CWC associated horses are represented by ~50% of a component maximized in Botai and NEO-ANA and ~50% by a component maximized in what seems to me Paleolithic and Mesolithic Russia and Europe (The light blue and dark purple samples in Figure 1). The authors comment on this inconsistency in the Supplementary text as a cline of ancestry formed via expansions out-of-Anatolia through prolonged contact that would not be picked up by OrientAGraph. I wonder if another admixture graph method would be able to test the fit of such a model (see comment below).

2) There are some groups ENEO-ROM, NEO-NCAS that are listed in the Extended Table 1 and seem to correspond with the samples ID clustering with C-PONT and TURG samples in Figure 1b and d (so related to the origin of the DOM2 lineage), that don't seem to be mentioned in the main text or included in other analyses. This potentially mixed genetic profiles seem also to be discordant with the OrientAGraph trees (analogous to the point 1 case) at least for C-PONT and TURG groups included in these trees that are modeled as un-admixed lineages. I wonder if the authors could comment and explore more the evolutionary relation of these lineages ancestral to the DOM2 branch.

3) In a previous study and here is confirmed that the Przewalski horses are the descendants of the ancient Botai horses. Here, while they do cluster together in the phylogeny and are close in the MDS plots, their f4-struct genetic profile seem to differ substantially: the Przewalski horses carrying high amount (~50%) of the ancestry maximized in DOM2 and lower proportion of other lineages including but at quite low proportion the component that is maximized in Botai horses. Can the authors comment on this? Are the Przewalski admixed with DOM2 and other sources or

what can cause this Struct-f4 pattern? Is this high admixture in line with previous results?

4) The two NEO-ANA horses in the phylogeny of Figure 1b cluster among the C-PONT or TURG samples (the closer lineages ancestral to DOM2), but in all other analyses they result closer to the Botai group (e.g. same component in f4-struct and cladal in OrientAGraph). Could the authors comment on this? Also, I find very interesting the similarity between the very geographically distant Eneolithic Botai and the Neolithic Anatolian horses (NEO-ANA), The authors comment on this in the supplementary as expansion of NEO-ANA into the CA steppe, this is indeed a possible explanation but what could be the drive for this expansion also in relation to what we know about ancient human populations? If these are wild horses, why in comparison to the CWC case (point 1) we don't observe a cline of two ancestries blending together (the new incoming NEO-ANA and the older CA lineages) but instead a complete replacement (assuming it is in fact an expansion from Anatolia into the CA steppe)?

Finally, in the OrientAGraph trees some key ancient lineages have not been included (e.g. ancient Iberians, IBE, and the ancient Urals, S-Ural and also other groups that are listed in the Extended Table 1 but not mentioned in the text). I think it would be worth to explore more the relationship between all these deep Eurasian lineages identified with the new data. This is just a suggestion, but I was wondering if the authors would consider running a tool like qpGraph or other demographic modeling methods (SFS based for example) that could test the fit of specific complex evolutionary models user-specified as an independent validation of the ones obtained with OrientAGraph. It could maybe help in reconciling the evidences from different analyses by testing for specific admixture events/gene flows vs monophyletic split between lineages.

MINOR POINTS

Main text

- Figure 2 - I would increase the visibility of the pie charts in the map. Suggestions: the sizes of the maps could be increased by decreasing the scatterplots on the right. Also, in Figure 2b since autosomes and chromosome X show the same trend only one of the two plots could be kept and the other added in the supplementary or the two trends could be superimposed in the same plot.

-Page 6: "Neighbor joining phylogenomic inference revealed highly supported monophyletic groups". See main point 2, there seem to be admixture between several lineages according to Struct-f4 and OrientAGraph so I would not generalize about monophyletic lineages.

-Page 6: Where are the *Equus lenensis* lineages mentioned in the text located on the tree and Struct-f4 plots (figure 1)?

-Page 6: "A second group comprises horses from Europe, including UP Romania, Belgium, France, and Britain as well as sixth-to-third mill BCE horses from Spain to Scandinavia, and France to Hungary, Czechia and Poland (HCP)". It is a long list of countries but acronym HCP refers only to Hungary Czechia and Poland? It is bit confusing phrased this way.

-Page 6: "The ancestry profile of sample RONPC06_Rom_m34801 indicated that the Anatolian expansion into the Romanian Carpathians started before ~35,000 BCE." I might be missing something but this sample is much older than the two NEO-ANA ones. I don't think an expansion from Anatolia into Romania is the only possibility explaining the f4-struct genetic profile of these samples especially given that the Romanian specimen is much older.

- Page 7: "qpAdm modelling indicated a genetic profile acquiring DOM2 ancestry following gene flow into CWC-related horses from southern Thrace (Kan22_Tur_m2386), but not from Crimea (Ukr11_Ukr_m4185)." Where can I see this result? no reference to figure or table.

- Page 7 DOM2-specific biological adaptation

I think the F_{st} signal in the two genomic regions identified is very strong so I say this as a minor comment and suggestion: given the rich dataset of ancient lineages across Eurasia and a model of their evolutionary relationships, would the author consider running a more evolution-informed positive selection test beside the F_{st} ? For example, PBS (first introduced in Yi et al. 2010) is an extension for F_{st} testing for deviations in a 3-population tree given a test group, a sister group and an outgroup. Also, maybe is beyond the scope of this study and maybe the sample size is too small but it would be interesting to see if it possible to pinpoint signals of adaptation to the different natural environments among the different very ancient lineages (e.g. the Paleolithic/Neolithic Siberian lineages compared to the European ones).

- Page 11 typo:

"For which were sequencing was done", remove "were".

-Page 13 "Was investigated using the f_3 -statistic-based ancestry... implemented in qpAdm" qpAdm is solely based on f_4 -statistics.

-Page 13 "Struct- f_4 , ancestry components and Multidimensional Scaling"

Struct- f_4 is also introduced in Fages et al. 2019 but here it seems presented as a newly developed method. Can the authors comment on what has been newly developed since Fages et al. 2019?

Supplementary text

- Page 31 Application of Struct- f_4 to empirical data:

How is the K-clustering done? Why the modern genomes were excluded? How would this batch effect influence f_4 -stat? Could the authors elaborate more on this?

- Page 28: The authors perform quite an accurate filtering for C to T ancient DNA damage (e.g. UDG-treatment, PMDtools to filter damaged reads, 5bp end trimming) but at the end they seem to retain only transversions for all the analyses presented. If the transitions are never used why performing these quality controls? Maybe I miss-understood something, could the authors elaborate on this?

- Extended Table 1: I counted 242 new genomes produced in "this study" not 264. Please double check the numbers. Also, can the authors elaborate on why they considered only subset of samples from previously published studies (for example from Fages et al 2019, only 13 were considered)?

Referee #4 (Remarks to the Author):

This is a landmark study of the domestication and spread of horses. It will be of interest to scholars in a broad range of fields including domestication, evolution, genetics, archaeology, history, agriculture, and to the general public.

As the authors say, domestication of horses transformed the subsistence, mobility, and geopolitics of ancient societies. Understanding the process of horse domestication has been challenging, though, because the mobility of early horse keepers resulted in interbreeding among distant horse populations, domestic and wild. Several generations of zooarchaeological and genetic studies identified early domestic horses at Botai in N Kazakhstan and suggested the possible existence of a second center further west. Genetic research demonstrated that the Botai horses were not ancestral to modern domestic horses but data bearing on a second center has been elusive. What makes this study by Librado et al so significant is the identification of the lower Don-Volga region of the Western Eurasian Steppe as the second center and origin of all modern horses

ridden today.

The sample size, broad regional and temporal coverage and clear shifts in population structure ~4200 years ago, from geographically isolated to a lack of population structure characteristic of mobile domesticates, makes this a robust finding. The authors sequenced by far the largest sample of ancient horse genomes studied to date, including 264 horses from a range of time periods and Eurasian regions. Beyond genetic information, the data sets presented in SI include detailed site context and a large and important new radiocarbon data-base, which strengthen the reliability of results.

The identification of the time and place of a second center for horse domestication is a sufficiently important and novel finding to warrant publication in Nature. But there is more. The authors present data that challenges longstanding assumptions about invasion, horses and the history of societies in Europe. Largely based in linguistics, it has long been argued that steppe herders associated with the Yamnaya culture expanded into Europe ~5000 years ago bringing Indo-European languages and steppe horses to the region. Horse-based mobility and warfare were thought to be important factors underlying this population movement. This is not supported by the results of Librado et al, which show no evidence for introduction of steppe horses to western Europe 5000 years ago. Instead, data from this study reveal that horses domesticated in the Don Volga region of the western steppe's spread rapidly to the west about 1000 years later, ~4000 BP associated with chariot use. Interestingly, Librado et al's data indicate a different pattern driving the eastward spread of modern horses into central Asia, where Don Volga horses spread with the steppe Sintashta culture eastward in association with chariots and Indo-Iranian language. This ultimately resulted in replacement of early domestic horses of Botai in Khazakstan and horses further east (eg Mongolia) with modern domestic horses. As a result, this study provides fundamental insights into the origin of modern horses and diverse social mechanisms that resulted in Bronze Age social change, as well as shifts in domestic horse biodiversity.

This research also finds evidence for selection on locomotion and behavior with horse domestication, which is an important contribution to understanding variability in sites of selection in animal domestication. The exceptional preservation of aDNA in Eurasian regions in which horses were domesticated also make this sample significant for domestication studies broadly. To date, there are few ancient genomic samples of this scope for large mammals.

Clarity and accessibility

Overall, the narrative structure is clear and logical. However, the introduction is not clearly written due to assumption of previous knowledge, phrasing and word choice (see below). The abstract is clear, the discussion mostly clear and conclusions are appropriate. This manuscript has a distinguished group of authors and I suggest greater recruitment for some edits by those with extensive experience contextualizing horse domestication for non-specialists.

An additional map depicting distribution of four monophyletic groups, especially the western boundary of the 4th cluster or key WE Steppe group, identified in relation to geographic features would help readers unfamiliar with larger Eurasia understand key findings regarding genetic structure. Some of key geographic features are shown in Fig 1a but not all of them eg Altay Mts.

Previous research is appropriately referenced. I cannot comment on the details of the statistics or genetic data.

This manuscript is novel, important and of extreme interest to many. I strongly support its publication in Nature. However, I recommend a revision designed to make key points in the introduction and discussion clearer and to strengthen accessibility for non-specialists.

Detailed comments

The introduction is not clearly written. The big questions and previous knowledge needs fleshing

out (brief) for general readers who are not following horse domestication or genetics, or the literature on social change in Bronze Age Eurasia.

As phrased, the assumptions regarding similarities between human and horse genetics at the beginning feel a bit simplistic. It is a current trend in genetic research, but why does it make sense to assume that migration rather than complex interactions of contact, exchange and trade characteristic of Bronze Age peoples would be driving horse variability? Clarify the argument for your position. Not a major point but distracting.

Related point. Suggest start the introduction with horses not human genetics. Clearer to delete first paragraph. Just add broad statement re horse significance to para 2 of introduction.

"Analyses of ancient horse genomes revealed that domestication occurred at least twice, once within the Eneolithic Botai culture (Central Asia (CA) steppes ~3,500 BCE), which provided the earliest evidence of bridling, milking and corralling^{4,6,7}. However, the origins for the second domestication have remained unknown, despite producing all modern domestic horses (DOM2)⁶."

This could be phrased a bit more clearly for non-specialists who don't know the history of research if you explain that Botai horses are not the same as horses ridden today world-wide before moving to Dom2

"Long-standing candidate regions for DOM2 domestication (eg. Iberia), were recently eliminated based on ancient genomic evidence⁸. Patterns of genetic variation at uniparentally"

For non-specialists clearer to list all candidate regions.

"DOM2 domestication, thus, truly revolutionized human mobility, long-range connectivity, trade and modes of warfare."

This last sentence of the introduction is important but unclear in phrasing and logic. I think the point is more nuanced than this sentence conveys. It needs less telegraphic treatment to be easily understood by the reader.

As mentioned below consider introducing Yamnaya and Sintashta cultures in the introduction.

General language issues

-Grammar (past/present tenses inconsistent, revealed or have revealed, remained or have remained, unspecified subjects, missing words) makes the introduction less clear.

-Word choice. Massive expansion – using in technical sense or colloquial?

-Here, we sequenced? Vs Here we discuss ...sequencing of xxx

Discussion

"Yamnaya horses had significantly more DOM2 genetic affinity than presumably wild horses from hunter-gatherer sites of the sixth mill BCE (NEO-NCAS, ~5,500-5200 BCE), possibly suggesting early horse management"

The significance of this point is not clear.

Yamnaya culture? Steppe pastoralists. How does this fit with previously mentioned social changes? Should be introduced for non-specialist reader. Previously only mentioned in abstract. Sintashta culture not mentioned in abstract or introduction but well introduced in discussion. Consider introducing Yamnaya and Sintashta cultures in the introduction

"This process likely involved first horseback riding only while chariotry soon contributed to further

the expansion. Spoke-wheeled chariots represent technological innovations that emerged ~2,000-1,800 BCE within the Trans-Ural Sintashta Culture, known for its fortified settlements and burials including chariots, horses, weaponry and warriors¹²."

A rewrite would clarify these sentences

"We acknowledge significant spatiotemporal variability and evidential bias towards elite activities, so do not discount additional, harder to evidence, factors in equine dispersion."

Great point

"However, increasing hierarchy within Bronze Age societies was likely associated with elite long-distance trade demands. The selection for horses with reduced back pathologies and enhanced docility would have both facilitated trading,"

This discussion assumes the readers knows that findings regarding genetic selection could be tied to back pathologies and enhanced docility. A clearer connection between results and this statement would be helpful.

"The adoption of this new institution, whether for warfare, prestige or both, probably varied between decentralized chiefdoms in Europe and urbanized civilisations in Western Asia. The results, thus, open up new avenues of research into these civilisations' trajectories."

Urban societies or urban states (vs mobile states) would be more specific than civilization.. Other issues, pastoralists not civilized? Most of this paper focuses on pastoralism. Some nuances to consider here.

(Colloquially people assume civilization = more advanced. 19C cultural evolution (civilization, barbarians, savages) problematic and still sensitive today). Just an observation.

Last observation-something to think about.

Figure 2b showing isolation by distance is really clear and compelling. Comparisons 3000-20,000 BP especially useful but except for this figure, BC is used throughout this paper. Using cal BP dates facilitate seamless comparisons with earlier periods. Archaeological authors will have opinions regarding whether this fits conventions for Eurasia, but BP is common in some world areas, earlier time periods, and in paleoenvironmental/environmental research. Domestication research would benefit from wider use of BP.

Author Rebuttals to Initial Comments:

Referee expertise:

Referee #1: equine genetics

Referee #2: population genetics, ancient DNA

Referee #3: animal domestication

Referees' comments:

Referee #1 (Remarks to the Author):

No other animal has had such a great historical impact on the ebb and flow of human civilizations. Yes, camels could go farther, cattle could feed us, and dogs have been with us longer, but only the horse was a king maker. The when and where of the beginnings of our relationship with the horse was the single greatest mystery among study of domesticated species. Previously these authors turned the history books on their heads, naming the oldest archeological record of equine-human collaboration as belonging to an entirely different species, the now wild-again Przewalski's horse. In a paleo-genomics tour-de-force their large collaborative team has now generated an impressive The older, slower, migration of domesticated horses with the expanse of the Yamnaya and PIE languages was a very logical hypothesis. I am indeed quite shocked that this wealth of new evidence supports an entirely different story. With the discovery that the Botai were not the origin of modern domesticated horses, my second guess would have been an older origin, and a slower entrance into domestication, as might have corresponded with the travel of the Yamnaya. What is most shocking, is the rapidity with which the new DOM2 line seems to have entirely replaced all other human-associated equid species. That this shift in species-use was not also associated with a shift in the human genetic diversity of the region is quite staggering. This implies two things: first, that the new species of DOM2 was vastly superior as a domesticated partner, and second, the peoples who developed and utilized these horses traded or gave them away. One need only look at the later example of Genghis Khan to see what potential power this new species conferred, and the magnitude of his impact on the human genome is readily visible as a result. And yet, the new horse took the ancient world not accompanied by force, but apparently, peaceably. To have convinced so many diverse cultures to abandon their local equine domesticates like the Tarpan and Przewalski's horse, replacing them with breeding stock from a foreign source, the new DOM2 horse must have been an enormous improvement in equine suitability for human purposes. From the physiological standpoint, the basis of this change is a tantalizing target for future study. The presented data on signatures of selection characteristic of the new DOM2 horse are remarkably clear for this type of analysis, suggesting that changes at just two loci may have driven much of the improvement in the suitability for domestication phenotype. These two loci hold genes with putative roles in features that remain key to the economic value of the domesticated horse today. Arguably, the behavior of the horse is only more important today than in pre-history, as their role has shifted from transportation and beast of burden to one of a companion animal and athletic partner. Intractable behavior in an animal that is easily 5-10 times (or more!) as massive as its human handler often quickly ends in disaster, and behavioral issues are often cited as the second most common rationale for relinquishing a horse to a rescue shelter (with incurable medical problems and ranking first). Given the complexities of the mammalian brain, documentation of the action of a neurodevelopmental pathway contributing to anxiety has obvious comparative biomedical applications.

POINT 17.

The identification of GSDMC is also a notable one, as the equine back remains and intractable source of performance limiting pain, and a source of frustration for owners and veterinarians alike. In addition to the cited role of this gene as a putative biomarker for spinal stenosis, it also recently appeared in a GWAS for chronic back pain (Suri et al. 2018, Genome-wide meta-analysis of 158,000 individuals of European ancestry identifies three loci associated with chronic back pain)

and this phenotype is perhaps more analogous to the modern clinical signs experienced by the equine athlete.

ANSWER 17.

We thank the reviewer for his/her suggestion, especially as we were not familiar with this publication. The suggested work is now cited, and the whole section has been rewritten to emphasize the interplay between the selection at GSDMC and ZFPM1:

Page 7, DOM2 biological adaptations: *“Human-induced DOM2 dispersal conceivably involved selection of phenotypic characteristics linked to horseback riding and chariotry. We therefore screened our data for genetic variants over-represented in DOM2 horses from the late third millennium BCE (Fig 4a). The first outstanding locus peaked immediately upstream of the GSDMC gene, where sequence coverage dropped at two L1 transposable elements in all lineages, excepting DOM2 (Fig 4c). The presence of additional exons in other mammals suggests that independent L1 insertions remodeled the DOM2 gene structure. In humans, GSDMC is a strong marker for chronic back pain²⁹ and lumbar spinal stenosis, a syndrome causing vertebral disk hardening and painful walking³⁰.*

The second most differentiated locus extended over ~16 Mb on chromosome 3 (Fig 4b), with ZFPM1 representing the gene closest to the selection peak. ZFPM1 is essential for the development of Dorsal Raphe serotonergic neurons involved in mood regulation³¹ and aggressive behavior³². ZFPM1 inactivation in mice causes anxiety disorders and contextual fear memory³¹. Combined, early selection at GSDMC and ZFPM1 suggests shifting use toward horses that were more docile, more stress-resilient and involved in new locomotor exercise, including endurance running, weight bearing and/or warfare.”

The Discussion also elaborates on possible societal consequences that may have resulted from DOM2 horse domestication during the Bronze Age:

Page 8, Discussion, end of first paragraph: *“In both cases, horses with reduced back pathologies and enhanced docility would have facilitated Bronze Age elite long-distance trade demands and have become a highly valued commodity and status symbol, causing rapid diaspora.”*

POINT 18.

As to the etiology of this syndrome in horses, there is much that is still unknown, though body size and conformation are likely risk factors. Modern domesticated horses vary considerably in the skeletal morphology of the back, encompassing polymorphisms in the number of thoracic and lumbar vertebrae, and in the rate of sacralization of the lumbar vertebrae (Stetcher, 1962, Anatomical Variations of the Spine in the Horse). I do not know that this variation has specifically been examined in historical or pre-historical samples, or for that matter, across equid species other than the rather early paper by Stetcher. The supplemental document notes that in many cases the skeletal remains sampled for DNA were unfortunately incomplete. It would be fascinating to know if these ancient species diverged for vertebral number, and therefore, likely in back conformation.

ANSWER 18.

We agree with the reviewer that this would provide exciting material for future studies, especially in light of the results from Fages et al. (2019), who reported selection signatures at the HOX-B/C loci in post-Muslim expansion horses. These loci are involved in vertebral formation during development. We cannot, however, develop this point further at that stage, as we did not collect any archaeological data on the number of vertebrae prior, during and after DOM2 domestication. It is worth emphasizing that collecting morphological data would actually be quite difficult, even if it constituted the main focus of future research. Prior to ritual burial of whole horses in graves, which does not happen until post-Sintashta, archaeological records are dominated by butchered mixed-up horse remains. There are often sections of spine but not whole vertebral columns from single horses. Levine's method, raised in POINT 19, works best scored along a whole spine. Whole horses, however, are also not available for all periods following Sintashta either. Sintashta is largely head and hoof (a common tradition). Andronovo tend not to bury horses in graves. Horse skeletons are only commonly found complete in Iron Age kurgans, rather too late to answer key questions associated with early domestication.

POINT 19.

Some evidence for back pathology has been observed in ancient, domesticated horses previously (Levine et al. 2004, Abnormal thoracic vertebrae and the evolution of horse husbandry) and may be worth mentioning.

ANSWER 19.

We knew the work from Levine et al. 2004. We have, however, decided not to cite this work for two reasons. First, the oldest DOM2 material investigated dates to the Scythian Iron Age, thus, post-dates DOM2 early spread by at least ~1,500 years, with back pathologies most likely reflecting other types of equestrianism (eg saddles). Second, the material examined at Botai was extremely limited, leading the authors themselves to acknowledge that '*Considering the enormous sample of material from Botai awaiting study, this conclusion must be regarded as extremely tentative.*'

POINT 20.

In any case, I hope I have articulated the reasons why the work is novel, and of extreme importance, not only to the field, but to related studies in biology and agriculture, as well as history and the social sciences. I also believe it will hold interest among a diverse audience. As befits this journal, it is well-written with accessible language to a scientific audience. The statistical methods are rigorous and well-applied to datasets that are described in excellent detail. The addition of the new Struc-f4 tool will be a valuable one to the field, providing a new way to visualize these highly dimensional comparisons. I have noted two typos in the main document that need attention: "bootstrap" rather than "bootstrap" and "spreading spread across".

ANSWER 20.

Done. Thanks for spotting these, and for all your positive and constructive criticism.

Referee #3 (Remarks to the Author):

The article from Librado et. al. reports the second (after Fages et al. 2019 of comparable size) most exhaustive collection of genomes from ancient horses across Eurasia produced so far (N = 264 or 242 see comment below).

Most importantly, they cover specimen from previously unreported time periods in key geographic regions such as the Bronze Age in the Pontic Steppe. They also produce the largest collection so far (to my knowledge) of very ancient, Paleolithic to Neolithic, horse genomes from various geographic regions across Eurasia (Including far east Siberia, the Urals, Europe and two Neolithic Anatolian wild specimen, I believe the first ever sequenced).

This impressive dataset allowed the authors to investigate the evolution horses and especially the spread of domestication across time and space at an unprecedented level, pinpointing a very plausible and convincing location and time period for the origin and spread of modern domestic horses (the so called DOM2 lineage). They further report indirect-insights into ancient human populations histories, showing that the genetic turnover in early Bronze Age Europe linked to the Yamnaya expansion and the spread of Indo-European languages was not accompanied by genetic turnover in horses. Whereas in Asia the spread of steppe people temporally coincides with horses' turnovers.

GENERAL OPINION

I find the size and novelty of the data generated and of the findings presented in this study to be worth for a Nature publication. I don't have major concerns regarding the experimental analyses, processing of the data and the validity of the population genetic analyses and the overall

interpretation of the main findings presented.

POINT 21:

Although I have two main points that I would like the authors to address. First, I find in general quite difficult to match the descriptions of the results in the text with the figures. This is especially true for Figure 1 and also it is in general hard to follow the link between the results from different analyses also because of use of acronyms in the plots that are not clearly defined in the text (see detail in Main points below).

ANSWER 21:

The reviewer is right and we also struggled to find an optimal balance between the amount of details needed and clarity. We thus sincerely thank the reviewer to have provided excellent suggestions for improving the figure (see ANSWER 23 below). In particular, removing individual sample names while defining all population groups further modeled and/or discussed in the text helped not only to significantly reduce the size of the figure but also to facilitate readability, especially when moving across figures and text.

POINT 22:

Second, I understand and agree that the findings of the Bronze Age post-Yamnaya expansion of DOM2 from the Pontic steppe is the main focus, but I still think that it would be worth dedicating more space or to better clarify the evolutionary relationship of the more ancient Eurasian horses (Paleolithic to Neolithic/Eneolithic) that I think constitute a second important novelty of the study and in the current draft there are some inconsistencies not fully solved or explained (see 2 major comments below).

ANSWER 22:

We agree with the reviewer that previous studies only provided a limited number of pre-domestication horse genomes. The more extensive characterization carried out in our study allowed us to identify a strong phylogeographic structure at the scale of the Eurasian continent. This was key to the identification of the DOM2 homeland, which is why this point is thoroughly covered in the revised manuscript. We also indicate that unsampled ghost lineages have existed, possibly even co-existed, and ultimately contributed to the genetic makeup of several Paleolithic to Neolithic/Eneolithic lineages. For the lack of space, and since the manuscript is focused on DOM2 origins, but also given the difficulties in properly modelling ghost influence(s) with current data, we have not extended the sections pertaining to the horse evolutionary history prior to domestication. Importantly, we have focused our analyses, especially statistical modelling, on those lineages that were key to build robust conclusions, and avoid speculation. Ongoing work in our group is aimed at extending the geographic and temporal sampling within Paleolithic to Neolithic/Eneolithic time periods, so as to build robust population models prior to domestication. We believe that a robust study focused on pre-domestication populations can be prepared only then. We have, however, addressed/clarified all inconsistencies, as per the reviewer's request (see ANSWERS below).

MAIN POINTS

POINT 23:

Throughout the text, names of geographic regions, genetic clusters and specific time periods are used to explain and interpret the results. Although, Figure 1 reports the entire set of ancient genomes, phylogeny (1b) and f4-struct-genetic-components (1d) plots with only the macro geographic coloring and the cryptic ID of each individual to orient the reader. I think this figure as it is could be a good supplementary figure as it exhaustively reports all the data but the main figure should allow the reader to match the descriptions of the main text. Maybe the authors can consider making a more summarized figure 1, collapsing some lineages in the phylogeny for example and providing "group" labels in the f4-Struct and phylogeny matching with the one mentioned in the text and the ones in the Extended Table 1 to better orient the reader. Beside this, I think that Fig 1a, is quite small and it's difficult to see properly the location of the samples on the map. Also, the

current time scale on the y-axis doesn't allow to appreciate the more recent and narrow time periods < 10,000 BP that is the more extensively discussed time period in the article. I would suggest to enlarge the time scale or add a zoom-in into this time window. To gain space I think that figure 1c (the f4-struct matrix) can be summarized with less entries or reduced since the macro clusters that the matrix distinguish would be still visible.

ANSWER 23:

We have followed the reviewer's suggestions:

- we have log-transformed the time axis so as to highlight the reflect the dominant density of samples within the second, third and fourth mill BCE (this also helps immediately spot the drastic ancestry change identified around ~2,200 BCE).
- we removed sample labels so as to reduce the size of the co-ancestry matrix
- we provide group labels that are consistent across all analyses, text, tables and figures
- ancestry profiles of each such groups are zoomed in on panel E, and placed according to their phylogenetic position.

POINT 24:

The second main point is that I cannot fully reconcile some patterns reported by different analyses (e.g. by the Strct-f4 analysis with the evolutionary models of OrientAGraph) concerning the more ancient lineages (while the main result, the origin and expansion of DOM2, is instead quite clear and solid). This is in part due to the difficulty of matching the clusters used for the analyses in Figure 3 with the description of the results in the text and the individual-based analyses of Figure 1 (so related to the first main point). Also, please check the definition of the acronyms used, for example I didn't find anywhere in the text UP-SFR and ELEN, used in OrientAGraph plots.

ANSWER 24:

All population clusters used for OrientAGraph modelling are now reported in Fig 1 and detailed in the corresponding captions. It is now also fully described in Supplementary Table 1, so that readers can identify the exact composition of each cluster at the individual level. We apologize for overlooking to have provided the full correspondence in the manuscript originally submitted.

More precisely, I cannot fully reconcile the following aspects:

POINT 25:

1) The CWC (Corded Ware) group in OrientAGraph is placed ancestral before the split of Botai/NEO-ANA and the lineages leading to DOM2. But in Strct-f4 the CWC associated horses are represented by ~50% of a component maximized in Botai and NEO-ANA and ~50% by a component maximized in what seems to me Paleolithic and Mesolithic Russia and Europe (The light blue and dark purple samples in Figure 1). The authors comment on this inconsistency in the Supplementary text as a cline of ancestry formed via expansions out-of-Anatolia through prolonged contact that would not be picked up by OrientAGraph. I wonder if another admixture graph method would be able to test the fit of such a model (see comment below).

ANSWER 25:

The gene flow between CWC and NEO-ANA is fully consistent with D-statistics in the form of (DOM2, NEO-ANA; CWC, URAL), which provides a significant excess of ABBA counts (77,446) vs BABA counts (69,433) ($|Z\text{-score}| \gg 3$). This is consistent with Struct-f4 ancestry profiles, showing an excess of the component colored in green on Fig 1ef, both for CWC and NEO-ANA. Hence, our interpretation. This has been clarified in the Supplementary Information, and likely indicates that OrientAGraph is not fully capable of accounting for the type of admixture characteristic of ancient horse populations (under the running options specified), which consists of continuous gene-flow (as per patterns of Isolation-By-Distance and genetic clines) rather than single unidirectional pulses. This is one of the reasons that motivated us to implement an additional analysis while preparing these revisions. This analysis builds on deep-neural network to learn from the spatial autocorrelation patterns in the genetic data pre-domestication what may be the most likely geographic origins of DOM2 horses (see the LOCATOR method published in *eLife* by Battey and colleagues (2020)). While relying on a totally different statistical approach, the results of this

analysis confirmed those from our other analyses and narrowed down the domestication centre to the lower Volga-Don region. This is now presented as Fig 3c.

POINT 26:

2) There are some groups ENEO-ROM, NEO-NCAS that are listed in the Extended Table 1 and seem to correspond with the samples ID clustering with C-PONT and TURG samples in Figure 1b and d (so related to the origin of the DOM2 lineage), that don't seem to be mentioned in the main text or included in other analyses. This potentially mixed genetic profiles seem also to be discordant with the OrientAGraph trees (analogous to the point 1 case) at least for C-PONT and TURG groups included in these trees that are modeled as un-admixed lineages. I wonder if the authors could comment and explore more the evolutionary relation of these lineages ancestral to the DOM2 branch.

ANSWER 26:

Thanks for this suggestion. We now more fully investigate the genetic affinities between all such populations, through extensive modelling with qpAdm. Full details are provided in the Supplemental Information as well as in Supplementary Table 3. In short, when considering ENEO-ROM and NEO-CAS as potential donor populations (under a qpAdm rotating scheme; Supplementary Table 3), they consistently appeared to not have genetically contributed to DOM2 (see also ANSWER 25 and the LOCATOR results). We, thus, used them as outgroup populations in further modelling and disregarded them in OrientAGraph analyses, as the graph reorientation step implemented in this method rapidly becomes intractable with increasing numbers of populations and migrations.

POINT 27:

3) In a previous study and here is confirmed that the Przewalski horses are the descendants of the ancient Botai horses. Here, while they do cluster together in the phylogeny and are close in the MDS plots, their f4-struct genetic profile seem to differ substantially: the Przewalski horses carrying high amount (~50%) of the ancestry maximized in DOM2 and lower proportion of other lineages including but at quite low proportion the component that is maximized in Botai horses. Can the authors comment on this? Are the Przewalski admixed with DOM2 and other sources or what can cause this Struct-f4 pattern? Is this high admixture in line with previous results?

ANSWER 27:

See ANSWER 41 below. This reflected a poor selection of the number of K ancestral groups. Przewalski's horses show consistent ancestry profiles for all other K values investigated (Supplementary Fig 3), including the now shown K=6 (Fig 1d). Our findings also confirm one our previous hypotheses (Gaunitz et al. 2018) revealing that Przewalski's horses experienced introgression from a ghost population during their feralization (ancestry component colored in purple on Fig 1f).

POINT 28:

4) The two NEO-ANA horses in the phylogeny of Figure 1b cluster among the C-PONT or TURG samples (the closer lineages ancestral to DOM2), but in all other analyses they result closer to the Botai group (e.g. same component in f4-struct and cladal in OrientAGraph). Could the authors comment on this?

ANSWER 28.

The reviewer is correct: NEO-ANA is misplaced in the Neighbor Joining phylogenetic reconstruction. We can demonstrate this by comparing patristic distances (ie pairwise genetic distances as inferred from the tree model) and raw pairwise genetic distances, in a classical 'goodness of fit' test. If the NJ tree provided a good representation of the true complexity underlying the data, such two measures of genetic distances should be perfectly correlated. They are not (Fig 1c), and NEO-ANA are precisely those horses with the largest deviation within the

most derived phylogenetic clade. This reflects the complex, admixed nature of NEO-ANA horses and is now fully acknowledged in the main section:

Page 5, Pre-domestication population structure: “Neighbor joining (NJ) phylogenomic inference revealed four geographically-defined monophyletic groups (Fig 1b). These closely mirrored clusters identified using an extension of the Struct-f4 method⁵ (Fig 1d-f; Supplementary Information; Extended Fig 2), except for Neolithic Anatolia (NEO-ANA), where the tree-to-data goodness of fit suggested phylogenetic misplacement (Fig 1c).”

Likewise, the goodness-of-fit test is further detailed in the Supplementary information. Furthermore, distributions of raw pairwise genetic distances to DOM2 horses confirm that NEO-ANA are in fact evolutionary more distant to DOM2 than other groups yet placed in their phylogenetic vicinity (eg C-PONT, NEO-CAS) (see the figure below). The phylogenetic misplacement of NEO-ANA is also apparent from our (1) direct ancestry genetic tests (showing NEO-ANA as the least related to DOM2, and measuring equivalent drift shared between NEO-ANA and DOM2, than between Botai and DOM2), (2) Struct-f4 MDS, (3) qpAdm (zero contribution to DOM2) and (4) OrientAGraph modelling. In fact, the NJ placement was the only analysis inconsistent with all other analyses.

Figure for revisions only. Raw pairwise genetic distances to DOM2 horses. Pairwise genetic distances between NEO-ANA and DOM2 horses were significantly larger than those between C-PONT or NEO-NCAS and DOM2 horses. This indicates that NEO-ANA is evolutionarily more distant to DOM2 than C-PONT and NEO-NCAS are, in contrast to the NJ phylogenetic reconstruction, but in agreement with all other analyses. Note how BOTAI horses are intermediately placed between ELEN and NEO-ANA, in line with them representing a mixture of these two parental sources, as inferred by OrientAGraph.

POINT 29:

Also, I find very interesting the similarity between the very geographically distant Eneolithic Botai and the Neolithic Anatolian horses (NEO-ANA), The authors comment on this in the supplementary as expansion of NEO-ANA into the CA steppe, this is indeed a possible explanation but what could be the drive for this expansion also in relation to what we know about ancient human populations?

If these are wild horses, why in comparison to the CWC case (point 1) we don't observe a cline of two ancestries blending together (the new incoming NEO-ANA and the older CA lineages) but instead a complete replacement (assuming it is in fact an expansion from Anatolia into the CA steppe)?

ANSWER 29:

Trying to link the genetic proximity between NEO-ANA and CA steppes (BOTA1) would be highly speculative. Indeed, specimen Shid1 shows a genetic ancestry profile mixing two main streams of ancestry: the first is colored in blue on Fig 1ef and is maximized in URAL; the second is colored in green and is maximized in NEO-ANA. The sample would correspond to a hybrid between those lineages. It was, however, located in the CA steppes and radiocarbon dated to prior 12kya, a time when horse domestication, and in fact no animal domestication except for the dog, already took place. Therefore, and recalling the patterns of isolation-by-distance, we believe that this observation is more suggestive of natural admixture gradient than human-induced animal translocation. This connectivity was most likely mediated South of the Caucasus mountains and the Caspian Sea, as (1) no such hybrids could be identified during the Neolithic period in WE steppes and (2) following the results of the LOCATOR analyses, projecting early DOM2 origins outside of the CA steppes and Anatolia.

We anticipate that the genetic cline could be better characterized following further sampling across the region between Anatolia and the Central Asian Steppes, South of the Caspian Sea.

POINT 30:

Finally, in the OrientAGraph trees some key ancient lineages have not been included (e.g. ancient Iberians, IBE, and the ancient Urals, S-Ural and also other groups that are listed in the Extended Table 1 but not mentioned in the text). I think it would be worth to explore more the relationship between all these deep Eurasian lineages identified with the new data. This is just a suggestion, but I was wondering if the authors would consider running a tool like qpGraph or other demographic modeling methods (SFS based for example) that could test the fit of specific complex evolutionary models user-specified as an independent validation of the ones obtained with OrientAGraph. It could maybe help in reconciling the evidences from different analyses by testing for specific admixture events/gene flows vs monophyletic split between lineages.

ANSWER 30:

qpGraph leverages the same information as qpAdm, which was used in this study. For revising our manuscript, we have followed the recommendations from Harney and colleagues (2021) and implemented a rotating scheme in which DOM2 were modeled as combinations of all possible 2-ways, 3-ways and 4-ways mixtures of donor populations. The analyses are summarized in the Supplementary Table 3. They consistently rule out any contribution from other lineages than those selected for OrientAGraph modelling (excepting sample Ukr11 showing limited sequence coverage). This demonstrates that no other populations were needed to be considered to address the origins of DOM2, which represents the main focus of this study. In the light of ghost contributions, and the strong patterns of isolation-by-distance observed, we believe that the limited samples sequenced pre-domestication and the unidirectional, single pulses assumptions of Graph-based modeling (including qpGraph) strongly limit our capacity to understand the full details of the population history pre-domestication. In fact, this was supported by our attempts to use the recently improved Admixtools2 framework while preparing these revisions, which returned a wide range of graph topologies of similar scores. This reflects lack of power in the data currently available to properly infer pre-domestication population history. Hence, our decision to only focus on the robust, and least speculative, conclusions. See also ANSWER 22 and ANSWER 26.

MINOR POINTS

Main text

POINT 31.

- Figure 2 - I would increase the visibility of the pie charts in the map. Suggestions: the sizes of the maps could be increased by decreasing the scatterplots on the right. Also, in Figure 2b since autosomes and chromosome X show the same trend only one of the two plots could be kept and the other added in the supplementary or the two trends could be superimposed in the same plot.

ANSWER 31.

Done. We followed the reviewers' suggestions. Therefore, the size of individual pie-charts was increased to improve readability. Additionally, the maps displayed are now wider while the panels showing geographic-to-genetic distance correlations are smaller. Finally, the isolation-by-distance trends for the autosomes and the X chromosome are merged within a single graph.

POINT 32.

-Page 6: "Neighbor joining phylogenomic inference revealed highly supported monophyletic groups". See main point 2, there seem to be admixture between several lineages according to Struct-f4 and OrientAGraph so I would not generalize about monophyletic lineages.

ANSWER 32:

We agree with the reviewer that NJ phylogenomic inference does not account for possible admixture and should only be taken as a first, preliminary attempt to describe the data. We now report the goodness of fit between pairwise distances as estimated from the tree model (ie patristic distances) and raw pairwise distances, in an attempt to identify those lineages potentially misplaced (Fig 1c). Considering a dataset with extremely limited and uniform sequencing errors (as it is the case in this study), such goodness of fit is expected to reflect admixture events and their magnitude (as individuals are forced to single locations in the tree, while in fact consisting of composite ancestries). We indeed found that specimens associated with greater NJ error (Fig 1c), thus more likely to be misplaced in the tree, had more complex compositions in ancestry, as measured by Struct-f4 (Fig 1ef).

Additionally, admixture was explicitly modeled using different approaches, including qpAdm (Supplemental Information) and OrientAGraph (Fig 3b).

In order to tone down the description of the NJ phylogenetic clusters, we have now rephrased the corresponding sentence and paragraph into:

Page 5, Pre-domestication population structure: *"Neighbor joining (NJ) phylogenomic inference revealed four geographically-defined monophyletic groups (Fig 1b). These closely mirrored clusters identified using an extension of the Struct-f4 method⁵ (Fig 1d-f; Supplementary Information; Extended Fig 2), except for Neolithic Anatolia (NEO-ANA), where the tree-to-data goodness of fit suggested phylogenetic misplacement (Fig 1c)."*

See ANSWER 28.

POINT 33:

-Page 6: Where are the *Equus lenensis* lineages mentioned in the text located on the tree and Struct-f4 plots (figure 1)?

ANSWER 33:

This lineage is now fully identified in Figs 1ef, together with all other lineages described in the text and used for data analyses.

Page 5, last sentence: *"The most basal cluster included *Equus lenensis* (ELEN), a lineage identified in Northeastern Siberia from the Late Pleistocene (LP) to the late fourth millennium BCE^{5,14,15}."*

POINT 34:

-Page 6: "A second group comprises horses from Europe, including UP Romania, Belgium, France, and Britain as well as sixth-to-third mill BCE horses from Spain to Scandinavia, and France to Hungary, Czechia and Poland (HCP)". It is a long list of countries but acronym HCP refers only to Hungary Czechia and Poland? It is bit confusing phrased this way.

ANSWER 34:

Agreed. We now removed the HCP acronym here, and throughout the whole text.

POINT 35:

-Page 6: "The ancestry profile of sample RONPC06_Rom_m34801 indicated that the Anatolian expansion into the Romanian Carpathians started before ~35,000 BCE." I might be missing something but this sample is much older than the two NEO-ANA ones. I don't think an expansion from Anatolia into Romania is the only possibility explaining the f4-strict genetic profile of these samples especially given that the Romanian specimen is much older.

ANSWER 35:

Agreed. This sentence has been removed. We interpreted their ancestry profiles as an expansion from the ancestors of NEO-ANA into Eastern and Central Europe, as the ancestry colored green is maximized in Anatolia and dilutes into Central and Eastern Europe. We implicitly assumed that NEO-ANA represents a stable population over time, also in ancestry composition, given that Anatolia was a well-known LGM refugium. Considering the time difference between samples, however, other less parsimonious explanations could have taken place, potentially leading to similar ancestry profiles. To prevent any misinterpretation, we thus removed this sentence from the main manuscript.

POINT 36:

- Page 7: "qpAdm modelling indicated a genetic profile acquiring DOM2 ancestry following gene flow into CWC-related horses from southern Thrace (Kan22_Tur_m2386), but not from Crimea (Ukr11_Ukr_m4185)." Where can I see this result? no reference to figure or table.

ANSWER 36:

This result was provided as a table in the Supplemental Table 3, as described below: Page 7, **Expansion of steppe-related pastoralism**: "*It reached at best 12.5% in one Hungarian horse dated to the mid-third millennium BCE and associated with the Somogyvár-Vinkovci Culture (CAR05_Hun_m2458). qpAdm¹⁷ modelling indicated that its DOM2 ancestry was acquired following gene flow from southern Thrace (Kan22_Tur_m2386), but not from the Dnieper steppes (Ukr11_Ukr_m4185) (Supplementary Table 3).*"

The corresponding Supplementary Table 3 has been now referenced in the text, and consistently shows, for feasible models, that *CAR05_Hun_m2458* is best explained as a mixture between Kan22_Tur_m2386 (~50% or more) and local horses from Central Europe.

POINT 37.

- Page 7 DOM2-specific biological adaptation

I think the Fst signal in the two genomic regions identified is very strong so I say this as a minor comment and suggestion: given the rich dataset of ancient lineages across Eurasia and a model of their evolutionary relationships, would the author consider running a more evolution-informed positive selection test beside the Fst? For example, PBS (first introduced in Yi et al. 2010) is an extension for Fst testing for deviations in a 3-population tree given a test group, a sister group and an outgroup. Also, maybe is beyond the scope of this study and maybe the sample size is too small but it would be interesting to see if it possible to pinpoint signals of adaptation to the different natural environments among the different very ancient lineages (e.g. the Paleolithic/Neolithic Siberian lineages compared to the European ones).

ANSWER 37.

We agree that this represents an extremely interesting question. It, however, extends beyond the scope of the present study and in fact provides the basis for an ongoing study in our laboratory where we apply a range of techniques extending the PBS framework to over 3 lineages (namely, LSD, Librado & Orlando 2018; and Ohana, Cheng et al. 2017). Preliminary results have revealed limited power in detecting adaptation in non-DOM2 horses, which also motivated the DOM2/non-DOM2 binning presented in our study. We believe that this was caused both by the limited size of the non-DOM2 panel (relative to the number of co-existing lineages) and the diverse, pervasive

ghost influence in some, but not all, non-DOM2 lineages. Our current strategy is thus to expand our current dataset for pre-domestication horses in order to both gain power and better characterize ghost lineages and admixture histories.

POINT 38:

- Page 11 typo:

“For which were sequencing was done”, remove “were”.

ANSWER 38:

Done.

POINT 39:

-Page 13 “Was investigated using the f3-statistic-based ancestry... implemented in qpAdm” qpAdm is solely based on f4-statistics.

ANSWER 39:

Thanks. This was a typo. We meant f4-statistics.

POINT 40.

-Page 13 “Struct-f4, ancestry components and Multidimensional Scaling”

Struct-f4 is also introduced in Fages et al. 2019 but here it seems presented as a newly developed method. Can the authors comment on what has been newly developed since Fages et al. 2019?

ANSWER 40.

Despite having similar objectives, the struct-f4 method presented in Fages et al. 2019 was based on a completely different statistical design, and computational implementation. The former version made use of Maximum Likelihood optimization to return estimates for three spatial coordinates of each sample. The inferred Euclidean distances between samples were proportional to the observed f4 permutations. The latter, new version of the struct-f4 method relies instead on a MCMC algorithm to learn from the f4-permutations the difference in allele frequencies between each pair of samples, which provides a measurement of their shared genetic drift. This shared genetic drift can be easily visualized through Multi-Dimensional Scaling, as shown in Fig 3a. Additionally, the shared genetic drift can serve as input for a statistical mixture model, reflecting the ancestry proportions of a pre-defined number of ancestral groups. Such a mixture model is now implemented in Struct-f4 for the first time, and is equivalent to other tools such as Admixture, excepting that no prior assumption is made on the equilibrium of ancestral groups. This is because f4-statistics reflect evolutionary distances between true internal nodes in a population tree, which represents true (as opposed to inferred) ancestral components. See also ANSWER 27 and ANSWER 11.

Earlier implementation of Struct-f4 is now fully acknowledged in the main text:

Page 5, one-to-the-last sentence: “*These closely mirrored clusters identified using an extension of the Struct-f4 method⁵ (Fig 1d-f; Supplementary Information; Extended Fig 2), except for Neolithic Anatolia (NEO-ANA), where the tree-to-data goodness of fit suggested phylogenetic misplacement (Fig 1c).*”

Supplementary text

POINT 41:

- Page 31 Application of Struct-f4 to empirical data:

How is the K-clustering done?

ANSWER 41.

See ANSWER 27 above. Similar to what done with other tools such as Admixture, Struct-f4 is run for an increasing number of K ancestral clusters (Extended Fig 3), and the improvement in the model likelihood gauges for the adequacy of opting for specific K-values. In our case, we originally opted for K=5, mostly due to the number of phylogenetic clusters identified, but with the revisions,

we scrutinized the likelihood space, which reaches a plateau for $K=6$, concurrently with consistent ancestry profiles, to finally opt for $K=6$. As a result, Figs 1def shows genetic ancestry profiles for this number of K ancestral groups. The whole batch of K values from 4 to 9 is still made available through Extended Fig 3.

POINT 42.

Why the modern genomes were excluded? How would this batch effect influence f_4 -stat? Could the authors elaborate more on this?

ANSWER 42:

We apologize for the confusion. Modern genomes were not excluded from such analyses as their genetic ancestry profiles were in fact displayed in Fig 1ef. We, however, excluded those f_4 -permutations in which modern genomes were placed as H1 or H2 (assuming topologies in the form of (H1,H2;H3,Donkey), where H_i represent horse samples). Modern genomes could still contribute in the analyses if placed as H3 together with two ancient samples placed as H1 and H2. The placement in H3 is known to be more robust to systematic errors, for example resulting from difference in sequence quality or technical batch effects, as documented extensively in the supplemental information file of Orlando and colleagues (2013).

POINT 43:

- Page 28: The authors perform quite an accurate filtering for C to T ancient DNA damage (e.g. UDG-treatment, PMDtools to filter damaged reads, 5bp end trimming) but at the end they seem to retain only transversions for all the analyses presented. If the transitions are never used why performing these quality controls? Maybe I miss-understood something, could the authors elaborate on this?

ANSWER 43:

Our strict data filtering procedure was aimed to increase the signal-to-noise in our sequence analyses, as the genetic proximity between some of candidate domestication sources was thin, we feared that even small but consistent patterns of error sharing in the data could result in spurious results. To achieve this, we binned sequence data according to the possible presence of post-mortem DNA damage using PMDtools. This provided a first, dominant class of alignments that were only trimmed for five base pairs at read termini (ie starts and ends). In this class, all substitution classes, including transitions, potentially remaining along each alignment were conserved. The second class of alignments included those reads suspected to carry post-mortem DNA damage. These were first treated with mapDamage2 to downgrade the base quality scores of all positions underlying potential damage. This only affected transitions as post-mortem DNA damage mainly results in C→T and G→A mis-incorporations. The resulting alignments were further trimmed for 10 base pairs at read termini, as the post-mortem signal was found to propagate to such an extent. This second procedure left transversions as the only substitution type contributing to sequence divergence. The merger of both classes of alignments provided the final BAM alignment files that were used in our analyses. This provided more data than simply blunt trimming 10 base pairs (ie 5 positions were saved for each sequence of the dominant class of alignments). These alignments were directly used for genotyping loci driving important Mendelian traits (such as performance and coat-coloration), as the underlying causative variants included transitions and transversions alike (Extended Fig 7). Disregarding all transitions would have hampered investigations of key loci for horses, such as the so-called speed allele at MSTN. For all other analyses, we used the more conservative data matrix in which only transversions were used, as the number of sites considered was sufficiently large and did not justify the risk of considering possibly unidentified remaining spurious nucleotide mis-incorporations resulting from post-mortem DNA damage.

POINT 44.

- Extended Table 1: I counted 242 new genomes produced in “this study” not 264. Please double check the numbers. Also, can the authors elaborate on why they considered only subset of samples from previously published studies (for example from Fages et al 2019, only 13 were

considered)?

ANSWER 44.

In this study, we have indeed generated new sequence data for a total of 264 ancient horses. A total of 245 of these were analyzed here for the first time. However, 19 samples already provided sequence data in some of our previous work (fully referred to in the main text), that we have merged to the new data generated here. We have clarified this in the Supplementary Table 1, indicating 'This study+X', where X directly refers to the original publication first reporting sequence data (eg 'This study+Gaunitz et al. 2018'). The reviewer is right that we have characterized more than 13 ancient horse genomes in our previous work (in fact, 87 above 1-fold coverage in the study from Fages et al. 2019). However, not all such genomes, eg the most recent (last 2,000 years), were relevant to investigate the early stages of horse domestication at the transition between the third and second mill BCE. Additionally, with the exception of 3 divergent genomes not contributing to DOM2 genomic makeup (*E. lenensis*, as confirmed with new samples sequenced here, eg samples RN109_Rus_m13477, R17x2_Rus_INF, and Rus30x31_Rus_m3435), we only included those genomes that were previously characterized using strictly similar technologies. This includes DNA extraction, USER-treatment, DNA library construction, library amplification and sequencing. This was done to avoid technical batch effects in the final data, that may have been introduced by different sequencing platforms (ie Illumina HiSeq2500 vs HiSeq4000) and/or different library types (our previous work was mostly based on single library indexing vs triple indexing here). This added to our data filtering procedure (Supplemental Information; see ANSWER 43) to provide the best possible data quality, which was confirmed through the temporal signals recovered (Extended Fig 1). Adopting such strict procedures proved important in the light of the strong genetic homogeneity found within DOM2 horses, and the close general proximity measured between some of the possible ancestral groups, such as C-PONT and TURG. Limiting the overall number of genomes considered also helped significantly reduce already extra-ordinary extensive computational times underlying most analyses (as the number of f4-permutations increases quadratically with the number of samples considered).

This is now succinctly described in the main text:

Page 5, Ancient Genome Sequencing, 2nd paragraph: *"DNA quality allowed shotgun sequencing of 264 ancient genomes at 0.10X-25.76X average coverage (239 above 1X), including 19 for which further sequencing added to previously reported data. Enzymatic¹³ and computational removal of post-mortem DNA damage produced high-quality data with derived mutations decreasing with sample age, as expected if mutations accumulate through time (Extended Fig 1). We added ten published modern genomes, and nine ancient genomes characterized with consistent technology or covering relevant time periods and locations, to obtain the most extensive high-quality genome time-series for horses."*

Referee #4 (Remarks to the Author):

This is a landmark study of the domestication and spread of horses. It will be of interest to scholars in a broad range of fields including domestication, evolution, genetics, archaeology, history, agriculture, and to the general public.

As the authors say, domestication of horses transformed the subsistence, mobility, and geo-politics of ancient societies. Understanding the process of horse domestication has been challenging, though, because the mobility of early horse keepers resulted in interbreeding among distant horse populations, domestic and wild. Several generations of zooarchaeological and genetic studies identified early domestic horses at Botai in N Kazakhstan and suggested the possible existence of a second center further west. Genetic research demonstrated that the Botai horses were not ancestral to modern domestic horses but data bearing on a second center has been elusive. What makes this study by Librado et al so significant is the identification of the lower Don-Volga region of the Western Eurasian Steppe as the second center and origin of all modern horses ridden today.

The sample size, broad regional and temporal coverage and clear shifts in population structure

~4200 years ago, from geographically isolated to a lack of population structure characteristic of mobile domesticates, makes this a robust finding. The authors sequenced by far the largest sample of ancient horse genomes studied to date, including 264 horses from a range of time periods and Eurasian regions. Beyond genetic information, the data sets presented in SI include detailed site context and a large and important new radiocarbon data-base, which strengthen the reliability of results.

The identification of the time and place of a second center for horse domestication is a sufficiently important and novel finding to warrant publication in Nature. But there is more. The authors present data that challenges longstanding assumptions about invasion, horses and the history of societies in Europe. Largely based in linguistics, it has long been argued that steppe herders associated with the Yamnaya culture expanded into Europe ~5000 years ago bringing Indo-European languages and steppe horses to the region. Horse-based mobility and warfare were thought to be important factors underlying this population movement. This is not supported by the results of Librado et al, which show no evidence for introduction of steppe horses to western Europe 5000 years ago. Instead, data from this study reveal that horses domesticated in the Don Volga region of the western steppe's spread rapidly to the west about 1000 years later, ~4000 BP associated with chariot use. Interestingly, Librado et al's data indicate a different pattern driving the eastward spread of modern horses into central Asia, where Don Volga horses spread with the steppe Sintashta culture eastward in association with chariots and Indo-Iranian language. This ultimately resulted in replacement of early domestic horses of Botai in Khazakstan and horses further east (eg Mongolia) with modern domestic horses. As a result, this study provides fundamental insights into the origin of modern horses and diverse social mechanisms that resulted in Bronze Age social change, as well as shifts in domestic horse biodiversity.

This research also finds evidence for selection on locomotion and behavior with horse domestication, which is an important contribution to understanding variability in sites of selection in animal domestication. The exceptional preservation of aDNA in Eurasian regions in which horses were domesticated also make this sample significant for domestication studies broadly. To date, there are few ancient genomic samples of this scope for large mammals.

POINT 45.

Clarity and accessibility

Overall, the narrative structure is clear and logical. However, the introduction is not clearly written due to assumption of previous knowledge, phrasing and word choice (see below). The abstract is clear, the discussion mostly clear and conclusions are appropriate. This manuscript has a distinguished group of authors and I suggest greater recruitment for some edits by those with extensive experience contextualizing horse domestication for non-specialists.

ANSWER 45.

The whole manuscript has been extensively revised in light of the reviewer's comment but also of the strict space limitations imposed by the journal. The two sections most extensively edited were the Introduction, which has now been replaced by a fully referenced Summary Paragraph (as per editorial request), and the Discussion, which now consists of two paragraphs instead of five. We also ensured to avoid jargon, and provide contextual information on archaeological material and culture that would otherwise remain unknown by most readers.

POINT 46.

An additional map depicting distribution of four monophyletic groups, especially the western boundary of the 4th cluster or key WE Steppe group, identified in relation to geographic features would help readers unfamiliar with larger Eurasia understand key findings regarding genetic structure. Some of key geographic features are shown in Fig 1a but not all of them eg Altay Mts.

ANSWER 46.

Fig 1 has now been clarified, including full referencing to the four monophyletic groups as well as to the population labels used in the different analyses. The dominant ancestry component in their genetic profile also serves as basis for the choice of color in the map shown as Fig 1a. Space restriction precludes additional contextual maps (the addition of Fig 3c, which projects the most likely origins of DOM2 horses, however, now provides a simple, highly impacting figure summarizing the main finding of our study). The size of the map panel is, however, extended to maximal possible limits to enhance geographic resolution, and all geographic features discussed in the main text are now annotated.

Previous research is appropriately referenced. I cannot comment on the details of the statistics or genetic data.

This manuscript is novel, important and of extreme interest to many. I strongly support its publication in Nature. However, I recommend a revision designed to make key points in the introduction and discussion clearer and to strengthen accessibility for non-specialists.

POINT 47.

Detailed comments

The introduction is not clearly written. The big questions and previous knowledge needs fleshing out (brief) for general readers who are not following horse domestication or genetics, or the literature on social change in Bronze Age Eurasia.

As phrased, the assumptions regarding similarities between human and horse genetics at the beginning feel a bit simplistic. It is a current trend in genetic research, but why does it make sense to assume that migration rather than complex interactions of contact, exchange and trade characteristic of Bronze Age peoples would be driving horse variability? Clarify the argument for your position. Not a major point but distracting.

ANSWER 47.

This sentence has been removed together with the removal of the Introduction section. However, regarding human migrations at the time, we refer to the original phrasing in corresponding publications and interpretations. It is true that shifts in human ancestry profiles can result from different societal dynamics, but the overly dominant narrative associated to the early Bronze Age has been, and remains, framed as '*massive migrations*', to only quote the title of Haak et al. (2015) *Nature* article, which reads: "*Massive migration from the steppe was a source for Indo-European languages in Europe*".

POINT 48.

Related point. Suggest start the introduction with horses not human genetics. Clearer to delete first paragraph. Just add broad statement re horse significance to para 2 of introduction.

ANSWER 48.

Agreed: the paragraph on human genetics was removed, together with the remaining part of the Introduction as per editorial request but also as it turned fully redundant with the original Abstract. The four sentences of the fully-referenced Summary paragraph are now aimed to provide context and establish the rationale for our study. It reads as follows:

Page 5, Summary Paragraph: "*Horse domestication fundamentally transformed long-range mobility and warfare¹. However, modern domesticates do not descend from the earliest domestic horse lineage associated with archaeological evidence of bridling, milking and corralling²⁻⁴ at Botai, Central Asia ~3,500 BCE (Before Common Era)³. Other long-standing candidate regions for horse domestication, such as Iberia⁵ and Anatolia⁶, were also recently challenged. Therefore, the genetic, geographic and temporal origins of modern domestic horses remain unknown.*"

POINT 49.

"Analyses of ancient horse genomes revealed that domestication occurred at least twice, once within the Eneolithic Botai culture (Central Asia (CA) steppes ~3,500 BCE), which provided the earliest evidence of bridling, milking and corralling^{4,6,7}. However, the origins for the second

domestication have remained unknown, despite producing all modern domestic horses (DOM2)⁶."

This could be phrased a bit more clearly for non-specialists who don't know the history of research if you explain that Botai horses are not the same as horses ridden today world-wide before moving to Dom2

ANSWER 49.

See ANSWER 48 above. We also avoided to introduce technical terms, such as DOM2, already in the Summary paragraph.

POINT 50.

"Long-standing candidate regions for DOM2 domestication (eg. Iberia), were recently eliminated based on ancient genomic evidence⁸. Patterns of genetic variation at uniparentally"

For non-specialists clearer to list all candidate regions.

ANSWER 50.

Done. See ANSWER 48 above.

POINT 51.

"DOM2 domestication, thus, truly revolutionized human mobility, long-range connectivity, trade and modes of warfare."

This last sentence of the introduction is important but unclear in phrasing and logic. I think the point is more nuanced than this sentence conveys. It needs less telegraphic treatment to be easily understood by the reader.

ANSWER 51.

This sentence has been removed while rewriting the original Abstract and Introduction into a Summary paragraph, as per editorial request. We also restructured our Discussion into two main paragraphs, the first of which specifically addressing the implications of our findings for horse and human dispersion, trade and warfare. It reads as follows:

Page 8, Discussion, end of first paragraph: *"The globalization stage started later, when DOM2 horses dispersed outside their core region, first reaching Anatolia, the lower Danube, Bohemia and Central Asia by ~2,200-2,000 BCE, then Western Europe and Mongolia soon after, ultimately replacing all local populations by ~1,500-1,000 BCE. This process first involved horseback riding as spoke-wheeled chariots represent later technological innovations emerging ~2,000-1,800 BCE within the Trans-Ural Sintashta Culture⁷. The weaponry, warriors and fortified settlements associated with this culture possibly arose in response to increased aridity and competition for critical grazing lands, intensifying territoriality and hierarchy³⁷. This may have provided the basis for the conquests over the subsequent centuries that resulted in an almost complete human and horse genetic turnover in CA steppes^{11,21}. The expansion to the Carpathian basin³⁸, and possibly Anatolia and the Levant, involved a different scenario where specialized horse trainers and chariot builders spread with horse trade and riding. In both cases, horses with reduced back pathologies and enhanced docility would have facilitated Bronze Age elite long-distance trade demands and have become a highly valued commodity and status symbol, causing rapid diaspora. We, however, acknowledge significant spatiotemporal variability and evidential bias towards elite activities, so we do not discount additional, harder to evidence, factors in equine dispersal."*

As mentioned below consider introducing Yamnaya and Sintashta cultures in the introduction.

POINT 52.

General language issues

-Grammar (past/present tenses inconsistent, revealed or have revealed, remained or have remained, unspecified subjects, missing words) makes the introduction less clear.

ANSWER 52.

Done. We ensured that consistent tenses were used within each paragraph.

POINT 53.

-Word choice. Massive expansion – using in technical sense or colloquial?

ANSWER 53.

The term '*Massive migration*' reflects that used in the title of the original publication by Haak and colleagues (2015) (see ANSWER 47).

POINT 54.

-Here, we sequenced? Vs Here we discuss ...sequencing of xxx

ANSWER 54.

The original sentence has been rephrased into (as per *Nature* recommendations for writing Summary paragraph):

Page 5, Summary Paragraph: *"Here, we pinpoint the Western Eurasian steppes, especially the lower Volga-Don region, as the homeland of modern domestic horses. Furthermore, we map the population changes accompanying domestication from 273 ancient horse genomes."*

POINT 55.

Discussion

"Yamnaya horses had significantly more DOM2 genetic affinity than presumably wild horses from hunter-gatherer sites of the sixth mill BCE (NEO-NCAS, ~5,500-5200 BCE), possibly suggesting early horse management"

The significance of this point is not clear.

ANSWER 55

We don't know whether Yamnaya pastoralists domesticated the horse or not. We noticed that the genetic makeup of Yamnaya horses was showing the typical DOM2 genetic ancestry. This does not imply that they were domesticated, as the wild ancestor of DOM2 horses also had a typical DOM2 genetic profile. This has now been rephrased to suggest that be domesticated or not, Yamnaya horses have not spread farther than the native Yamnaya geographic range.

Page 8, Discussion, beginning of first paragraph: *"Our work solves long-standing debates about the origins and spread of domestic horses. While horses living in the WE steppes in the late fourth/early third millennium BCE were the ancestors of DOM2 horses, there is no evidence that these horses facilitated the expansion of the human genetic steppe ancestry into Europe^{8,9}, as previously hypothesized⁷. Instead of horse-mounted warfare, declining populations during the European late Neolithic³⁵ may, thus, have opened up an opportunity for a westward expansion of steppe pastoralists. Yamnaya horses at Repin and Turganik carried more DOM2 genetic affinity than presumably-wild horses from hunter-gatherer sites of the sixth millennium BCE (NEO-NCAS, ~5,500-5,200 BCE), which may suggest early horse management and herding practices. Regardless, Yamnaya-pastoralism did not spread horses far outside their native range, similar to the Botai horse domestication, which remained a localized practice within a sedentary settlement system^{2,36}."*

POINT 56.

Yamnaya culture? Steppe pastoralists. How does this fit with previously mentioned social changes? Should be introduced for non-specialist reader. Previously only mentioned in abstract. Sintashta culture not mentioned in abstract or introduction but well introduced in discussion. Consider introducing Yamnaya and Sintashta cultures in the introduction

ANSWER 56.

Reformatting the Abstract and the Introduction within a single Summary paragraph limited the space available for cultural description. We limited thus the context provided to minimum (Yamnaya people are described as steppe pastoralists, and the Sintashta Culture associated with

spoke-wheeled chariotry). The discussion, however, provides all necessary background information to inform on the societal consequences of our findings. See ANSWER 51 above.

POINT 57.

"This process likely involved first horseback riding only while chariotry soon contributed to further the expansion. Spoke-wheeled chariots represent technological innovations that emerged ~2,000-1,800 BCE within the Trans-Ural Sintashta Culture, known for its fortified settlements and burials including chariots, horses, weaponry and warriors¹²."

A rewrite would clarify these sentences

ANSWER 57.

Done.

Page 7, Discussion, first paragraph: *"This process first involved horseback riding as spoke-wheeled chariots represent later technological innovations emerging ~2,000-1,800 BCE within the Trans-Ural Sintashta Culture⁷. The weaponry, warriors and fortified settlements associated with this culture possibly arose in response to increased aridity and competition for critical grazing lands, intensifying territoriality and hierarchy³⁷."*

POINT 58.

"We acknowledge significant spatiotemporal variability and evidential bias towards elite activities, so do not discount additional, harder to evidence, factors in equine dispersion."

Great point

"However, increasing hierarchy within Bronze Age societies was likely associated with elite long-distance trade demands. The selection for horses with reduced back pathologies and enhanced docility would have both facilitated trading,"

This discussion assumes the readers knows that findings regarding genetic selection could be tied to back pathologies and enhanced docility. A clearer connection between results and this statement would be helpful.

ANSWER 58.

These statements have been restructured in the Discussion, and the logic order reverted. It now reads:

Page 8, Discussion, end of first paragraph: *"In both cases, horses with reduced back pathologies and enhanced docility would have facilitated Bronze Age elite long-distance trade demands and have become a highly valued commodity and status symbol, causing rapid diaspora. We, however, acknowledge significant spatiotemporal variability and evidential bias towards elite activities, so we do not discount additional, harder to evidence, factors in equine dispersal."*

Additionally, the section on selection has been refocused on the back pathologies and docility, to maximize impact and clarity.

Page 7, DOM2 biological adaptations: *"Human-induced DOM2 dispersal conceivably involved selection of phenotypic characteristics linked to horseback riding and chariotry. We therefore screened our data for genetic variants over-represented in DOM2 horses from the late third millennium BCE (Fig 4a). The first outstanding locus peaked immediately upstream of the GSDMC gene, where sequence coverage dropped at two L1 transposable elements in all lineages, excepting DOM2 (Fig 4c). The presence of additional exons in other mammals suggests that independent L1 insertions remodeled the DOM2 gene structure. In humans, GSDMC is a strong marker for chronic back pain²⁹ and lumbar spinal stenosis, a syndrome causing vertebral disk hardening and painful walking³⁰.*

The second most differentiated locus extended over ~16 Mb on chromosome 3 (Fig 4b), with ZFPM1 representing the gene closest to the selection peak. ZFPM1 is essential for the development of Dorsal Raphe serotonergic neurons involved in mood regulation³¹ and aggressive

behavior³². ZFPM1 inactivation in mice causes anxiety disorders and contextual fear memory³¹. Combined, early selection at GSDMC and ZFPM1 suggests shifting use toward horses that were more docile, more stress-resilient and involved in new locomotor exercise, including endurance running, weight bearing and/or warfare."

POINT 59.

"The adoption of this new institution, whether for warfare, prestige or both, probably varied between decentralized chiefdoms in Europe and urbanized civilisations in Western Asia. The results, thus, open up new avenues of research into these civilisations' trajectories."

Urban societies or urban states (vs mobile states) would be more specific than civilization.. Other issues, pastoralists not civilized? Most of this paper focuses on pastoralism. Some nuances to consider here.

(Colloquially people assume civilization = more advanced. 19C cultural evolution (civilization, barbarians, savages) problematic and still sensitive today). Just an observation.

ANSWER 59.

Absolutely correct. We have now edited the last two concluding sentences accordingly:

"The adoption of this new institution, whether for warfare, prestige or both, probably varied between decentralized chiefdoms in Europe and urbanized states in Western Asia. The results, thus, open up new research avenues into the historical developments of these different societal trajectories."

POINT 60.

Last observation-something to think about.

Figure 2b showing isolation by distance is really clear and compelling. Comparisons 3000-20,000 BP especially useful but except for this figure, BC is used throughout this paper. Using cal BP dates facilitate seamless comparisons with earlier periods. Archaeological authors will have opinions regarding whether this fits conventions for Eurasia, but BP is common in some world areas, earlier time periods, and in paleoenvironmental/environmental research. Domestication research would benefit from wider use of BP.

ANSWER 60.

We now provide all dating information in BCE, as the majority of the literature on early horse domestication, including our own work, makes use of such units. We believe that this will facilitate cross studies comparisons. We, nonetheless, have added a column to Supplementary Table 1, which provides a direct time equivalence between BCE and BP dates.

Reviewer Reports on the First Revision:

Referee #1 (Remarks to the Author):

Credit to the authors for taking the comments from all reviewers and pulling together a much improved manuscript. I have just a few suggestions for the main text.

The last sentence of the abstract begins "This contrasts with the situation...". I think here the word "scenario" might work just a little better, as it took me two or three attempts to grasp the concept as currently worded.

Other reviewers noted the difficulty in switching from BCE to BP years, especially for readers outside of the field. The only remaining use of BP is in the second sentence of the Results-Ancient genome sequencing section. I suggest converting this to BCE as well, for consistency (I immediately found myself attempting the mental math to compare this earlier date with the range given just before it in the phrase. Skipping the need for mental math will improve readability).

Second paragraph, page 6, please spell out "LP". The manuscript has a challenging number of acronyms and abbreviations. Again, for the reader outside of the field I think this one in particular presents a troublesome stumbling block.

First sentence of the section entitled "The origins of DOM2 horses". Many changes in the organization and structure of the manuscript improved the clarity of the text. This sentence though, is still a bit rough. I suggest revising this to read "The western lower Volga-Don region not only possessed moderate NEO-ANA ancestry, but also was the first region where the typical DOM2 ancestry began to dominate (colored...."

Or something similar. As phrased currently it's a little wordy.

Page 7, under "DOM2 biological adaptations", second paragraph: delete the phrase "representing the gene" and instead state "...with the ZFPM1 gene closest to...".

The changes to the figures incorporating comments from the first review are well executed and have greatly improved the communicative power of these images. I particularly like the overlay of the skyline plots for the X and autosomes.

One last comment with regards to figure 4: while the EquCab3.0 genome assembly is broadly an improvement over the EquCab2.0 assembly, I have found that in some regions EquCab 3.0 is prone to in silico structural artifacts that actually make the EquCab2.0 assembly the preferred reference, especially for gene models. It is a rare occurrence, but when it happens to your gene of interest it can be distressing! Have the authors examined gene annotation for both assemblies to double check the gene/exon order presented in figure 4?

Referee #3 (Remarks to the Author):

The authors replied exhaustively to all my comments, solved or explained convincingly all the points I raised. Also, the readability of the revised text has greatly improved and the main figures are now clear and easier to follow. So I don't have additional comments and I encourage the publication of this exciting and ground breaking study.

Referee #4 (Remarks to the Author):

This paper is now clear and compelling. The rewrite has greatly strengthened this paper, which represents a major step forward in understanding of horse domestication and spread.

Author Rebuttals to First Revision:

Referee #1 (Remarks to the Author):

Credit to the authors for taking the comments from all reviewers and pulling together a much improved manuscript. I have just a few suggestions for the main text.

POINT 1. The last sentence of the abstract begins "This contrasts with the situation...". I think here the word "scenario" might work just a little better, as it took me two or three attempts to grasp the concept as currently worded.

RESPONSE 1. Done, the sentence now reads: "*This contrasts with the scenario in Asia where Indo-Iranian languages, chariots and horses spread together, following the early second millennium BCE Sintashta culture^{11,12}.*" (page 5, last sentence of the Summary paragraph).

POINT 2. Other reviewers noted the difficulty in switching from BCE to BP years, especially for readers outside of the field. The only remaining use of BP is in the second sentence of the Results-Ancient genome sequencing section. I suggest converting this to BCE as well, for consistency (I immediately found myself attempting the mental math to compare this earlier date with the range given just before it in the phrase. Skipping the need for mental math will improve readability).

RESPONSE 2. Done, the sentence now reads: "Sampling targeted previously under-represented time-periods, with 201 radiocarbon dates spanning 44,426-202 BCE, and five beyond 50,250-47,950 BCE (Supplementary Table 1)." (page 5, Ancient genome sequencing, last sentence of the first paragraph).

POINT 3. Second paragraph, page 6, please spell out "LP". The manuscript has a challenging number of acronyms and abbreviations. Again, for the reader outside of the field I think this one in particular presents a troublesome stumbling block.

RESPONSE 3. Done, Late Pleistocene (one such change was also necessary in the Supplementary Methods for consistency).

POINT 4. First sentence of the section entitled "The origins of DOM2 horses". Many changes in the organization and structure of the manuscript improved the clarity of the text. This sentence though, is still a bit rough. I suggest revising this to read "The western lower Volga-Don region not only possessed moderate NEO-ANA ancestry, but also was the first region where the typical DOM2 ancestry began to dominate (colored...." Or something similar. As

phrased currently it's a little wordy.

RESPONSE 4. Done, the sentence now reads: "*The western lower Volga-Don not only possessed moderate NEO-ANA ancestry but also was the first region where the typical DOM2 ancestry component (colored in orange on Fig 1ef) became dominant during the sixth millennium BCE.*" (page 6, The origins of DOM2 horses, first sentence).

POINT 5. Page 7, under "DOM2 biological adaptations", second paragraph: delete the phrase "representing the gene" and instead state "...with the ZFPM1 gene closest to..."

RESPONSE 5. Done, the sentence now reads: "*The second most differentiated locus extended over ~16 Mb on chromosome 3, with the ZFPM1 gene closest to the selection peak.*" (page 7, DOM2 biological adaptations, second paragraph, first sentence).

The changes to the figures incorporating comments from the first review are well executed and have greatly improved the communicative power of these images. I particularly like the overlay of the skyline plots for the X and autosomes.

POINT 6. One last comment with regards to figure 4: while the EquCab3.0 genome assembly is broadly an improvement over the EquCab2.0 assembly, I have found that in some regions EquCab 3.0 is prone to in silico structural artifacts that actually make the EquCab2.0 assembly the preferred reference, especially for gene models. It is a rare occurrence, but when it happens to your gene of interest it can be distressing! Have the authors examined gene annotation for both assemblies to double check the gene/exon order presented in figure 4?

RESPONSE 6. Thanks for this suggestion. To investigate this, we connected to Archive!Ensembl, which provides access to gene model annotations for the EquCab2 genome. However, *GSDMC* (or gasdermin C) returns no entries, suggesting that this gene was not annotated in the previous reference genome. This was confirmed as our attempts on the UCSC genome browser to Liftover the gene boundaries on EquCab3 to EquCab2 also failed. Similarly, the gene *ZFPM1* was not annotated on EquCab2, and only *ZFPM2* could be identified (on another chromosome, number 9, not 3). Therefore, our discovery on DOM2 specific adaptations would have remained impossible, even in the presence of the extensive ancient genome panel generated, should the quality of the horse genome reference assembly and the underlying gene models had not been improved. See below for screenshots of the results returned by Archive!Ensembl.

[Redacted]